# Cooperative thalamocortical circuit mechanism for sensory prediction errors

Shohei Furutachi[1✉], Alexis D. Franklin[1], Andreea M. Aldea[1], Thomas D. Mrsic-Flogel[1✉] & Sonja B. Hofer[1✉]

The brain functions as a prediction machine, utilizing an internal model of the world to anticipate sensations and the outcomes of our actions. Discrepancies between expected and actual events, referred to as prediction errors, are leveraged to update the internal model and guide our attention towards unexpected events[1–10]. Despite the importance of prediction-error signals for various neural computations across the brain, surprisingly little is known about the neural circuit mechanisms responsible for their implementation. Here we describe a thalamocortical disinhibitory circuit that is required for generating sensory prediction-error signals in mouse primary visual cortex (V1). We show that violating animals' predictions by an unexpected visual stimulus preferentially boosts responses of the layer 2/3 V1 neurons that are most selective for that stimulus. Prediction errors specifically amplify the unexpected visual input, rather than representing non-specific surprise or difference signals about how the visual input deviates from the animal's predictions. This selective amplification is implemented by a cooperative mechanism requiring thalamic input from the pulvinar and cortical vasoactive-intestinal-peptide-expressing (VIP) inhibitory interneurons. In response to prediction errors, VIP neurons inhibit a specific subpopulation of somatostatin-expressing inhibitory interneurons that gate excitatory pulvinar input to V1, resulting in specific pulvinar-driven response amplification of the most stimulus-selective neurons in V1. Therefore, the brain prioritizes unpredicted sensory information by selectively increasing the salience of unpredicted sensory features through the synergistic interaction of thalamic input and neocortical disinhibitory circuits.

Although our senses are continuously bombarded with inputs from the environment, only a subset of the sensory information is perceived or affects behaviour. Our brains thus prioritize important sensory features among irrelevant ones[11]. Psychological and physiological studies indicate that the brain generates internal predictions about incoming sensory information and compares them with actual sensory inputs[5–10], resulting in prediction errors when sensory inputs do not match internal predictions. Error signals could mediate prioritization of unexpected—and therefore possibly relevant—sensory inputs, and be used to update internal predictions[5–10]. Indeed, sensory prediction-error signals have been observed in multiple cortical areas upon the violation of subjects' predictions[9,10,12–16]. Despite their prevalence across the brain and importance for perception and learning, it is still unclear what information is encoded by sensory prediction error signals, how they affect cortical networks, and through which circuit mechanisms they arise.

To study the neural implementation of predictive processing in cortical sensory networks, we used a paradigm in which head-fixed, food-deprived mice running on a cylinder navigated a virtual corridor in which they developed spatial predictions about stimulus identity at particular locations along the corridor. The corridor walls displayed alternating grating stimulus patterns (grating A–grating B–grating A–grating B) separated by distinct landmarks (Fig. 1a). The visual stimuli appeared abruptly when mice reached the corresponding position in the corridor and were presented at constant visual flow independent of the running speed of the mice, to enable precise control over stimulus features and timing (Methods). Upon reaching the reward zone at the end of the corridor, mice received a liquid food reward and their position was reset to the beginning of the corridor, starting a new trial. Mice traversed the corridor many times for five days of training (90 ± 48 trials (traversals) per day, 59 ± 21 s per trial; mean ± s.d.) during which the sequence of the gratings was identical on every trial. On day six (C session), the identity of the stimulus at the fourth position changed in a subset of trials: a novel grating stimulus C was first shown instead of the second grating stimulus B in 10% of trials (block 1, 160 trials in total; Fig. 1a). Subsequently, stimulus C was shown at the fourth location in all trials (block 2, 40 trials). Previous studies using similar paradigms showed that mice form predictions of which stimuli to expect at specific locations in the corridor[14,17]. Accordingly, we found that mice interrupted their running behaviour when their expectations were violated by encountering stimulus C (Extended Data Fig. 1a,b), although running speed was not always a reliable behavioural indicator of the increasing familiarity of the novel stimulus with repeated exposure (Extended Data Fig. 1a,b).

[1]Sainsbury Wellcome Centre, University College London, London, UK. ✉e-mail: s.furutachi@ucl.ac.uk; t.mrsic-flogel@ucl.ac.uk; s.hofer@ucl.ac.uk

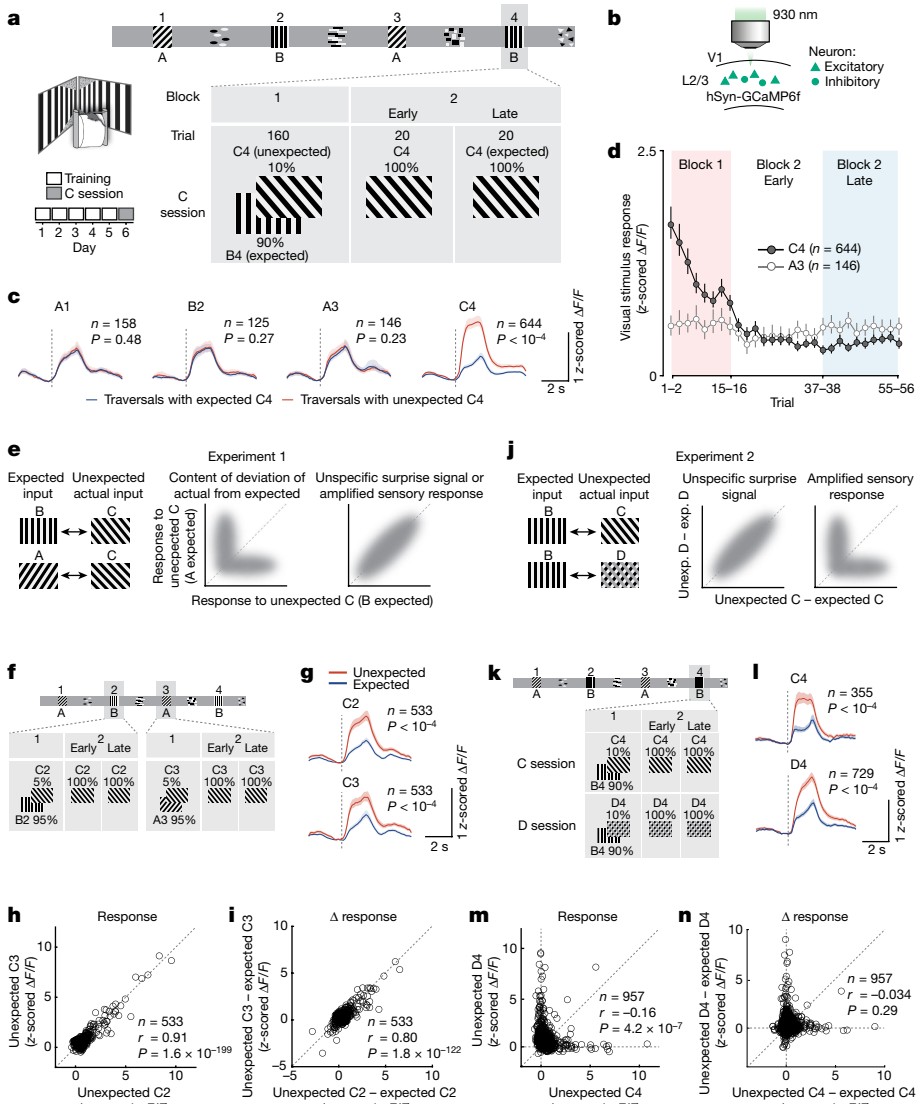

**Fig. 1 | Prediction errors amplify unexpected visual information. a**, Structure of the virtual corridor and experimental design. **b**, Two-photon calcium imaging approach. **c**, Average calcium responses to different stimuli in corridor traversals with unexpected C (red, in block 1) and with expected C (blue, in late block 2). V1 neurons responsive to the presented stimulus in unexpected C trials, expected C trials or both were included. Dotted vertical lines indicate grating onsets. Data from 9 mice; *P* values from hierarchical bootstrapping test. See also Extended Data Fig. 2b for combined responses of all grating-responsive neurons. **d**, Average calcium responses to stimuli C4 (dark grey) and A3 (light grey) during C trials across trials and blocks. **e**, Thought experiment to disambiguate information represented by prediction errors. **f**, Experimental design. Stimulus C was presented at position 2 (C2) or at position 3 (C3) in 5% of trials each in block 1. **g**, Average calcium responses to unexpected (red) and

expected (blue) stimuli C2 (top) and C3 (bottom). Data from 9 mice, *P* values from hierarchical bootstrapping test. **h**, Responses to unexpected stimulus C2 plotted against responses to unexpected C3 for individual V1 layer 2/3 neurons; Pearson correlation. **i**, Difference in response strength between unexpected and expected C2 plotted against response strength difference between unexpected and expected C3 responses for individual V1 layer 2/3; Pearson correlation. **j**, Similar to **e**, but for a second thought experiment. Exp., expected; unexp., unexpected. **k**, Experimental design. Stimuli C or D were presented at position 4 (C4 and D4) in 10% of trials in different sessions. **l**, Same as **g**, but for stimuli C4 (top) and D4 (bottom). Data from 5 mice. **m**, Same as **h**, but for stimuli C4 and D4. **n**, Same as **i**, but for stimuli C4 and D4. **c,d,g,l**, Data are mean ± bootstrap 95% confidence intervals. See also Extended Data Figs. 1–3.

We recorded neural activity of layer 2/3 neurons in V1 using two-photon calcium imaging[18] (Fig. 1b and Methods), and observed a stronger response to a visual stimulus that was novel and therefore unexpected (stimulus C in block 1) compared with the same stimulus when it was expected (stimulus C in second half of block 2, $P < 1 \times 10^{-4}$, hierarchical bootstrapping test; Fig. 1c and Extended Data Fig. 2a,b), consistent with previous studies in humans, non-human primates and rodents[9,10,12–14,16,19–23]. This difference in neural responses could not be explained by a drift in general behavioural state, such as arousal or task engagement across the imaging session, as responses to expected grating stimuli A and B were constant throughout the session

(Fig. 1c,d, all *P* > 0.05; see also Extended Data Fig. 2a,b). The increased response to unexpected visual stimuli could also not be accounted for by changes in the animal's motor behaviour (Extended Data Fig. 1). Specifically, the response increase was not correlated with running speed, stimulus-induced deceleration or pupil size (Extended Data Fig. 1). V1 responses to an unexpected stimulus were slightly larger when this stimulus was encountered closer to the reward location (Extended Data Fig. 3a–c), consistent with potentially higher behavioural relevance of visual stimuli at such a location[17]. However, the increased neural responses to unexpected stimuli were independent of reward-related signals in V1 (Extended Data Fig. 3c–e).

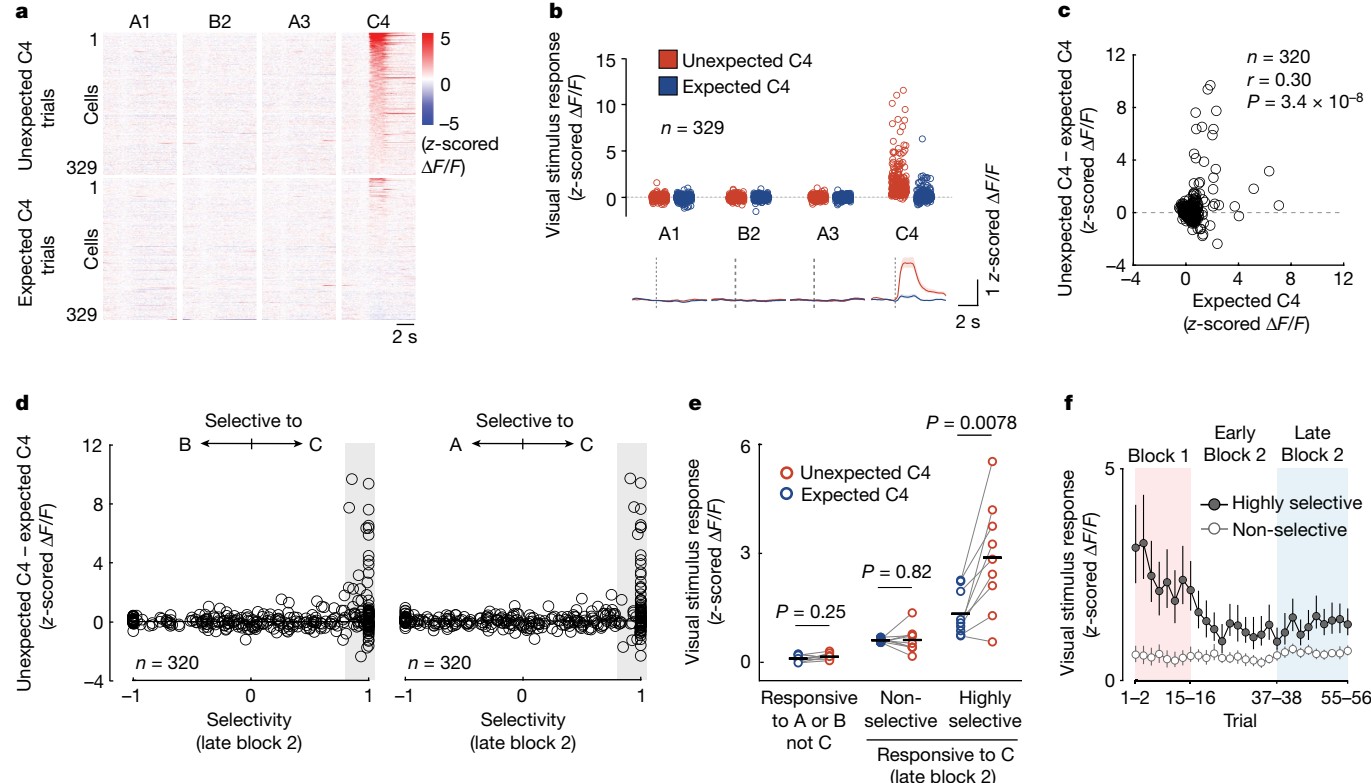

**Fig. 2 | Prediction error specifically boosts the most stimulus-selective neurons. a**, Trial-averaged responses of all prediction-error-responsive neurons ($n = 329$ cells, 9 mice) to all grating stimuli in traversals with unexpected C4 (top; block 1) and expected C4 (bottom; late block 2), sorted by response to unexpected C4. **b**, Same as **a**, but average response strength of individual neurons (top) and mean calcium responses of all neurons (bottom). Shading indicates bootstrap 95% confidence intervals. **c**, Difference in response strength to unexpected (block 1) and expected C4 (late block 2) for all grating-responsive cells in late block 2, plotted against response to expected C4 in late block 2 for individual neurons. Pearson correlation; 9 mice. **d**, Left, difference in response strength between unexpected and expected C4 responses of individual neurons, plotted against their response selectivity to stimulus C versus stimulus B in late block 2 (Methods) for all neurons

responsive to at least one of the grating stimuli in late block 2. −1 indicates only responsive to B, +1 indicates only responsive to C, and 0 indicates similar responses to both. Right, same as on the left but for response selectivity to stimulus C versus stimulus A. **e**, Mean responses to expected (blue) and unexpected (red) C4, of V1 neurons responsive to A or B (left), of non-selective (middle; responsive to C with selectivity < 0.6) and highly selective neurons (right; responsive to C with selectivity towards C, compared to B > 0.8) in late block 2. Data are mean responses for individual mice ($n = 9$), black horizontal bars indicate mean across mice. Two-sided signed-rank test. **f**, Mean calcium responses to stimulus C4 across all trials of highly selective (dark grey, $n = 77$ cells from 9 mice) and non-selective (light grey, $n = 53$) grating C4-responsive cells in late block 2. Error bars indicate bootstrap 95% confidence intervals. See also Extended Data Figs. 4–6.

Neural responses to grating stimulus C strongly decreased over time as mice encountered the visual stimulus more often, and responses were asymptotic within several trials in block 2 when stimulus C was encountered in every trial (Fig. 1d and Extended Data Fig. 2g). This gradual decrease in response cannot simply be explained by visual adaptation to repetitive stimuli, as C was only presented every $448 \pm 364$ s (mean ± s.d.) in block 1, owing to the considerable length of the virtual corridor. Of note, responses also significantly increased when the familiar stimulus A was presented at an unexpected location in the corridor (Extended Data Fig. 4a–d, $P < 1 \times 10^{-4}$), and some neurons responded to the omission of an expected stimulus[14] (Extended Data Fig. 2e,f, $P < 1 \times 10^{-4}$ for visual stimulus omission). The elevated neural response to an unexpected stimulus does thus not only constitute a response to stimulus novelty, but also is most consistent with a prediction-error signal. Moreover, the gradual decrease and eventual cessation of the prediction-error signal after repeated exposure to the novel stimulus at the same location indicates that mice learned to update their spatial expectations about stimulus identity over time.

## Nature of prediction-error signals

What information sensory prediction error signals represent is currently unclear. According to theories of predictive coding, prediction

error signals have been proposed to encode the difference between predicted and actual visual input[5–8] (encoding the content of how the actual visual input is different from predictions). However, error responses could also represent a more unspecific surprise signal, encoding only the magnitude of the deviation without its content (also called unsigned prediction error[9]), or could enhance the representation of unpredicted sensory input (encoding the content of the actual input). We designed further experiments to disambiguate between these options. First, in a small subset of trials, we presented stimulus C at one of two different locations in the corridor, at which either stimulus B (position 2) or stimulus A (position 3) were expected (experiment 1; Fig. 1e,f). Grating stimulus C elicited a stronger response in V1 in either location when it was unexpected (Fig. 1g). In these two instances the actual visual stimulus is the same, but the predictions are likely to be different. If the prediction-error signal contains information about the predicted stimulus and/or how the actual stimulus deviates from this prediction, V1 responses should differ to stimulus C at the two different locations. However, V1 prediction-error responses to the unexpected stimulus C in the two locations were notably similar (Fig. 1h,i; $r = 0.91$, $P = 1.6 \times 10^{-199}$ and $r = 0.80$, $P = 1.8 \times 10^{-122}$ (Pearson correlation for Fig. 1h,i, respectively); Extended Data Fig. 3g), indicating that—at least at the level of individual neurons in V1—the sensory prediction-error signal contains little information about how the actual input differs from predictions.

Next, we tested whether the prediction-error signal represents the actual visual input or instead a non-specific surprise or motor-related signal (experiment 2; Fig. 1j,k). To this end we introduced an additional unexpected visual stimulus D that was presented at corridor position 4 in a subset of trials in a separate imaging session of the same neuronal populations (Fig. 1j,k). Both stimuli C and D evoked strong prediction-error responses when they were unexpected (Fig. 1l and Extended Data Fig. 2c,d). Neural responses to C and D should be similar if they simply represented a non-specific surprise signal, or activity related to surprise-triggered movement, such as deceleration in response to an unexpected stimulus. However, most neurons responded strongly to only one of the two unexpected stimuli, and V1 population responses to these stimuli were thus different and specific to stimulus features (Fig. 1m,n and Extended Data Fig. 5a–e). This was also the case when comparing prediction-error responses to two more similar visual stimuli (two gratings of different orientation; Extended Data Fig. 5l–p).

Indeed, V1 neurons that responded to an unexpected stimulus (that is, grating C) often also responded to the same stimulus when it was expected, but not to gratings A or B (Fig. 2a–c). Importantly, only visually driven neurons that responded highly selectively to a stimulus showed amplified responses when this stimulus was unexpected (Fig. 2d–f; $P = 0.0078$ for highly selective cells), whereas more broadly tuned neurons that also responded to other visual stimuli did not show prediction-error signals (Fig. 2e,f: $P = 0.82$ for non-selective cells). This selective amplification was equally evident in the V1 responses to a different unexpected stimulus (stimulus D; Extended Data Fig. 6a–h), and could not be explained by differences in response strength between selective and non-selective neurons (Extended Data Fig. 6i,j). Notably, increased V1 activity in response to a familiar stimulus (A) at an unexpected location was also restricted to those visually responsive neurons selective for the presented stimulus (Extended Data Fig. 4e,f), indicating that selective amplification of visual information that is unexpected may be a general feature of sensory prediction-error signals in V1.

In addition to visually driven neurons, a subset of non-visually responsive neurons was also recruited by prediction errors (Fig. 2a and Extended Data Fig. 4i). Responses of these neurons were nevertheless highly stimulus-selective, and restricted to specific unexpected stimuli (Extended Data Fig. 5f–k). Neurons responding to the unexpected omission of a stimulus constituted an additional V1 population, which was not activated when the omitted stimulus was instead replaced by a different, unexpected stimulus (Extended Data Fig. 5q–z). This indicates that negative prediction errors (responses to the unexpected absence of a stimulus or event[10,14]) are not significantly contributing to the V1 prediction-error signal in response to a novel, unexpected stimulus.

Together, these experiments indicate that the prediction-error signal evoked in layer 2/3 of V1 by unexpected visual stimuli is not a non-specific surprise or a difference signal about how the visual input deviates from the animal's predictions. Instead, prediction error signals are specific to the features of the unexpected visual input and amplify the activity of neurons that respond highly selectively to the unexpected visual features, thereby selectively increasing the salience of unpredicted—and therefore potentially most relevant—sensory information.

## Circuits mediating V1 prediction-error signals

We next examined the circuit mechanisms by which sensory prediction error signals are implemented in V1 networks. VIP inhibitory interneurons in V1 receive cortical top-down and neuromodulatory inputs, and can disinhibit local principal cells through prominent inhibitory connections onto somatostatin-expressing (SOM) inhibitory interneurons[24–28], providing a circuit for top-down gain modulation of sensory responses[29,30]. VIP cells have also been shown to respond strongly to novel, but not familiar, visual stimuli[20,23]. To assess whether VIP interneuron activity is important for prediction-error signals in V1, we first examined how VIP interneurons respond to unexpected and expected visual information by using the experimental paradigms described in Fig. 1k (Fig. 3a). VIP interneurons were suppressed by expected visual stimuli, but strongly responded to unexpected visual stimuli (Fig. 3b–d and Extended Data Fig. 7a,b), consistent with previous studies[15,20,23]. VIP neurons also responded to familiar stimuli encountered at an unexpected location (Extended Data Fig. 8a–d), showing that they are not only activated by novel stimuli, but also by sensory prediction errors more generally. Prediction-error responses of VIP neurons were much less selective than those of putative excitatory neurons in V1: many VIP neurons responded to both unexpected stimuli C and D (Extended Data Fig. 7c–e). Responses of VIP interneurons decreased over time as mice encountered the same stimulus more often, in parallel with the gradual cessation of the prediction-error signal in the layer 2/3 network (Fig. 3d; see also Fig. 1d), suggesting that the recruitment of VIP interneurons may be causally related to the generation of prediction-error signals in V1.

To test whether the recruitment of VIP interneurons is required for the prediction-error signal in the general V1 population, we optogenetically silenced VIP interneurons as mice encountered expected or unexpected visual stimuli while recording calcium responses of V1 layer 2/3 neurons (Fig. 3e–g and Methods). This manipulation was highly effective as VIP neurons were fully inactivated during light stimulation (Extended Data Fig. 9a–c). Inactivating VIP neurons significantly reduced the responses of V1 layer 2/3 cells to unexpected visual stimuli (Fig. 3f, middle, $P < 1 \times 10^{-4}$; Extended Data Fig. 10a–h), whereas it had no effect on responses to expected visual stimuli A and B (Fig. 3f, left; $P = 0.24$), consistent with the specific recruitment of VIP interneurons by unexpected sensory stimuli (Fig. 3a–d). Furthermore, the effect of VIP inactivation on individual V1 layer 2/3 cells could not be explained by light artefacts (Extended Data Fig. 9g,h), and it was not uniform, but highly correlated with how strongly V1 neurons were facilitated by prediction errors, much more so than with their visual response strength: neurons with the strongest prediction-error signal were the ones that were most suppressed by VIP interneuron inactivation (Fig. 3g and Extended Data Fig. 10c,e,f). V1 prediction-error signals in response to familiar stimulus A at an unexpected location were also abolished when VIP neurons were inactivated (Extended Data Fig. 8e,f), demonstrating that the recruitment of VIP neurons is required more generally for prediction-error signals in layer 2/3 of V1, rather than specifically for V1 signals related to stimulus novelty.

We next explored the identity of the long-range inputs to V1 that could mediate the activation of VIP neurons by prediction errors. The pulvinar is a higher-order visual area in thalamus, also called lateral posterior nucleus in mice, that integrates information from many cortical and subcortical areas and sends prominent feedback projections to V1[31–36]. Notably, pulvinar projections to V1 carry information about visual input that is not predicted by the animal's own actions, indicating that the pulvinar conveys sensory–motor prediction errors to V1[31]. To test whether pulvinar projections to V1 also signal prediction errors arising from spatial predictions of visual input in our task, we used two-photon imaging to record calcium signals from pulvinar axons in V1[31]. Calcium activity of pulvinar axons was strongly and non-selectively boosted when a visual stimulus was unexpected (Fig. 3h–k and Extended Data Fig. 7h–n), and this prediction-error response decreased with repeated exposure to the same stimulus, with a time course similar to responses in V1 neurons (Fig. 3k). Pulvinar axons were also activated by a familiar stimulus at an unexpected location (Extended Data Fig. 8g–i).

To determine whether pulvinar input to V1 is required for prediction-error signals in V1 neurons, we optogenetically inactivated pulvinar axons in V1 while recording calcium responses of V1

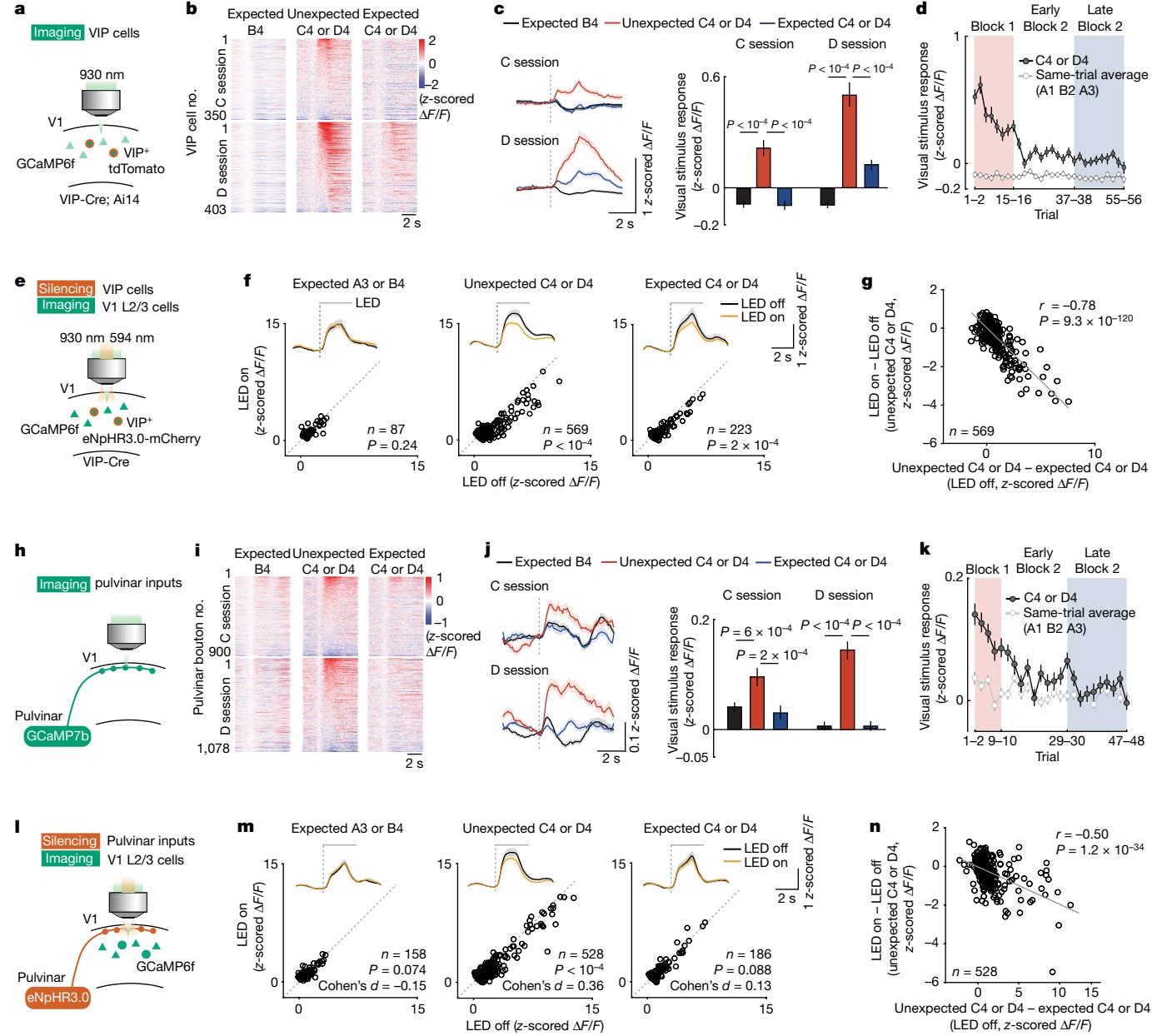

**Fig. 3 | Activity of VIP interneurons and pulvinar input is required for V1 prediction-error signals. a**, Experimental design. Calcium activity of VIP cells in V1 layer 2/3 was recorded during the experiment depicted in Fig. 1k. **b**, Single-cell responses for all VIP cells (individual rows) in the session with unexpected stimulus C (top; C session, $n = 350$ cells from 7 mice) and with unexpected stimulus D (bottom; D session, $n = 403$ cells from 7 mice) to expected B4 (left), unexpected C4 or D4 (middle; block 1) and expected C4 or D4 (right; late block 2), sorted by response strength to unexpected C4 or D4. **c**, Cell- and trial-averaged stimulus responses of all VIP cells in **b**. $P$ values from hierarchical bootstrapping test with Bonferroni correction. **d**, Average calcium responses of all VIP cells to grating stimulus C4 or D4 (dark grey) and other gratings in the same trial (average of A1, B2 and A3, light grey) over time. **e**, Experimental design. Calcium activity of V1 layer 2/3 cells was recorded while VIP cells were optogenetically silenced during visual stimulus presentation.

**f**, Top, cell- and trial-averaged responses of V1 neurons significantly responsive to the presented visual stimuli with (amber) or without (black) VIP silencing. Bottom, responses of individual neurons to the visual stimulus indicated above during VIP cell silencing (LED on), plotted against responses to the same stimulus in control trials (LED off). $P$ values from hierarchical bootstrapping test, from 9 mice. **g**, Effect of VIP neuron silencing (LED on − LED off during unexpected stimulus C4 or D4) plotted against the strength of prediction-error signals (response to unexpected C4 or D4 − response to expected C4 or D4); Pearson correlation. **h**–**k**, Same as **a**–**d**, but for calcium responses of pulvinar axonal boutons in V1 layer 1. **l**–**n**, Same as **e**–**g**, but the activity of V1 layer 2/3 cells was recorded while pulvinar axons in V1 were optogenetically silenced. **c**,**d**,**f**,**j**,**k**,**m**, Data are mean ± bootstrap 95% confidence intervals (shading or error bars). See also Extended Data Figs. 7–10.

layer 2/3 neurons (Fig. 3l–n). This manipulation—light stimulation of eNpHR3.0-expressing pulvinar axons in V1—reduced activity of pulvinar axons, but had only a partial effect (Extended Data Fig. 9d–f). Nevertheless, suppressing pulvinar input to V1 specifically reduced the responses of V1 layer 2/3 neurons to unexpected visual stimuli (Fig. 3m, middle,

$P < 1 \times 10^{-4}$, and Extended Data Fig. 10i–p), but not to expected stimuli (Fig. 3m, $P = 0.074$ and $P = 0.088$ for visual stimuli A and B, and expected C and D, respectively). Similar to the effect of VIP neuron silencing, V1 neurons with strong prediction-error responses were more likely to be strongly suppressed by pulvinar inactivation (Fig. 3n and Extended Data

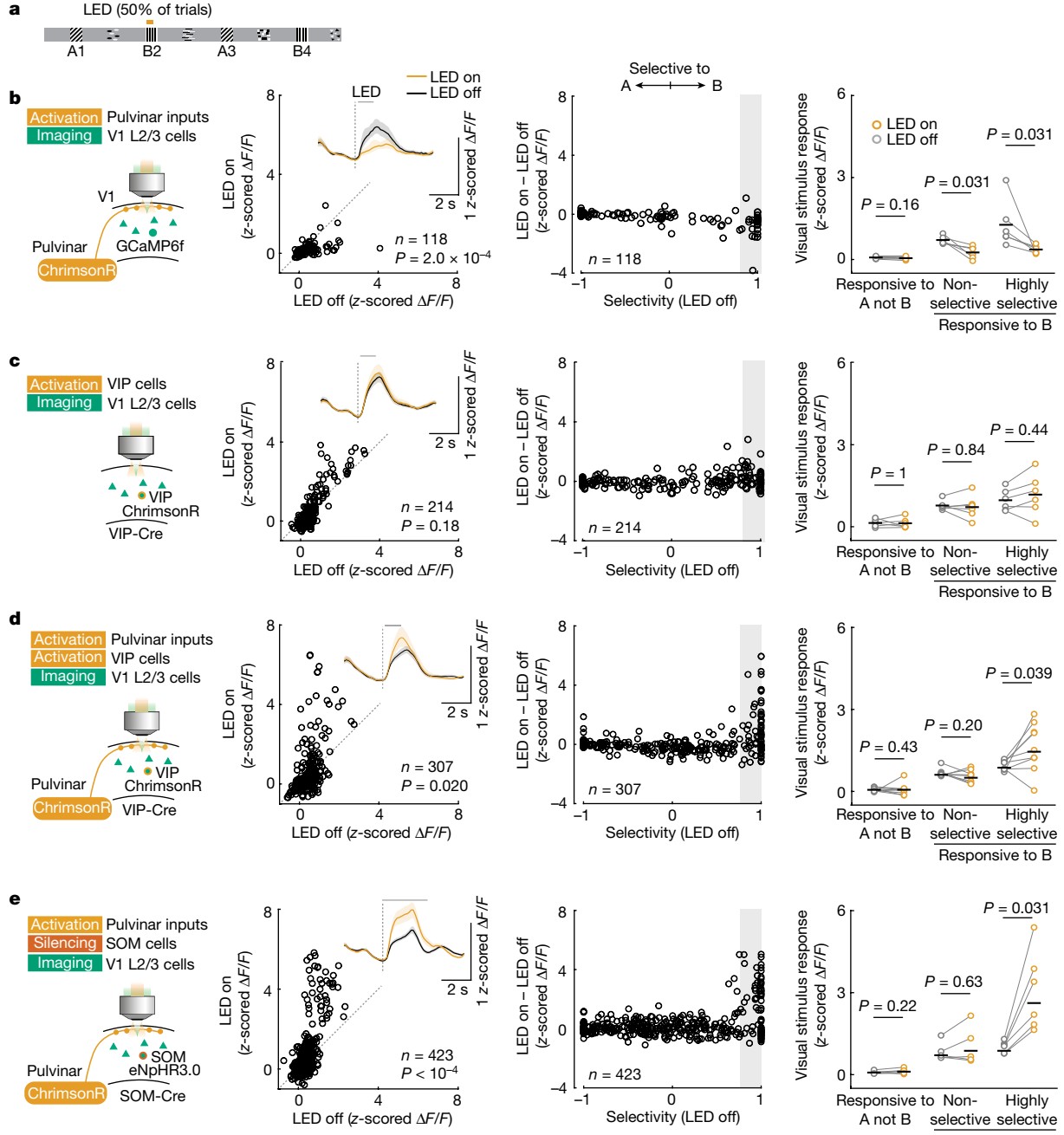

**Fig. 4 | Neocortical disinhibition and pulvinar input act synergistically.**
**a**, Experimental design. After training in the virtual corridor (stimuli A–B–A–B), optogenetic manipulation was paired with grating B2 in 50% of trials. **b**, Left, the activity of V1 layer 2/3 cells was recorded while pulvinar axons were optogenetically stimulated. Stimulation started 0.1 s after grating onset and lasted for 1 s. Second column, responses of individual V1 neurons with and without pulvinar axonal stimulation (LED on versus LED off). $n = 118$ grating A or B responsive cells from 6 mice, Hierarchical bootstrapping test. Inset, cell-averaged calcium responses with (amber) or without (black) optogenetic stimulation. Lines and shaded regions are mean and bootstrap 95% confidence intervals. Third column, effect of optogenetic stimulation (difference of response to grating B2 with and without LED stimulation) plotted against response selectivity (Methods) of individual V1 neurons. Right, calcium response

strength to stimulus B2 of neurons selective to A (left), and non-selective (selectivity B versus A < 0.6, middle) and highly selective (selectivity B versus A > 0.8, right) grating B2 responsive cells in V1 layer 2/3 with (amber) or without (grey) optogenetic stimulation. *P* values from two-sided signed-rank test. Data points depict mean responses for individual imaging sessions; $n = 6$ mice; black horizontal bars indicate mean across animals. **c**, Same as **b**, but the activity of V1 layer 2/3 cells was recorded while VIP cells were optogenetically stimulated. $n = 6$ mice. **d**, Same as **b**, but the activity of V1 layer 2/3 cells was recorded while pulvinar axons and VIP cells were optogenetically stimulated simultaneously. $n = 9$ mice. **e**, Same as **b**, but the activity of V1 layer 2/3 cells was recorded while pulvinar axons and SOM cells were optogenetically co-manipulated for 3 s starting at grating stimulus onset. $n = 6$ sessions from 4 mice. See also Extended Data Fig. 11.

Fig. 10k,m), independent of their visual response strength (Extended Data Fig. 10n). Moreover, pulvinar input was also required for V1 prediction-error responses to a familiar stimulus at an unexpected location (Extended Data Fig. 8j,k). Together, these cell-type-specific

inactivation experiments indicate that both intracortical VIP interneurons and pulvinar inputs contribute to prediction-error signals in V1. Next, we investigated how these two circuit elements interact to generate the amplified responses to unexpected stimuli.

## Cooperative thalamocortical circuit

Pulvinar axons make synaptic connections onto VIP neurons in the neocortex[30]. A plausible scenario for how the pulvinar and neocortical VIP neurons interact to mediate prediction-error signals may therefore involve pulvinar input activating VIP neurons in V1, which in turn boost pyramidal neuron responses to unexpected visual stimuli through the VIP–SOM disinhibitory circuit[24–26,28]. To directly test this hypothesis, we optogenetically stimulated either pulvinar axons or VIP interneurons in V1 while monitoring neural responses of V1 layer 2/3 neurons to the expected grating stimuli in the virtual corridor (Fig. 4a–c; see also Extended Data Fig. 11). Consistent with a previous report[37], stimulating pulvinar axons broadly suppressed responses to visual stimuli in V1 (Fig. 4b, $P = 2.0 \times 10^{-4}$). Moreover, stimulating pulvinar axons excited only a small subset of VIP neurons, and decreased VIP neuron prediction-error responses (Extended Data Fig. 12a–f). Optogenetically stimulating VIP interneurons had a minor effect on V1 activity, with a non-significant trend towards facilitating visual responses, unlike the strong amplification of stimulus-selective V1 neurons by prediction errors (Fig. 4c; $P = 0.18$). Remarkably, simultaneous co-activation of both pulvinar axons and VIP neurons strongly facilitated visual responses of a subset of V1 neurons (Fig. 4d, $P = 0.020$), indicating that pulvinar input and VIP neurons act synergistically, not additively. Moreover, response facilitation was specific to those visually driven neurons that responded highly selectively to the visual stimulus that was paired with optogenetic stimulation, mimicking the prediction-error signal in V1 (Fig. 4d, $P = 0.039$; Extended Data Fig. 11f–i; compared to Fig. 2d,e). Our experimental evidence therefore does not support a direct pathway from pulvinar inputs onto VIP neurons to facilitate V1 responses, but pulvinar and VIP neurons are likely to be recruited independently, and act synergistically to provide stimulus-selective amplification of responses to unexpected stimuli in V1.

Our results indicate that when VIP neurons are activated, they can counteract the inhibitory influence pulvinar activation has on the V1 network. The main synaptic targets of VIP neurons are SOM interneurons that inhibit the apical dendrites of pyramidal cells[24–26,28]. VIP neurons can therefore disinhibit pyramidal cells via the inhibition of SOM neurons. We hypothesized that pulvinar activation may recruit SOM neurons whose inhibitory influence on the V1 network may be alleviated when VIP neurons are simultaneously active. If this were the case, silencing SOM neurons while activating pulvinar should have effects similar to VIP neuron and pulvinar co-activation. Indeed, simultaneous optogenetic stimulation of pulvinar axons and inactivation of SOM neurons in V1 completely abolished the pulvinar-driven suppression of V1 activity (Fig. 4e; compared to Fig. 4b). Remarkably, this manipulation also strongly and specifically facilitated visual responses of V1 neurons responding highly selectively to the visual stimulus paired with the optogenetic manipulation, again mimicking the V1 prediction-error signal (Fig. 4e, $P = 0.031$), and suggesting that the pulvinar's excitatory drive onto V1 pyramidal neurons is accompanied by a strong feed-forward inhibitory drive via SOM neurons.

Although higher-order sensory thalamocortical pathways do not prominently target cortical SOM neurons[37–39], at least a subset of SOM neurons in V1 has been shown to receive input from the pulvinar[30,40,41]. We imaged responses of V1 layer 2/3 SOM neurons while optogenetically stimulating pulvinar axons in V1, and found that although most SOM neurons were either not affected or even suppressed, a subset of SOM neurons (16 ± 9%; mean ± s.d.) was strongly activated by pulvinar stimulation (Fig. 5a–c and Extended Data Fig. 12g,h). Notably, SOM neurons that were recruited by pulvinar stimulation were suppressed by unexpected visual input, suggesting that this subset of SOM neurons is inhibited by VIP neurons[28] (Fig. 5d,e). By contrast, layer 2/3 SOM neurons that are not recruited by pulvinar stimulation were activated by unexpected visual stimuli, similar to VIP neurons, suggesting that

they do not receive strong inhibition from VIP neurons and/or are more strongly driven by the local excitatory layer 2/3 network (Fig. 5d,e), consistent with previous studies[28,42,43]. Together, these results show that excitatory drive from the pulvinar onto V1 pyramidal neurons is paralleled by a powerful inhibitory pathway via a specific subpopulation of SOM neurons. When VIP neurons are active simultaneously with pulvinar input they inhibit SOM neurons, thus reducing feed-forward inhibition from pulvinar to V1, and enabling pulvinar drive to strongly activate a subset of layer 2/3 pyramidal cells (Fig. 5f). These results therefore reveal a circuit driving V1 prediction-error signals through synergistic interactions of pulvinar inputs and VIP neurons.

## Discussion

Here we describe a mechanism for boosting sensory responses by prediction errors in V1 when animals' expectations of visual stimuli at specific locations of a virtual environment are violated. Prediction errors selectively amplify the representation of unexpected visual input, via synergistic interactions of higher-order thalamic input and local VIP–SOM disinhibitory circuits in V1.

Prediction-error responses are dependent on VIP neuron activity as well as input from the pulvinar, a higher-order visual nucleus in the thalamus that has previously been implicated in predictive processing, and conveys prediction-error signals to V1[31,32,44]. Co-activation of pulvinar axons and VIP neurons in V1 can reproduce the selective amplification of V1 neurons even in the absence of prediction errors. Notably, we found that pulvinar input to V1 is gated by VIP–SOM inhibitory interactions. The pulvinar suppresses the activity of V1 cells via a subpopulation of SOM neurons. To allow pulvinar input to amplify V1 responses, this inhibition has to be alleviated by activity in VIP neurons that inhibit SOM neuron responses (Fig. 5f). This mechanism may explain seemingly contradictory findings about how the pulvinar affects cortical activity[37,45] and establishes VIP neurons as a gate for higher-order thalamic input to V1. VIP neurons receive prominent neuromodulatory and top-down cortical input, and have been shown to be activated by salient events such as reward, punishment and novel stimuli[20,23,24,26,27,29,30,46–48]. They can therefore regulate the influence of pulvinar input on visual processing in V1, depending on the relevance of visual stimuli or the animal's behavioural state. As VIP–SOM disinhibitory circuits and higher-order thalamic feedback input are present throughout the cortical hierarchy[24–26,28,30,34,47], this cooperative circuit mechanism may serve as a common computational motif in neocortical networks.

Although VIP neurons and pulvinar inputs to V1 are broadly recruited by unexpected stimuli (Extended Data Fig. 7), prediction-error signals in V1 are observed only in subpopulations of neurons that are highly selective for the visual stimulus encountered. Our results point to a potential circuit mechanism for this selective response amplification in V1. We reproduced the selective amplification of only stimulus-selective V1 neurons by co-activating VIP neurons with pulvinar input to V1, but also when bypassing VIP activation by silencing SOM neurons while stimulating pulvinar input (Fig. 4d,e). Thus, selectivity of response amplification in V1 neurons does not depend on VIP neuron recruitment or the activity of SOM neurons, but rather on pulvinar input more effectively driving V1 neurons with sharp tuning. This suggests a selective influence of pulvinar on subpopulations of stimulus-selective V1 neurons, balanced by inhibition from pulvinar-driven SOM neurons (Extended Data Fig. 11j–m). This pulvinar-dependent response enhancement may be further amplified via recurrent excitation within subnetworks of selective V1 neurons tuned to the same stimulus[49] and lateral suppression of the rest of the network via parvalbumin-expressing neurons[50–52], collectively leading to selective amplification of unexpected input.

Which inputs drive pulvinar and VIP neurons, and what information do they convey? Visual prediction errors are derived through

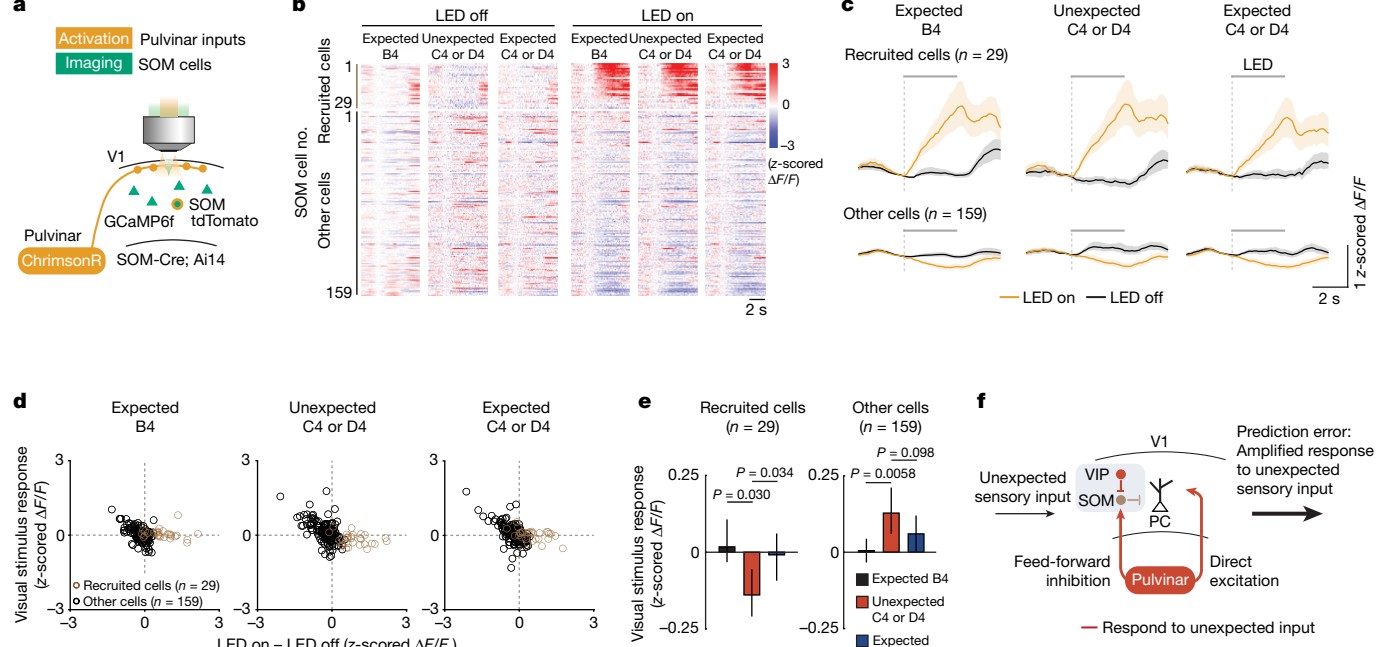

**Fig. 5 | Pulvinar activates a specific subpopulation of SOM cells.**
**a**, Experimental design. The activity of SOM cells was recorded while pulvinar axons were optogenetically stimulated for 3 s starting at visual stimulus onset. **b**, Single-cell responses to expected and unexpected visual stimuli of all SOM cells (individual rows, *n* = 6 sessions from 4 mice) with (right) or without (left) optogenetic stimulation. **c**, Cell-averaged calcium responses with (amber) or without (black) optogenetic stimulation of SOM cells significantly activated by pulvinar stimulation (recruited cells, *n* = 29) and other cells (*n* = 159). Lines represent the mean and shaded regions indicate 95% confidence intervals. **d**, Visual stimulus responses of individual SOM neurons to expected B4 stimulus (left), unexpected C4 or D4 stimulus (middle; in block 1) and expected C4 or D4 stimulus (right; in late block 2) plotted against the effect of pulvinar stimulation

(difference in strength of visual stimulus responses with and without optogenetic pulvinar axon stimulation) for recruited (brown) and other (black) SOM cells. **e**, Cell-averaged strength of calcium response to expected B4 (black), unexpected C4 or D4 (red) and expected C4 or D4 (blue) stimuli of recruited and other SOM cells. *P* values from hierarchical bootstrapping test with Bonferroni correction. Data are mean ± 95% bootstrap confidence intervals. **f**, Proposed circuit mechanism for sensory prediction errors. VIP neurons inhibit a specific subpopulation of SOM cells that otherwise gate pulvinar input to V1, resulting in specific pulvinar-driven response amplification of the most stimulus-selective neurons in V1. See also Extended Data Fig. 12.

a comparison of the actual visual input with internal predictions of expected visual input. Several top-down pathways have been proposed to convey different types of stimulus predictions to V1, including higher visual areas and anterior cingulate cortex[6,14,53]. In our paradigm, prediction errors may arise from violations of spatial predictions of the expected visual scene at a given location. Such spatio-visual predictions necessitate neural representations of space and spatial memory, and are thus likely to originate from hippocampus or related areas such as the retrosplenial cortex[54,55]. Previous studies have proposed that visual prediction errors may be computed in V1[6,14,53]. We observed sensory prediction-error signals not only in V1, but also in the pulvinar, and V1 prediction errors were dependent on pulvinar input. Prediction-error signals may therefore be computed outside of these visual areas—for instance, within the hippocampal formation—and conveyed to V1 by top-down projections via pulvinar and local VIP interneurons. Alternatively, errors could be computed in the pulvinar or in V1 from the comparison of visual input with spatio-visual predictions[5–10,14], and could then be amplified through pulvinar–V1 recurrent connections. The generation of other types of visual prediction errors observed in V1, such as those signalling deviations from visuo-motor predictions given the animal's own actions[15,31], probably involves different, motor-related pathways, including superior colliculus, anterior cingulate cortex or secondary motor cortex[10,53,56,57]. In general, prediction-error signals in V1 may be further enhanced by neuromodulators such as acetylcholine or noradrenaline that may signal stimulus saliency and novelty, or surprise more generally[27,48,58,59], and these signals are likely to influence the activity of VIP neurons[27,48,60].

Our results indicate that individual V1 neurons do not signal how the actual visual input deviates from the animal's predictions, as postulated within the predictive coding framework[5–8]. Instead, we propose an alternative view of predictive processing in sensory circuits: prediction errors amplify the representation of feed-forward sensory input in neocortex, while the extent of amplification may depend on how much the visual stimulus deviates from expectations and therefore the magnitude of animals' surprise. This would explain the particularly strong responses to novel stimuli that were not encountered before, as these are the least expected[20,23]. The amplified responses to unexpected stimuli may serve as a neural substrate for attentional shifts towards surprising events in the environment. However, the content of how actual input deviates from predictions may still be encoded in other brain areas or higher-dimensional population activity in V1.

In summary, sensory prediction errors in V1 increase the saliency of unexpected, and thus probably relevant, visual information. This enables downstream brain areas to prioritize these signals and potentially utilize them for updating internal predictions.

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

# Methods

## Mice

All experiments were performed under the UK Animals (Scientific Procedures) Act of 1986 (PPL PD867676F) following UK Home Office approval and local ethical approval by the Sainsbury Wellcome Centre Animal Welfare Ethical Review Body. A total of 105 mice, including 27 C57BL/6J mice, 24 VIP-Cre mice (JAX 010908, Jackson Laboratory; Cre expressed in VIP interneurons), 43 VIP-Cre × Ai14 mice (JAX 010908 and JAX 007914, Jackson Laboratory; tdTomato expressed in VIP interneurons), 7 SOM-Cre mice (JAX 013044, Jackson Laboratory; Cre expressed in SOM interneurons) and 4 SOM-Cre × Ai14 mice (JAX 013044 and JAX 007914, Jackson Laboratory; tdTomato expressed in SOM interneurons) were used in this study. Both female and male mice, at least 7 weeks old at the start of the experiments, were used. Mice were co-housed with littermates in IVC cages, in reversed day–night cycle lighting conditions, with the ambient temperature and humidity set to 23 °C and 56% relative humidity, respectively. Standard environment enrichment was provided in the form of a running wheel, a clear tube and wooden toys.

## Surgical procedures

Prior to surgery, Dexadreson (2–3 mg kg$^{-1}$) and Carprofen (5 mg kg$^{-1}$) were administered. General anaesthesia was induced with 2.5–3% isoflurane, which was then reduced to maintain a breathing rate of around 1 Hz. A 3- or 4-mm craniotomy was made over the right V1, centred on 2.45 mm lateral and 3.6 mm posterior of bregma. For two-photon calcium imaging and optogenetic manipulations of V1 cells, we injected adeno-associated virus (AAV) vectors into right monocular V1 (centred on 2.45 mm lateral and 3.7 mm posterior of bregma, 1–3 injections per mouse, 100–150 nl per injection). For two-photon calcium imaging and optogenetic manipulations of pulvinar axons, we injected AAV vector into the right pulvinar (calcium imaging and optogenetic activation: 1.6 mm lateral and 2.1 mm posterior of bregma, 2.35 below the cortical surface, 1 injection per mouse, 60 nl per injection; optogenetic inactivation: 1.55 mm lateral and 2.0 mm posterior of bregma, 2.3 mm below the cortical surface, 1.60 mm lateral and 2.2 mm posterior of bregma, 2.4 mm below the cortical surface, 2 injections per mouse, 60 nl per injection). All injections were performed using glass pipettes and Nanoject III microinjector (Drummond Scientific) or a pressure injection system (Picospritzer III, Parker). A 3- or 4-mm circular cover glass was glued in place using cyanoacrylate glue (Pattex). A custom-designed stainless steel head plate was attached to the skull using dental cement (Super-Bond C&B, Sun Medical). Animals were given analgesics (Carprofen; 5 mg kg$^{-1}$) at 24 and 48 h after surgery. Imaging started approximately 3 weeks after the virus injection.

## Viral constructs

We used AAV1-hSyn-GCaMP6f ($2 × 10^{13}$ vg ml$^{-1}$ Penn Vector Core/Addgene; diluted 1:8 to 1:15 in saline) for experiments involving two-photon calcium imaging of V1 layer 2/3 cells; AAV1-hSyn-GCaMP7b ($2 × 10^{13}$ vg ml$^{-1}$ Penn Vector Core/Addgene; diluted 1:2 in saline) or AAV1-hSyn-axon-GcaMP6s ($9 × 10^{12}$ vg ml$^{-1}$ Penn Vector Core/Addgene; diluted 1:2 in saline) for imaging of pulvinar axons; AAV2-EF1a-DIO-eNpHR3.0-mCherry ($4.0 × 10^{12}$ vg ml$^{-1}$, 1:2 to 1:10 dilution, UNC vector core) for optogenetic silencing of VIP cells or SOM cells; AAV2-hSyn-eNpHR3.0-mCherry ($3.3 × 10^{12}$ vg ml$^{-1}$, 1:2 to 1:4 dilution, UNC vector core) for optogenetic silencing of pulvinar axons; AAV1-hSyn-Flex-ChrimsonR-tdTomato ($3.9 × 10^{12}$ vg ml$^{-1}$, 1:2 to 1:5 dilution, UNC vector core) for optogenetic activation of VIP cells; AAV1-Syn-ChrimsonR-tdTomato ($4.1 × 10^{12}$ vg ml$^{-1}$, 1:2 to 1:5 dilution, UNC vector core) for optogenetic activation of pulvinar inputs; AAV1-hEF1a-mCherry ($5.7 × 10^{12}$ vg ml$^{-1}$, 1:2 to 1:5 dilution, Zurich vector core) for control experiment for LED light stimulation.

## Behavioural setup

Behavioural setups consisted of a styrofoam running wheel, two visual stimulation display monitors (see below), a reward delivery spout, and a camera for recording the pupil. Mice were head-fixed and placed on a styrofoam wheel (20 cm diameter, 12 cm width). Their running speed was monitored using a rotary encoder (Kubler Encoder 1000 ppr) coupled to the wheel axle. Reward (a drop of strawberry milk, 50% Ensure nutrition shake, Abbott Laboratories) was delivered by a lick spout in front of the mouse and was regulated via a solenoid pinch valve (161P011, NResearch). Licks were detected with a piezoelectric diaphragm sensor (7BB-12-9, Murata) placed under the spout. Images of the left eye were recorded with a CMOS camera (22BUC03, Imaging Source) at 30 Hz in order to track eye movements and pupil size. The recording of the encoder, presentation of visual stimuli, opening of the reward valves, and camera recordings were controlled by custom-written software in LabView. Behavioural data were acquired using a PCIe 6321 acquisition card (National Instruments).

## Food restriction and pre-training

Before mice underwent training in the virtual environment, they were food-restricted and pre-trained to encourage continuous running on the styrofoam wheel. Four to seven days after surgery, food restriction and pre-training started. Mice were weighed daily and given typically 2–3 g of food pellet in addition to strawberry milk given in training sessions to ensure they maintained around 90%, but at least 85%, of their starting body weight. For the first few days, animals were handled in a soft cloth and iteratively fed strawberry milk (Abbott Laboratories) through a syringe until they got used to short manual restraint of the head plate. Mice were then head-fixed and put on the freely rotating styrofoam wheel for 15–60 min. Mice were encouraged to run on the wheel by delivering strawberry milk rewards after they moved a short distance (initially set to ~10 cm). This distance was adjusted (up to 500 cm) depending on the running speed of the mouse, such that mice received roughly one reward every 30 s. Additional rewards were occasionally delivered by the experimenter. This pre-training took 4–10 days.

## Virtual corridor

Once mice were running continuously, they were moved to a virtual environment consisting of a linear corridor with varying wall patterns as described previously[14]. The cylinder's rotation (the instantaneous running speed of the animal) was used to control the speed at which the animal moved through the virtual environment. The virtual environment was displayed on two monitors (U2715H, Dell; 60 Hz refresh rate), placed 21 cm away from both eyes of mice and oriented at 35° relative to the midline. Each monitor covered a visual field of approximately 110° horizontally and 60° vertically. All elements of the corridor including the gratings were calibrated to be isoluminant (10.1 cd m$^{-2}$). The luminance of the monitor was set at 0.1 cd m$^{-2}$, 10.1 cd m$^{-2}$ and 20.1 cd m$^{-2}$, at black, grey and white values, respectively. The luminance of visual stimuli was measured using a luminance meter (Konica Minolta, LS-100). The grey walls of the virtual corridor were lined with four different landmarks. The last landmark represented the reward zone located at the end of the corridor. Reaching the reward zone triggered an automatic reward delivered by a spout located in front of the mouse. After the reward delivery, the virtual environment was reset to the beginning of the corridor to start the next trial.

Grating stimuli were suddenly presented on full screen once the mouse entered a certain position in the corridor. This was done to ensure precise control of when the mouse would first see the grating. Grating stimuli were presented at four different positions between landmarks. The optic flow of the grating stimuli was 'uncoupled' from the running speed for 2.4 s, such that the animal's locomotion did not affect its temporal frequency. Gratings were square-wave gratings, with

the spatial frequency of approximately 0.04 cycles per degree (cpd) at the centre of the monitor and the temporal frequency approximately 2 cycles per second (Hz). Duration of the grating presentation was approximately 2 s at the centre of the monitors. The precise timing of visual stimulus onsets was recorded with a photodiode (Thorlabs) attached to the monitor.

During 5 training sessions, the virtual corridor and the sequence of the four grating stimuli was identical (A–B–A–B) on every trial. In the subsequent imaging session, the identity of one of the four grating stimuli was changed. In block 1 of this session (160 trials), the identity of the 4th grating stimulus B changed either to a novel grating stimulus C (C session), a novel stimulus D (D session), familiar stimulus A (A session) or no stimulus was shown (omission session) on randomly chosen 10% of trials. In block 2, the novel stimulus or no stimulus was shown at the fourth position in 100% of trials. Occasionally one mouse underwent several sessions with unexpected stimuli. In that case, mice went through another training session (with gratings A–B–A–B) in between. For imaging of pulvinar axons, block 1 was shortened to 60 trials, and either a novel grating stimulus C (C session) or a novel stimulus D (D session) was shown at the fourth position on randomly chosen 15% of trials. In block 2, the novel stimulus was shown in 100% of trials, as for imaging of V1 layer 2/3 cells. For experiments in Extended Data Figs. 3a–e and 5l–p, a horizontal grating stimulus E was shown at position 1 and 3 instead of grating stimulus A (E–B–E–B). A novel stimulus C (C session) or a novel stimulus A (A session) was shown on randomly chosen 10% of trials. For experiments in Extended Data Figs. 8 and 9a–c, we used a short version of the virtual corridor with two grating stimuli (A–B) and the identity of the 2nd grating stimulus B changed to familiar stimulus A on randomly chosen 10% of trials.

## Visual stimulation

For experiments in Extended Data Fig. 9d–f, visual stimuli were generated using the open-source Psychophysics Toolbox[61] based on MATLAB (MathWorks) and were presented full-field on one monitor at approximately 21 cm from the left eye of the mouse, covering 110° of visual space. Square-wave gratings (spatial frequency: 0.04 cpd, temporal frequency: 2 Hz, duration: 2 s, interval: 4 s, directions: 0 to 360° in 45° increments) were randomized in order and presented 10 times per direction.

## Two-photon calcium imaging

Two-photon calcium imaging was performed using a commercial resonance scanning two-photon microscope (B-Scope; Thorlabs) with a 16× water-immersion objective (NA 0.8, Nikon), with a Ti::Sapphire laser at 930 nm excitation wavelength (Mai Tai, SpectraPhysics). Emission light was band-pass filtered using a 525/50 filter for GCaMP and a 607/70 filter for tdTomato/mCherry (Semrock). Images of 512 × 512 pixels from four imaging planes with fields of view ranging from 380 × 380 µm to 440 × 440 µm were acquired at 7.5 Hz frame rate for imaging of V1 neurons and of a single plane of 160 × 160 µm at 15 Hz frame rate for imaging of pulvinar axonal boutons using ScanImage[62]. For imaging of V1 neurons, we used a piezo-actuator (Physik Instrumente) to move the objective in steps of 15 µm between frames to acquire images at four different depths, thus reducing the effective frame rate to 7.5 Hz. Imaging of V1 neurons was performed in layer 2/3 (typically 150–200 µm below the cortical surface). The laser power under the objective never exceeded 35 mW. Axonal bouton calcium measurements were performed in cortical layer 1 (35–55 µm below the cortical surface), with laser powers below 20 mW.

To avoid cross-talk between imaging and visual stimulation, the monitor backlight was controlled using a custom-built circuit to present visual stimuli only at the resonant scanner turnaround points in between two subsequent imaging lines (when data were not acquired)[63]. The frame trigger signal during two-photon calcium imaging was recorded by Labview and used for synchronization between the calcium imaging frames and task related data (for example, behaviour data and visual stimuli onsets).

For imaging of pulvinar axons, we used VIP-Cre × Ai14 mice. We simultaneously imaged pulvinar axons expressing GCaMP and neurites of VIP neurons expressing tdTomato in layer 1. We then used the red signal (tdTomato) as a structural marker to perform Z-drift correction during imaging and frame registration during data pre-processing.

## Optogenetic manipulation

Simultaneous two-photon imaging and optogenetic stimulations were performed as previously described[15]. Briefly, 595 nm light was delivered through the objective lens using a fast LED (UHP-T-595, Prizmatix). The LED light power was set to 8 mW in front of the objective. To combine two-photon imaging and optogenetic manipulation, the LED for optogenetic manipulation was synchronized to the resonant scanner turnaround points (when data were not acquired). The propagation of reflected light to the eyes of the mouse was blocked by a metal light shield cone placed on the head plate and a black cement wall around the implant. Optogenetic manipulation occurred in randomly chosen 10–50% of each trial type. For most of optogenetic manipulations, LED stimulation was applied continuously for 3 s, starting at visual stimulus onset. For optogenetic silencing during passive visual stimulation (Extended Data Fig. 9d–f), LED stimulation was applied throughout visual stimulus presentation (2 s). For optogenetic activation (Fig. 4b–d and Extended Data Fig. 11a–c,f–i), LED stimulation was applied at a frequency of 20 Hz, with 40% duty cycle (20 ms pulses) for 1 s starting 0.1 s after visual stimulus onset.

## Histology

At the end of each experiment, targeting of virus injections was confirmed by histology. Brains were extracted and fixed overnight in 4% paraformaldehyde, and stored in a 50 mM phosphate buffer. Brains were embedded in 5% agarose and imaged using serial section[64] two-photon[65] microscopy. Our microscope was controlled by Scan-Image Basic (MBF Bioscience) using BakingTray, a custom software wrapper for setting up the imaging parameters[66]. Images were assembled using StitchIt[67]. Coronal slices were cut at a thickness of 40 µm using a vibratome (Leica VT1000), and imaged every 20 µm with a 16× water-immersion objective (NA 0.8, Nikon). Whole brain coronal image stacks were acquired at a resolution of 4.4 × 4.4 × 20 µm in $xyz$, with a two-photon laser wavelength of 780 nm, and approximately 130 mW at the sample. Selected brain images were registered to the adult mouse Allen common coordinate framework[68] using The Slice Histology Alignment, Registration, and Probe-Track analysis (SHARP-Track), a MATLAB based registration pipeline with optimized parameters for mouse brain registration at various cutting angles[69]. A subset of brains was embedded in 4% agarose (A9539, Sigma), cut in 200 µm coronal slices on a vibratome (HM650V; Microm), mounted in a mounting medium containing DAPI (Vectashield; Vector Laboratories) and imaged on a slide scanner (Zeiss AxioScan) or on a confocal microscope (Leica SP8).

## Quantification and statistical analysis

**Two-photon imaging.** Two-photon imaging frames were motion corrected and segmented using custom-written scripts in MATLAB as previously described[31]. In brief, to correct for $x$–$y$ motion, two-photon imaging frames were registered to a 1,200-frame average (40 frames × 30 batches) using a phase-correlation algorithm. When the same V1 neurons were imaged over multiple sessions, images from those sessions were registered together, and identical cells were matched across sessions by using custom-written software. Frames with large motion were detected by inspecting the registration displacement results and were discarded from further analysis. Regions of interest (ROIs) were detected semi-automatically using intensity thresholding combined with principal component analysis–independent component analysis refinement and validated and refined manually. All time series were

extracted and analysed with custom-written functions using the Time-SeriesAnalysis package[70]. All pixels within each ROI were averaged to give a single time course. Contaminating signals from neuropil were subtracted using an asymmetric Student's $t$ model (ast_model; https://github.com/BaselLaserMouse/ast_model). Calcium $\Delta F/F_0$ signals were obtained by using the baseline fluorescence $F_0$, which is estimated by a Gaussian mixture model with two components fitted on the raw fluorescence data. The mean parameter of the lowest Gaussian component is used as $F_0$. To be able to compare calcium activity across sessions and mice, $z$-scored $\Delta F/F$ was computed by subtracting the mean value of $\Delta F/F$ of a session and dividing the resulting trace by the standard deviation.

**Analysis of visual responses.** The response to each grating was calculated using the mean $z$-scored $\Delta F/F$ calcium signal averaged over a window from 0.4 s to 2 s after grating onset, baseline-subtracted using the mean $z$-scored $\Delta F/F$ signal during 0.5 s before stimulus onset for each grating presentation. Neurons were classified as stimulus-responsive if their mean response was bigger than 0.5 $z$-scored $\Delta F/F$. In Fig. 1, cell-averaged calcium traces are from neurons responsive to the presented grating in trials with unexpected C or D (block 1), trials with expected C or D trials (late block 2) or both. For comparison, Extended Data Fig. 2 shows cell-averaged calcium responses of all neurons responsive to any grating. In Fig. 2a,b and Extended Data Figs. 5u–z and 6a,b, cells were defined as prediction-error-responsive if the responses were significantly different between unexpected stimuli C4, D4 or stimulus omission (block 1) and expected stimuli C4, D4 or stimulus omission (second half of block 2, two-sided $t$-test; $\alpha = 0.05$; unexpected C4, D4 or omission versus expected C4, D4 or omission) and the difference in response was larger than 0.5 $z$-scored $\Delta F/F$. Similarly, in Extended Data Figs. 5e,p and 7e, cells were defined as prediction-error-responsive if the responses were significantly different between unexpected C4 or D4 (block 1) and expected C4 or D4 (second half of block 2, two-sided $t$-test; $\alpha = 0.05$; unexpected C4 or D4 versus expected C4 or D4) and the difference in response was larger than 0.3 $z$-scored $\Delta F/F$. In Fig. 3f,g,m,n, average response in LED on and off trials was used for classification of stimulus-responsive cells to avoid selection bias towards LED off trials. In Fig. 4, response in LED off trials was used for classification of stimulus-selective cells to avoid inclusion of opsin-expressing, therefore directly activated VIP cells. In Fig. 5 and Extended Data Fig. 12, SOM cells or VIP cells were defined as 'recruited' if their responses were significantly different between with and without optogenetic pulvinar axon stimulation (two-sided $t$-test; $\alpha = 0.016$; with versus without LED light stimulation) and the difference in response was larger than 0.3 $z$-scored $\Delta F/F$, during at least one of the visual stimulus presentations (expected B4, unexpected C4 or D4, expected C4 or D4). In Extended Data Fig. 5f–k, cells were defined as prediction-error (C or D) responsive but not responsive to expected C or D if the responses were significantly different between unexpected C4 or D4 and expected C4 or D4 (two-sided $t$-test; $\alpha = 0.05$) and the difference in response was larger than 0.5 $z$-scored $\Delta F/F$, but the response to expected C4 or D4 was smaller than 0.5 $z$-scored $\Delta F/F$. In Extended Data Fig. 6g,h, in which raw $\Delta F/F$ rather than $z$-scored $\Delta F/F$ was used, neurons were defined as stimulus-responsive if their stimulus response strength was larger than 0.2 $\Delta F/F$ in the second half of block 2. In Extended Data Fig. 7j,k, boutons were defined as stimulus-responsive if the response to any expected stimulus in late block 2 was larger than 0.1 $z$-scored $\Delta F/F$.

**Selectivity and selectivity index.** To quantify the selectivity of neural responses we computed a response selectivity measure for individual V1 layer 2/3 cells and pulvinar boutons:

$$\text{Selectivity} = (R_{\text{C4 or D4}} - R_{\text{A3 or B2}})/(R_{\text{C4 or D4}} + R_{\text{A3 or B2}})$$

Where $R_{\text{C4 or D4}}$ is the mean response to the gratings C4 or D4 in late block 2, and $R_{\text{A3 or B2}}$ is the mean response to the gratings A3 or B2 in late block

2. Selectivity values of >1 or <−1 were shown as 1 or −1, respectively. If selectivity of neurons responsive to a specific visual stimulus was less than 0.6 or more than 0.8, they were classified as either non-selective or highly selective to that stimulus, respectively. We also used an additional selectivity index (SI) to quantify response selectivity of individual pulvinar boutons (Extended Data Fig. 7l–n), since this index provided a more reliable measure for the noisy bouton calcium traces. SI was calculated as previously described[71]. In brief, it was computed from the difference between the mean response to the expected stimulus C4 or D4 and expected stimulus B2 in late block 2, divided by the pooled standard deviation of the responses.

**Fast and slow running trials.** For the analysis in Extended Data Fig. 1k,l, trials in block 1 and 2 were divided into fast and slow running trials based on mean running speed during presentation of grating C4. A time window starting 0.4 s and ending 2 s after the grating onset, similar to the response window used for calcium responses, was used to calculate the mean running speed. A trial was defined as 'fast' or 'slow' if the mean running speed during the time window was in the top 50th or bottom 50th percentile of all visual stimulus C4 presentations in block 1 or 2.

**Correlation of running speed and neuronal activity.** To determine the effect of running speed on neuronal activity (Extended Data Fig. 1m,n), we computed for each cell the correlation between mean $\Delta F/F$ and mean running speed in a time window (starting 0.4 s and ending 2 s after grating stimulus onset) of each trial in block 1 or 2. For the analysis in Extended Data Fig. 1o, we used the square of the correlation coefficient ($R^2$, coefficient of determination) of running speed and $\Delta F/F$ across the recording, to quantify the strength of the modulation of neural responses by running speed across the entire session (block 1 and 2).

**Pupil size.** Pupil size was computed offline. The pupil was detected using a binary threshold and centre of mass of the detected regions. We then applied a one-dimensional filter to the traces using the filloutlier function in MATLAB.

**Statistics.** We used two-sided Wilcoxon signed-rank tests for comparisons across animals and hierarchical bootstrapping test for comparisons across cells unless otherwise stated. Hierarchical bootstrap procedure was performed as previously described[72,73]. In short, we first randomly resampled animals with replacement and then resampled cells with replacement from each of the resampled animals. We then randomly shuffled the paired data and calculated the statistic of interest. This process was repeated 10,000 times. The statistic values were compared against the value of the original data to calculate $P$ values. Where relevant $P$ values were adjusted for multiple comparisons using Bonferroni correction, as indicated in the figure legends. For the randomization test, we computed the statistic of interest with randomly shuffled data (10,000 times). The statistic values were compared against the value of the original data to calculate $P$ values. All tests were performed using MATLAB. Mean and bootstrap 95% confidence intervals were used for display purposes, as stated in the figure legends. Confidence intervals were estimated using bootci function in MATLAB, with 10,000 bootstrap samples with replacement. Cohen's $d$ was computed from the difference between the two mean responses, divided by the pooled standard deviation of the responses. No statistical methods were used to predetermine sample sizes, but our sample sizes are similar to those generally used in the field. Experimenters were not blinded to experimental groups. Animals were allocated to experimental groups pseudo-randomly, and trial types (expected or unexpected stimuli, with or without optogenetic manipulation) were randomly interleaved.

**Reporting summary**

Further information on research design is available in the Nature Portfolio Reporting Summary linked to this article.

## Data availability

The data that support the main findings of this study are publicly available at https://doi.org/10.5281/zenodo.11403111 (ref. 74). Other data that are generated in this study are available from the corresponding author upon reasonable request. Source data are provided with this paper.

## Code availability

The analysis code is publicly available at https://doi.org/10.5281/zenodo.11403111 (ref. 74).

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

**Acknowledgements** The authors thank M. Li for help with animal husbandry and pre-training; M. Rio for help with the calcium imaging data pre-processing pipeline and virtual reality; R. A. A. Campbell and SWC Advanced Microscopy Facility for help with microscopy; N. Vissers for help with histology; A. Fiser and G. Keller for the initial design of the virtual corridor; T. Kanamori, I. Voitov and M. Javadzadeh for feedback on the manuscript; and T.D.M.-F. laboratory members and S.B.H. laboratory members for discussions. This work was supported by Osamu Hayaishi Memorial Scholarship (to S.F.); a Sainsbury Wellcome Centre Core Grant from the Gatsby Charitable Foundation and Wellcome (219627/Z/19/Z and 090843/F/09/Z); a Wellcome Investigator Award (219561/Z/19/Z to S.B.H.); the Gatsby Charitable Foundation (GAT3212 and GAT3361 to T.D.M.-F.); the Wellcome Trust (090843/E/09/Z and 217211/Z/19/Z to T.D.M.-F.); European Research Council (HigherVision 337797 to S.B.H.; NeuroV1sion 616509 to T.D.M.-F.); the SNSF (31003A 169525 to S.B.H.); and Biozentrum core funds (University of Basel).

**Author contributions** S.F., S.B.H. and T.D.M.-F. conceived the study. S.F. performed the experiments and analysed the data. A.D.F. assisted with surgical procedures, animal pre-training and preliminary optogenetic experiments. A.M.A. assisted with animal pre-training and histology. S.F., S.B.H. and T.D.M.-F. wrote the manuscript.

**Competing interests** The authors declare no competing interests.

**Additional information**
**Correspondence and requests for materials** should be addressed to Shohei Furutachi, Thomas D. Mrsic-Flogel or Sonja B. Hofer.

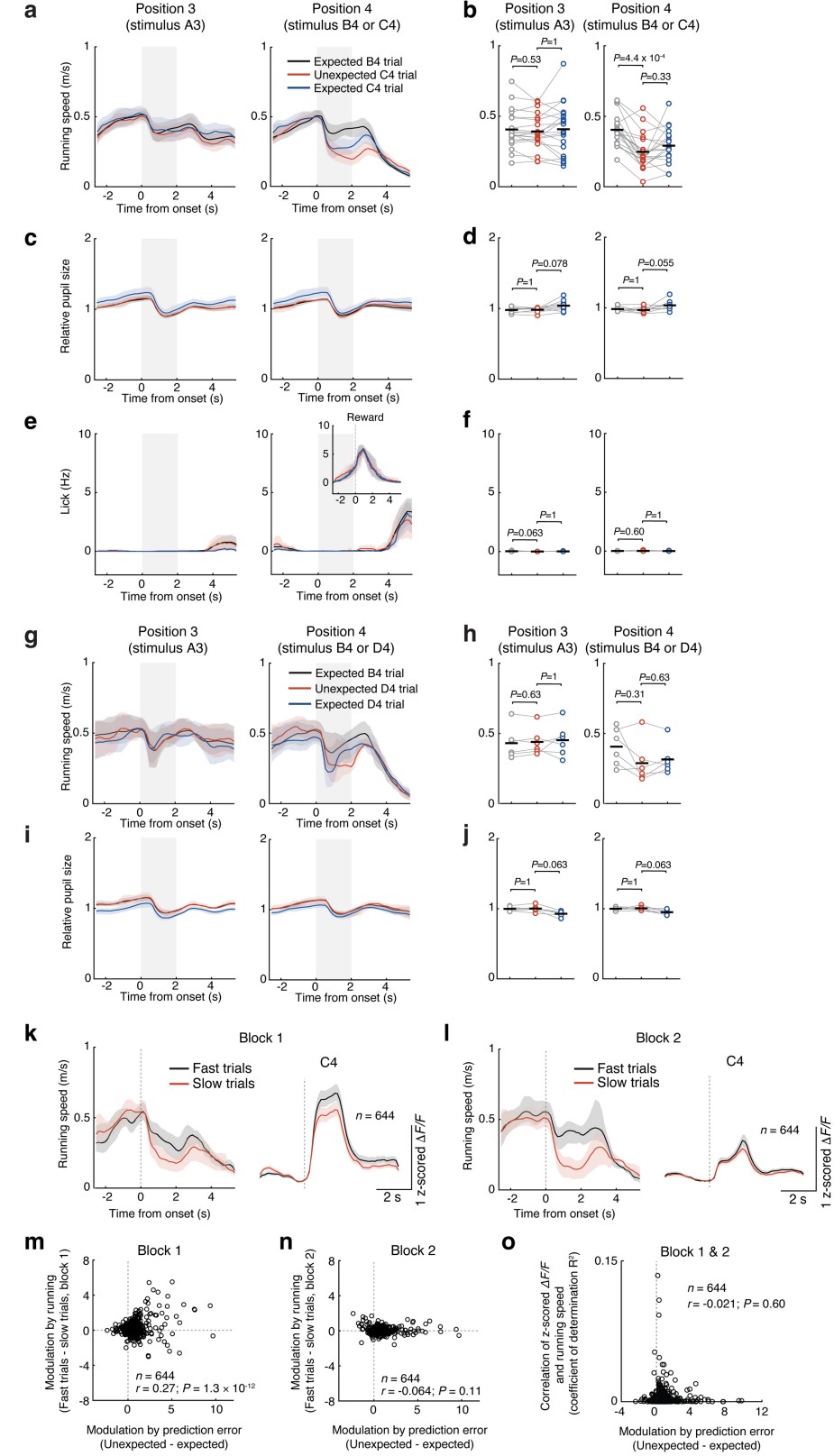

**Extended Data Fig. 1** | See next page for caption.

**Extended Data Fig. 1 | Running and licking behaviour and pupil size during presentation of expected and unexpected gratings.** Related to Figs. 1 and 2. **a**, Running speed at virtual corridor position 3 (left, grating A3 shown) or at position 4 (right, grating B4 or C4 shown) in trials in which grating B was presented at position 4 (black, expected B4 trials, 90% of trials in block 1), trials in which grating C was presented at position 4 (red, 10% of trials in block 1, unexpected C4 trials) and trials in the second half of block 2 (blue, expected C4 trials, late block 2). Light grey shading indicates length of visual stimulus at the centre of monitors. Lines and shading are mean and bootstrap 95% CI ($n = 20$ mice). **b**, Same as a, but data from individual animals are shown separately. Black bars represent mean across animals. Position 3, running speed during grating A3 presentation in B4 vs unexpected C4 trials: $P = 0.53$; running speed during grating A3 presentation in unexpected vs expected C4 trials: $P = 1$. Position 4, running speed during B4 vs unexpected C4 presentation: $P = 4.4 \times 10^{-4}$; running speed during unexpected vs expected C4 presentation: $P = 0.33$. $n = 20$ mice, two-sided signed-rank test with Bonferroni correction. **c** and **d**, Same as a and b, but for relative pupil size (normalized by each session's median value). **d**, Position 3, pupil size during grating A3 presentation in B4 vs unexpected C4 trails: $P = 1$; pupil size during grating A3 presentation in unexpected vs expected C4 trails: $P = 0.078$. Position 4, pupil size during B4 vs unexpected C4 presentation: $P = 1$; pupil size during unexpected vs expected C4 presentation: $P = 0.055$; $n = 9$ mice, two-sided signed-rank test with Bonferroni correction. **e** and **f**, Same as a and b, but for lick rate. **e**, Inset shows lick rate around the reward delivery. **f**, Position 3, lick rate during A3 presentation in B4 vs unexpected C4 trials: $P = 0.063$; lick rate during A3 presentation in unexpected vs expected C4 trials: $P = 1$. Position 4, lick rate during B4 vs unexpected C4 presentation: $P = 0.60$; lick rate during unexpected vs expected C4 presentation: $P = 1$. $n = 9$ mice, two-sided signed-rank test with Bonferroni correction. **g-j**, Same as a-d, but for a different unexpected visual stimulus D. **h**, Position 3, running speed during grating A3 presentation in B4 vs unexpected D4 trials: $P = 0.63$; running speed during grating A3 presentation in unexpected vs expected D4 trials: $P = 1$. Position 4, running speed during B4 vs unexpected D4 presentation: $P = 0.31$; running speed during unexpected vs expected C4 presentation: $P = 0.63$. $n = 6$ mice, two-sided signed-rank test with Bonferroni correction. **j**, Position 3, pupil size during grating A3 presentation in B4 vs unexpected D4 trials: $P = 1$; pupil size during grating A3 presentation in unexpected vs expected D4 trials: $P = 0.063$. Position 4, pupil size during B4 vs unexpected D4 presentation: $P = 1$; pupil size during unexpected vs expected D4 presentation: $P = 0.063$; $n = 6$ mice, two-sided signed-rank test with Bonferroni correction. **k**, Running speed (left) and responses to grating C4 (right) on trials with fast (black, top 50%) and slow (red, bottom 50%) running speed during grating C presentation at position 4 in block 1 (see Methods). **l**, Same as k, but for block 2. **m**, Scatterplot showing the relationship between response modulation by running speed (difference in calcium response between fast and slow trials) and strength of prediction error responses (Pearson correlation: $r = 0.27$, $P = 1.3 \times 10^{-12}$; $n = 644$ cells from 9 mice) in block 1. The positive correlation shows that the response to unexpected grating C was larger in trials with higher running speed, as expected from previous studies. This shows that running speed changes (deceleration in response to the unexpected stimulus) cannot explain the increased neural responses to unexpected stimuli. **n**, Same as m, but for block 2 (Pearson correlation: $r = -0.064$, $P = 0.11$; $n = 644$ cells from 9 mice). **o**, Scatterplot showing the relationship between correlation of z-scored $\Delta F/F$ and running speed (coefficient of determination $R^2$, over the entire recording session) and strength of prediction error responses (Pearson correlation: $r = -0.021$, $P = 0.60$; $n = 644$ cells from 9 mice).

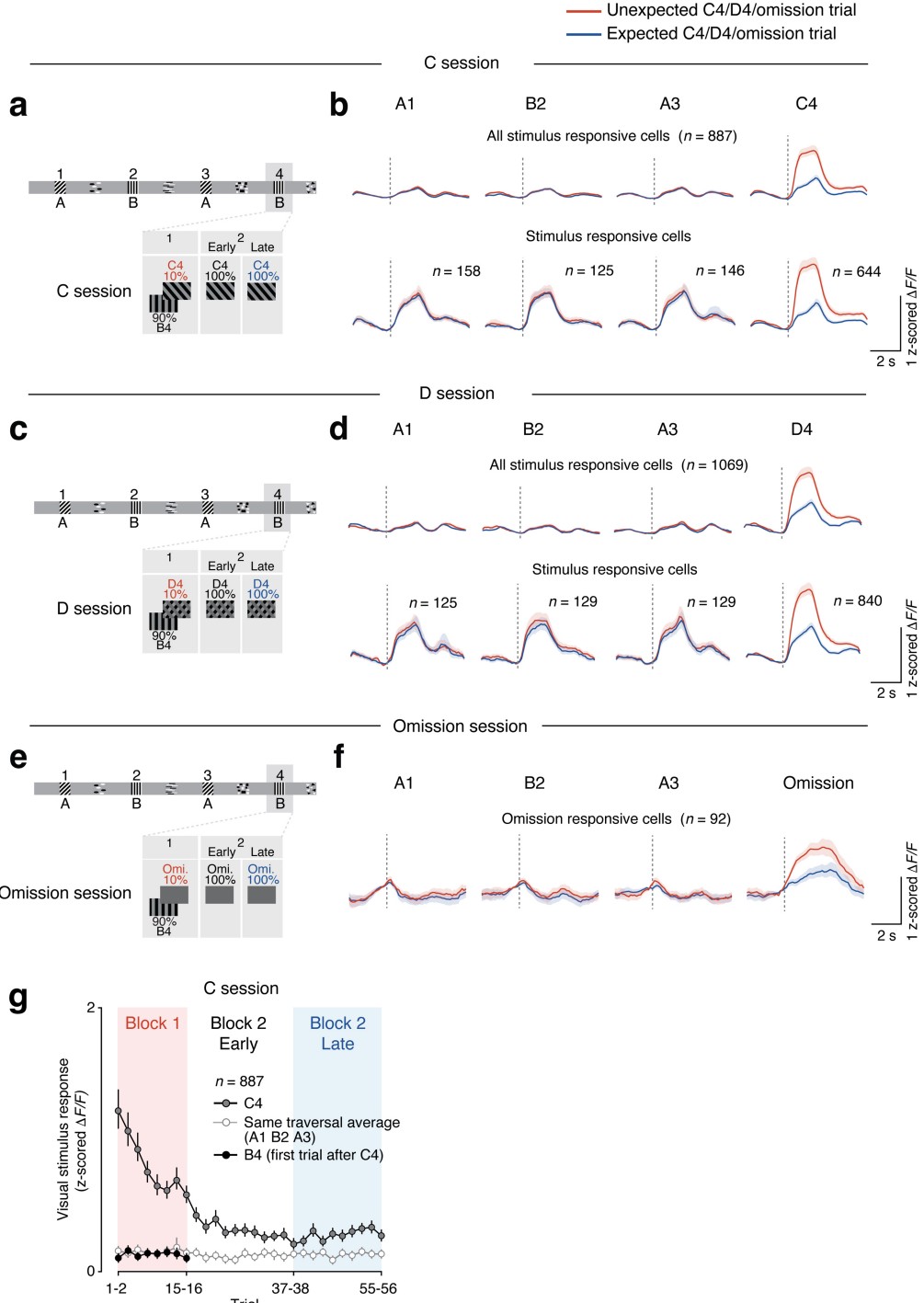

**Extended Data Fig. 2 | Average calcium responses in V1 to expected and unexpected stimuli and unexpected stimulus omissions. a**, Schematic of visual stimuli shown in a C session (unexpected C stimulus presented in 10% of trials instead of B at position 4 in block 1 and in 100% of trials in block 2). **b**, Top: Average calcium responses of all cells significantly responsive to any of the presented gratings in unexpected C4 (block 1) or expected C4 (late block 2) trials ($n$ = 887). Dotted line indicates grating stimulus onset. Bottom: Average calcium responses as on top, but only of neurons significantly responsive to each presented grating stimulus ($n$ = 158, 125, 146, 644; Gratings A1, B2, A3, C4 responsive cells). Same as Fig. 1c. Data from 9 mice. **c**, Schematic of visual stimuli shown in a D session (unexpected D stimulus presented in 10% of trials

instead of B at position 4 in block 1 and in 100% of trials in block 2). **d**, Same as b but responses to stimulus D4 on the right. Top: $n$ = 1,069. Bottom: $n$ = 125, 129, 129, 840; A1, B2, A3, D4 responsive cells. Data from 7 mice. **e**, Schematic of visual stimuli shown in an omission session (stimulus B4 omitted in 10% of trials in block 1 and in 100% of trials in block 2). **f**, Average calcium responses to gratings A1, B2, A3 and omission of B4 of all omission responsive cells ($n$ = 92 from 5 mice). Lines and shading are mean and bootstrap 95% CI. **g**, Average calcium responses to C4 (dark grey), average responses to A1, B2, A3 (light grey) and to B4 (black) of all cells significantly responsive to any of the presented gratings in block 1 or late block 2 trials ($n$ = 887). Symbols and error bars depict mean and bootstrap 95% CI.

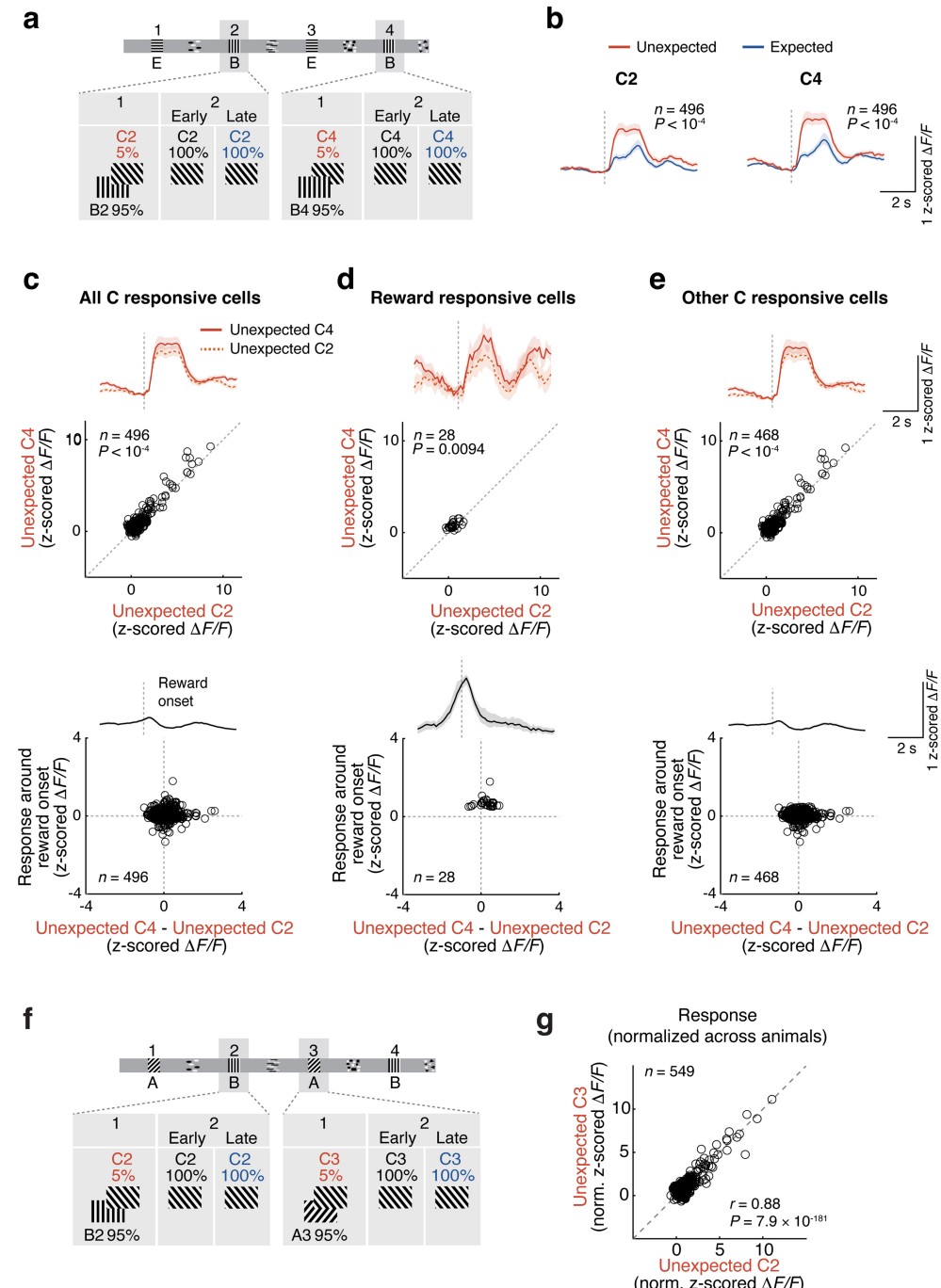

**Extended Data Fig. 3 | Prediction error responses to the same unexpected visual stimulus encountered at different locations. a**, Experimental design. Grating stimulus C was presented at position 2 (C2) or at position 4 (C4) in 5% of trials each in block 1. **b**, Average calcium responses to unexpected (red) and expected (blue) C2 (left, $n = 496$ from 5 mice, $P < 1 \times 10^{-4}$, hierarchical bootstrapping test) and C4 (right, $n = 496$ from 5 mice, $P < 1 \times 10^{-4}$). Cells responsive to unexpected or expected C2 or C4 were pooled. Lines and shading indicate mean and bootstrap 95% CI. **c**, Top: average calcium responses to unexpected C2 (dotted red) and unexpected C4 (red, $n = 496$ from 5 mice, $P < 1 \times 10^{-4}$, hierarchical bootstrapping test) and responses of individual V1 neurons to C2 plotted against their responses to C4. Bottom: average calcium response

aligned to reward onset of all neurons and reward response (from 0.5 s before to 0.5 s after reward onset) plotted against difference in response to unexpected C2 and unexpected C4 ($n = 496$ from 5 mice). **d**, Same as c, but only for reward responsive cells ($n = 28$ from 5 mice, $P = 0.0094$, hierarchical bootstrapping test). **e**, Same as c, but for the remaining, reward non-responsive cells ($n = 468$ from 5 mice, $P < 1 \times 10^{-4}$, hierarchical bootstrapping test). **f**, Experimental design. Grating C was presented at position 2 (C2) or at position 3 (C3) in 5% of trials each in block 1. Same as Fig. 1f. **g**, Same as Fig. 1h, but responses are normalized across animals by mean response of all responsive neurons (C2 or C3) of individual animals ($n = 549$ from 9 mice, Pearson correlation: $r = 0.88$, $P = 7.9 \times 10^{-181}$).

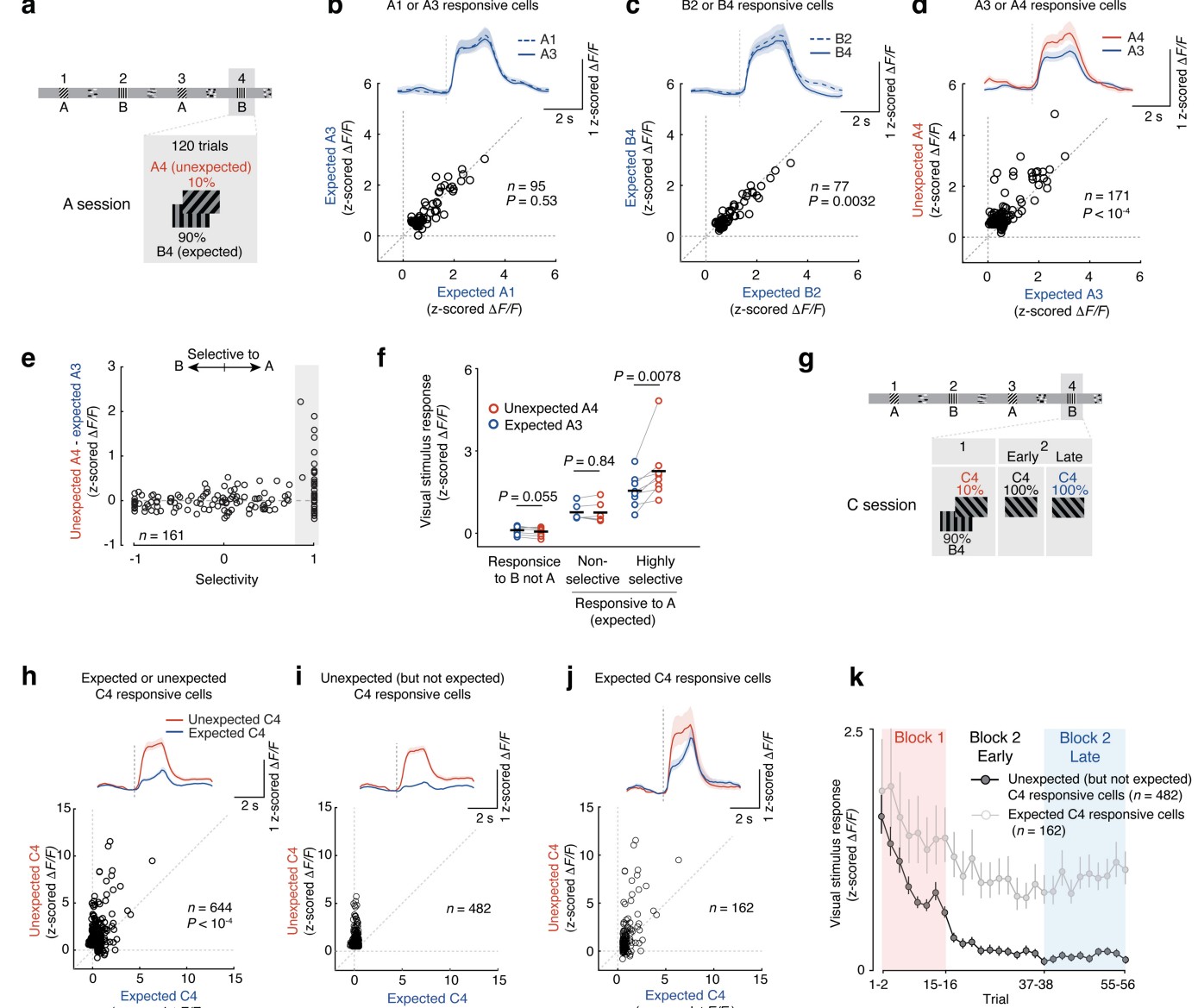

**Extended Data Fig. 4 | Prediction error signal in response to a familiar visual stimulus encountered at an unexpected location (grating A presented at location 4 instead of grating B). a**, Schematic of experimental design (A session). **b**, Calcium responses of individual V1 neurons to expected grating A1 plotted against responses to expected grating A3. Neurons responsive to either A1 or A3 were included in the analysis ($n$ = 95 cells from 8 mice, $P$ = 0.53, hierarchical bootstrapping test). **c**, Calcium responses of individual V1 neurons to expected grating B2 plotted against responses to expected grating B4. Neurons responsive to either B2 or B4 were included in the analysis ($n$ = 77 cells from 8 mice, $P$ = 0.0032, hierarchical bootstrapping test). **d**, Calcium responses of individual V1 neurons to expected grating A3 plotted against responses to unexpected grating A4. Neurons responsive to either A3 or A4 were included in the analysis ($n$ = 171 cells from 8 mice, $P < 1 \times 10^{-4}$, hierarchical bootstrapping test). **e**, Strength of prediction error signal (difference in response to unexpected grating A4 and expected grating A3) plotted against grating response selectivity (difference in response to grating A3 and grating B2 divided by the sum of responses) for all cells responsive to expected gratings. **f**, Cell-averaged response strength to expected grating A3 (blue) and unexpected grating A4 (red) of B-selective (left, $n$ = 8 mice, $P$ = 0.055,

two-sided signed-rank test), and non-selective (selectivity A3 vs B2 < 0.6, left, $n$ = 8 mice, $P$ = 0.84, two-sided signed-rank test) and highly selective (selectivity A3 vs B2 > 0.8, right, $n$ = 8 mice, $P$ = 0.0078) grating A3 responsive cells. Data points depict mean responses for individual animals, $n$ = 8 mice, black horizontal bars indicate mean across animals. **g**, Schematic of experimental design (C session). **h**, Same as d, but for C session. Calcium responses of individual V1 neurons to expected grating C4 plotted against responses to unexpected grating C4. Neurons responsive to either expected for unexpected C4 were included in the analysis ($n$ = 644 cells from 9 mice, $P < 1 \times 10^{-4}$, hierarchical bootstrapping test). **i**, Calcium responses of individual V1 neurons to expected grating C4 plotted against responses to unexpected grating C4. Neurons responsive to unexpected C4 but not expected C4 were included in the analysis ($n$ = 482 cells from 9 mice). Cell- and trial-averaged calcium responses of the same cells to unexpected C4 (red) and expected C4 (blue) were plotted on top. **j**, Same as i, but for neurons responsive to expected C4 ($n$ = 162 from 9 mice). **k**, Average calcium responses to C4 of neurons responding to unexpected but not expected C4 (dark grey, $n$ = 482) and of neurons responding to expected C4 (light grey, $n$ = 162) across trials and blocks. Symbols and error bars depict mean and bootstrap 95% CI.

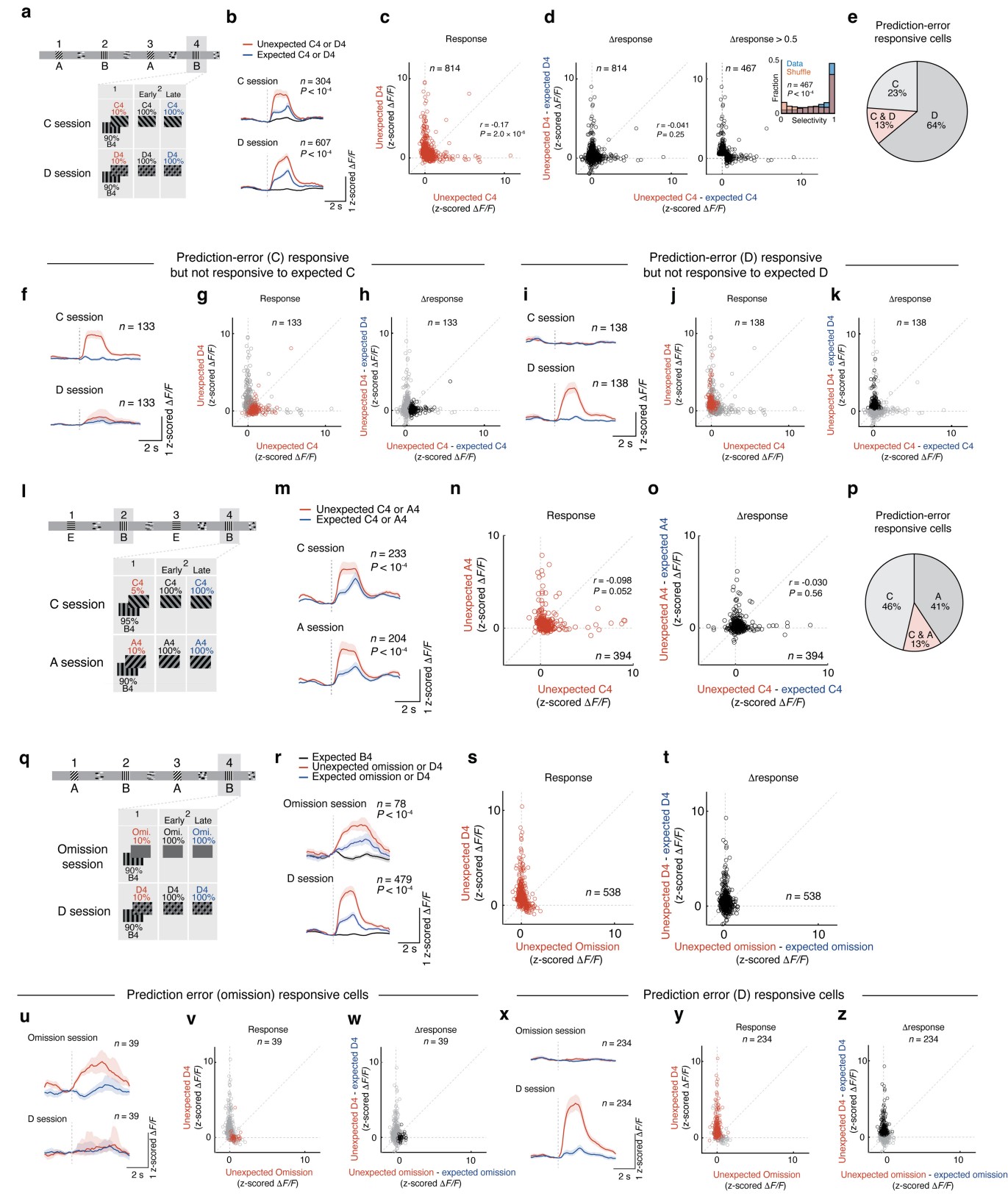

**Extended Data Fig. 5 |** See next page for caption.

**Extended Data Fig. 5 | Prediction error responses of layer 2/3 cells in V1 to different visual stimuli. a-d**, Same as Fig. 1k–n, but excluding VIP neurons which were labelled with tdTomato in these experiments. **b**, Top: cell- and trial-averaged calcium responses of C4-responsive neurons to expected B4 (black, block 1), unexpected C4 (red, block 1) and expected C4 (blue, late block 2). $n = 304$ cells from 5 mice, $P < 1 \times 10^{-4}$, Hierarchical bootstrapping test. Bottom: cell- and trial-averaged calcium responses to expected B4 (black), unexpected D4 (red) and expected D4 (blue). $n = 607$ cells from 5 mice, $P < 1 \times 10^{-4}$, Hierarchical bootstrapping test. Lines and shading are mean and bootstrap 95% CI. **c**, Calcium responses of individual V1 neurons to unexpected D4 plotted against unexpected C4 responses. Pearson correlation: $r = -0.17$, $P = 2.0 \times 10^{-6}$, $n = 814$ cells from 5 mice. **d**, Left: the difference of responses between unexpected and expected D4 plotted against the difference of responses between unexpected and expected C4. Pearson correlation: $r = -0.041$, $P = 0.25$, $n = 814$ cells from 5 mice. Right: same as on the left but excluding neurons not responsive to C and D. Inset: distribution of prediction error absolute selectivity $|(C - D)/(C + D)|$ of V1 neurons in the right scatter plot compared to a shuffled data set. V1 responses to the two stimuli are more selective than expected by chance. $n = 467$ cells from 5 mice, $P < 1 \times 10^{-4}$, randomization test. **e**, Pie chart with proportion of prediction-error responsive non-VIP neurons in V1 for stimulus C, stimulus D, or both (see Methods). $n = 960$ cells from 5 mice. **f-h**, Same as b-d, but for cells responsive to prediction error (C), but not responsive to expected C4 ($n = 133$). **i-k**, Same as f-h, but for cells responsive to prediction error (D), but not responsive to expected D4 ($n = 138$). **l**, Experimental design. Gratings C or A were presented at position 4 (C4 and A4) in 5% or 10% of trials in different sessions (C and A sessions, respectively). Note that a horizontal grating E was presented at position 1 and 3 in these experiments and during training. **m-p**, Same as b-e, but for comparison of unexpected C4 and unexpected A4 responses. **m**, Top: cell- and trial-averaged calcium responses of C4-responsive neurons to unexpected C4 (red, block 1) and expected C4 (blue, late block 2). $n = 233$ cells from 3 mice, $P < 1 \times 10^{-4}$, Hierarchical bootstrapping test. Bottom: cell- and trial-averaged calcium responses to unexpected A4 (red) and expected A4 (blue). $n = 204$ cells from 3 mice, $P < 1 \times 10^{-4}$, Hierarchical bootstrapping test. Lines and shading are mean and bootstrap 95% CI. **n**, Calcium responses of individual V1 neurons to unexpected A4 plotted against unexpected C4. Pearson correlation: $r = -0.098$, $P = 0.052$, $n = 394$ cells from 3 mice. **o**, The difference of responses between unexpected and expected A4 plotted against the difference of responses between unexpected and expected C4. Pearson correlation: $r = -0.030$, $P = 0.56$, $n = 394$ cells from 3 mice. **p**, Pie chart with proportion of prediction-error responsive non-VIP neurons for stimulus C, stimulus A, or both (see Methods). $n = 464$ cells from 3 mice. **q**, Schematic of visual stimuli shown in an omission session (stimulus B4 omitted in 10% of trials in block 1 and in 100% of trials in block 2) and a D session (stimulus D was presented at position 4 in 10% of trials in block 1 and in 100% of trials in block 2). **r**, Average V1 calcium responses to unexpected (red) and expected (blue) omission (top, $n = 78$ from 4 mice, $P < 1 \times 10^{-4}$) and D4 (bottom, $n = 479$ from 4 mice, $P < 1 \times 10^{-4}$). Hierarchical bootstrapping test. Lines and shading indicate mean and bootstrap 95% CI. **s**, Responses to unexpected omission plotted against responses to unexpected D4 for individual V1 layer 2/3 neurons ($n = 538$ cells from 4 mice). **t**, Difference between responses to unexpected omission and expected omission of B4 plotted against response difference between unexpected D4 and expected D4 stimulus responses for individual V1 layer 2/3 neurons. **u-w**, Same as r-t, but for cells with a significant difference in response between expected and unexpected stimulus omission ($n = 39$). **x-z**, Same as r-t, but for cells with a significant difference in response between expected and unexpected stimulus D4 ($n = 234$).

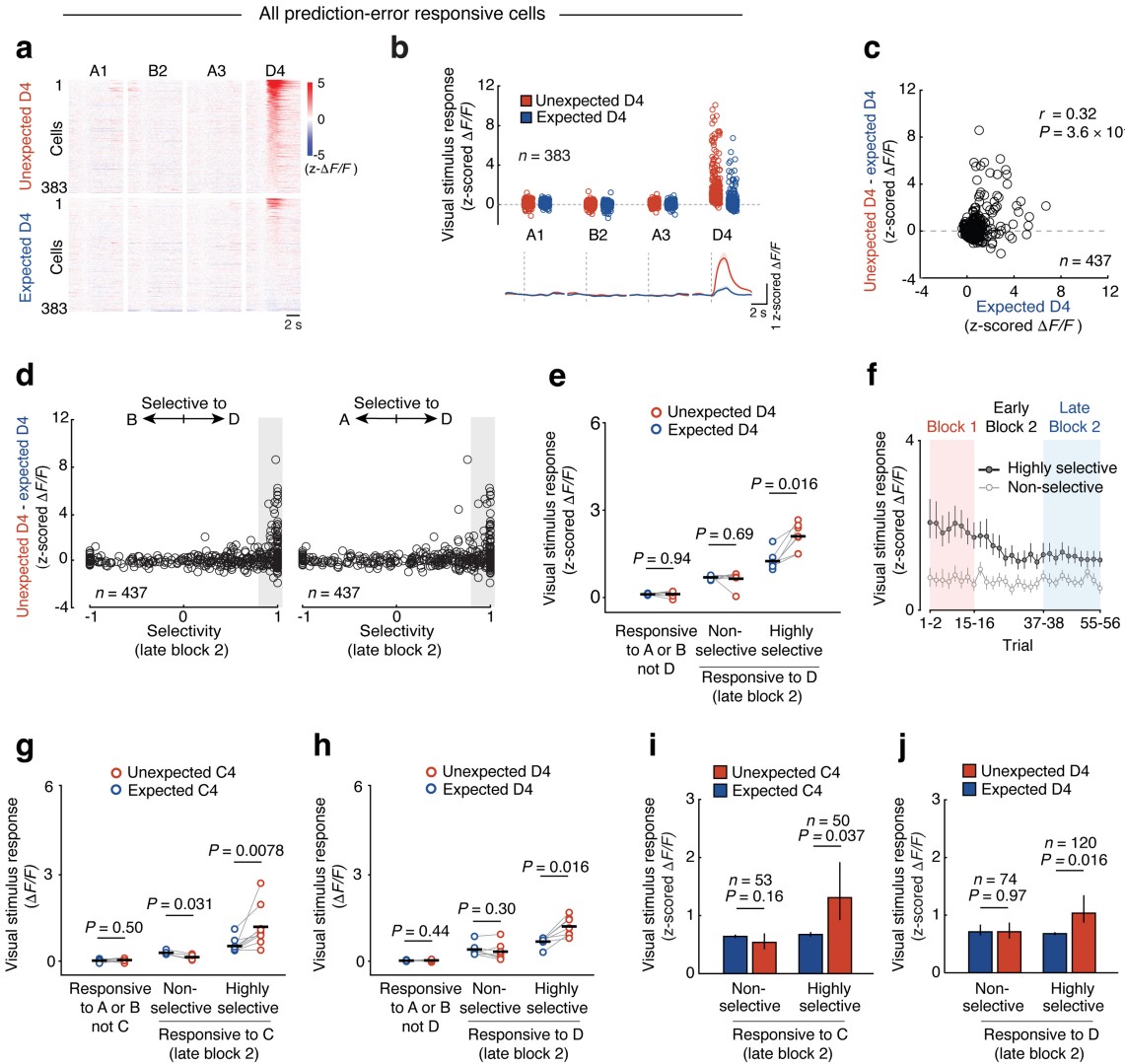

**Extended Data Fig. 6 | Prediction error specifically boosts neurons most selective to the presented visual stimulus (stimulus D). a-f**, Same as Fig. 2, but for a second unexpected visual stimulus D. **a**, Single-cell responses for all prediction-error responsive cells (individual rows) (*n* = 383 cells, 7 mice) to visual stimuli A1, B2, A3 and D4 in unexpected D4 (top) and expected D4 (bottom) trials, sorted by response to unexpected D4. **b**, Top: calcium responses for all prediction-error responsive cells (individual dots) (*n* = 383 cells, 7 mice) to visual stimuli A1, B2, A3 and D4 in unexpected D4 (red) and expected D4 (blue) trials. Bottom: Cell-averaged calcium responses. Lines and shading are mean and bootstrap 95% CI. **c**, Difference in response strength between unexpected (block 1) and expected D4 (late block 2) for all visual stimulus-responsive cells in late block 2, plotted against response to expected D4 (late block 2) for individual neurons. *r* = 0.32, *P* = 3.6 × 10⁻¹², Pearson correlation; *n* = 437, 7 mice. **d**, Left: difference in response strength between unexpected and expected D4 responses of individual neurons, plotted against their response selectivity to stimulus D vs. stimulus B in late block 2 (difference in response strength between expected D4 and B2 divided by the sum of responses to both stimuli) for all neurons responsive to at least one of the visual stimuli in late block 2. −1 indicates only responsive to B, +1 only responsive to D, and 0 equal responses to both. Right: same as on the left but for response selectivity to stimulus D vs. stimulus A in late block 2. **e**, Mean responses to

expected (blue) and unexpected (red) stimulus D4 of A or B selective cells (left, *n* = 7 mice, *P* = 0.94; two-sided signed-rank test), and non-selective (selectivity towards D, compared to B < 0.6, middle, *n* = 7 mice; *P* = 0.69; two-sided signed-rank test) and highly selective (selectivity towards D, compared to B > 0.8, right, *n* = 7 mice; *P* = 0.016) stimulus D4 responsive cells in late block 2. Data points depict mean responses for individual animals, *n* = 7 mice, black horizontal bars indicate mean across animals. **f**, Mean calcium responses to stimulus D4 over all trials in the imaging session of highly selective (dark grey, *n* = 185) and non-selective (light grey, *n* = 75) stimulus D4 responsive cells in block 2. Responses were averaged over two trials. Error bars are bootstrap 95% CI. **g**, Same as Fig. 2e, but showing responses as raw *ΔF/F₀* without z-scoring. **h**, Same as g, but for sessions with unexpected stimulus D, equivalent to panel e. **i**, Same as Fig. 2e, but highly selective cells were sub-selected to match their average response strength to the expected stimulus C4 with the average response to expected stimulus C4 of non-selective cells. To achieve this, highly selective cells that responded strongly to expected gratings (top 35%) were removed from the analysis. Hierarchical bootstrapping test. Bars and error bars are mean and 95% bootstrap CI. **j**, Same as i, but for sessions with unexpected stimulus D, equivalent to panel **e**, but with matched average response strength to expected stimulus D4 of highly selective and non-selective V1 cells.

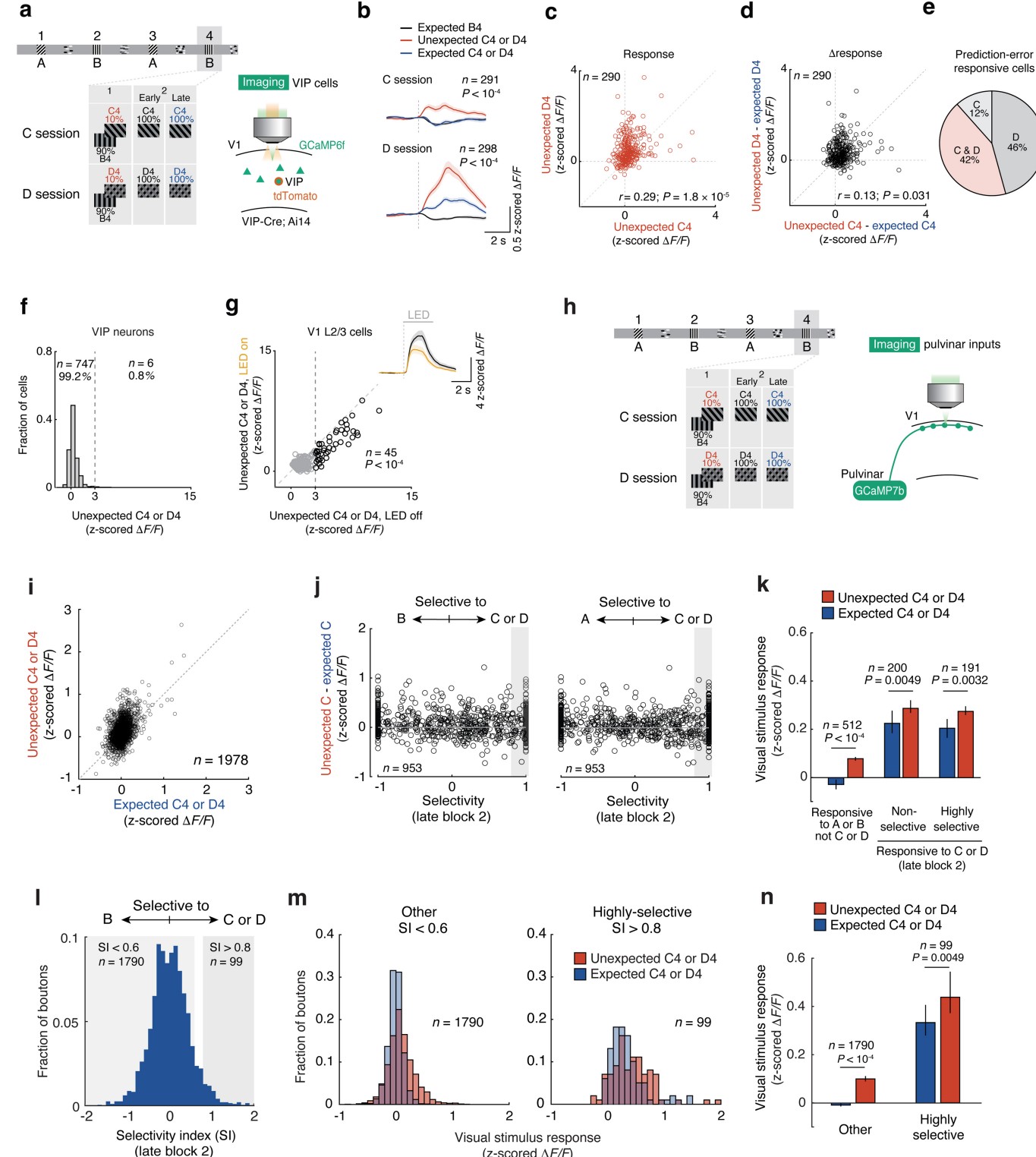

**Extended Data Fig. 7 |** See next page for caption.

**Extended Data Fig. 7 | Prediction error responses of VIP cells to different visual stimuli, effect of optogenetic VIP neuron silencing on strongly responding layer 2/3 cells, and broad facilitation of pulvinar inputs by prediction errors. a**, Schematic of the experimental design. Stimulus C or D was presented at position 4 (C4 and D4) in 10% of trials in different sessions (C and D sessions, respectively). Calcium activity of VIP cells in V1 layer 2/3 was recorded. **b**, Top: cell- and trial-averaged VIP calcium responses to expected B4 (black), unexpected C4 (red, block 1) and expected C4 (blue, late block 2). $n = 291$ VIP cells from 5 mice, $P < 1 \times 10^{-4}$, Hierarchical bootstrapping test. Bottom: cell- and trial-averaged VIP calcium responses to expected B4 (black), unexpected D4 (red) and expected D4 (blue). $n = 298$ VIP cells from 5 mice, $P < 1 \times 10^{-4}$, Hierarchical bootstrapping test. Lines and shading are mean and bootstrap 95% CI. **c**, Calcium responses of individual VIP neurons to unexpected D4 plotted against responses to unexpected C4. Pearson correlation: $r = 0.29$, $P = 1.8 \times 10^{-5}$, $n = 290$ from 5 mice. **d**, Difference of responses to unexpected and expected stimulus D4 plotted against the difference of unexpected and expected C4 responses (Pearson correlation: $r = 0.13$, $P = 0.031$, $n = 290$ from 5 mice. **e**, Pie chart with proportion of prediction-error responsive VIP cells for stimulus C, stimulus D, or both (see Methods). $n = 199$ from 5 mice. **f**, Distribution of stimulus response strength for VIP cells to unexpected C4 or D4 ($n = 753$, 14 sessions from 7 mice). **g**, Same as Fig. 3f, but only cells exhibiting a visual stimulus response of more than 3 z-scored $\Delta F/F$ were included in order to avoid inclusion of opsin-expressing and therefore directly silenced VIP cells, which cannot be visually identified in these experiments ($n = 45$ from 7 sessions; $P < 1 \times 10^{-4}$; Hierarchical bootstrapping test). Neurons indicated in black have responses > 3 z-scored $\Delta F/F$. Inset: Responses to unexpected stimulus C4 or D4 of V1 layer 2/3 cells with (amber) or without (black) VIP silencing. Lines and shading are mean and bootstrap 95% CI. **h**, Experimental design. The calcium activity of axonal boutons of pulvinar projections in V1 L1 was recorded. **i**, Stimulus responses of individual pulvinar boutons to unexpected C4 or D4 plotted against responses to expected C4 or D4. $n = 1,978$ pulvinar boutons from 10 sessions, 7 mice. **j**, Left: difference in response strength between unexpected and expected C4 or D4 responses of individual neurons, plotted against their response selectivity to stimulus C or D vs. stimulus B in late block 2 (difference in response strength between expected C4 or D4 and B2 divided by the sum of responses to both stimuli) for all boutons responsive to at least one of the visual stimuli in late block 2. −1 indicates only responsive to B, +1 only responsive to C or D, and 0 equal responses to both. Right: same as on the left but for response selectivity to stimulus C or D vs. stimulus A in late block 2. **k**, Mean responses to expected (blue) and unexpected (red) stimuli C4 or D4, of boutons selective to A or B (left, $n = 512$ boutons; $P < 1 \times 10^{-4}$; Hierarchical bootstrapping test), and non-selective (selectivity towards C, compared to B < 0.6, middle, $n = 200$ boutons; $P = 0.0049$) and highly selective (selectivity towards C, compared to B > 0.8, right, $n = 191$ boutons; $P = 0.0032$) grating C4 responsive neurons in late block 2. Bars and error bars are mean and 95% bootstrap CI. **l**, Distribution of selectivity index (difference in response strength between expected C4 or D4 and B2 divided by the pooled standard deviation, see methods) for all pulvinar boutons. **m**, Distribution of stimulus response strength of non-selective (selectivity index C4/D4, compared to B2 < 0.6, left) and highly selective (selectivity index > 0.8, right) pulvinar boutons to unexpected C4 or D4 (red) and expected C4 or D4 (blue). **n**, Cell- and trial-averaged stimulus responses to expected C4 or D4 (blue) and unexpected C4 or D4 (red), of non-selective (left) and highly selective (right) pulvinar boutons. $n = 1,790$ and $n = 99$; $P < 1 \times 10^{-4}$, $P = 0.0049$; non-selective and highly selective boutons, hierarchical bootstrapping test. Bars and error bars are mean and 95% bootstrap CI.

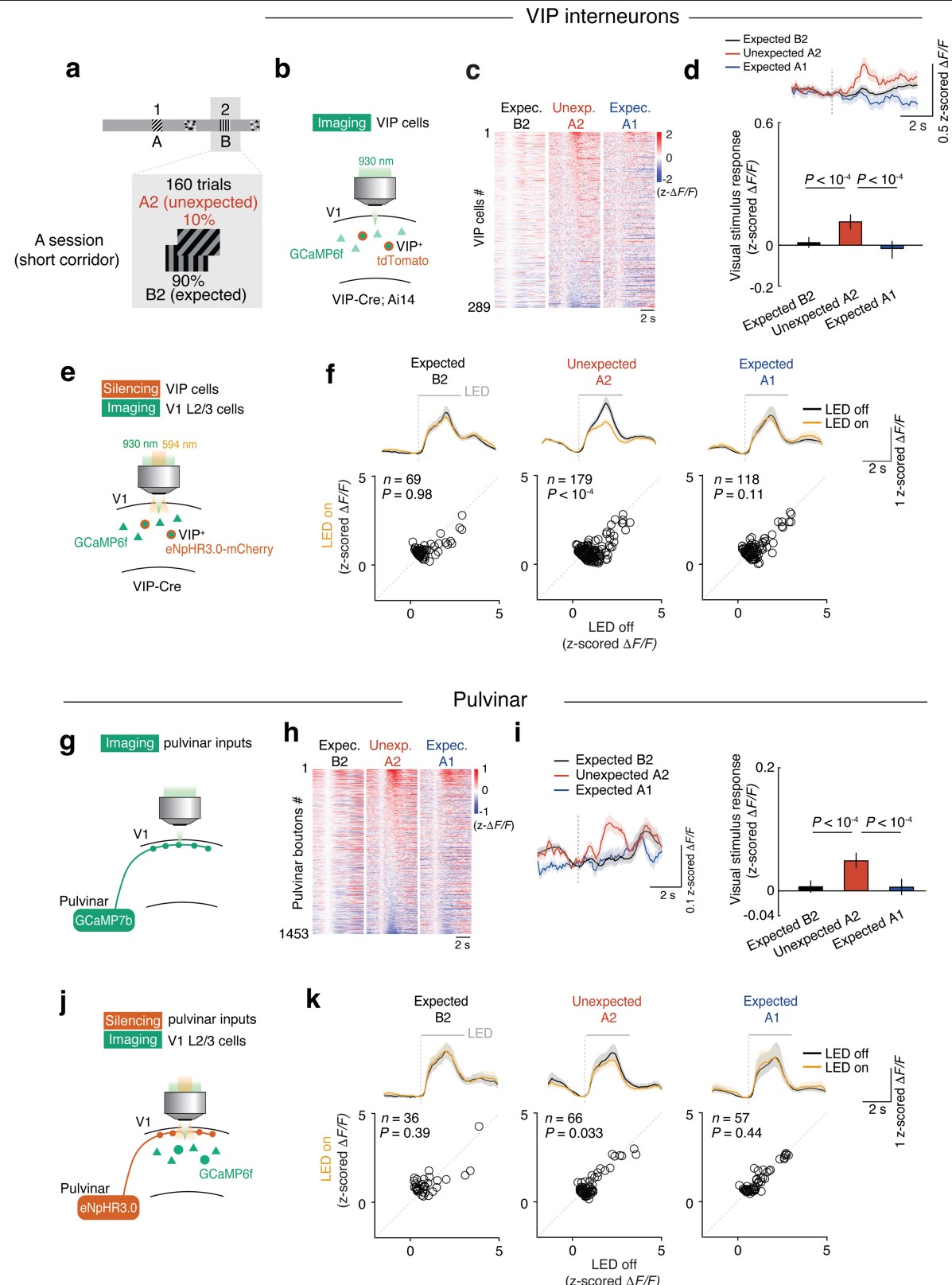

**Extended Data Fig. 8 |** See next page for caption.

**Extended Data Fig. 8 | Activity of VIP interneurons and pulvinar input is required for prediction error signals to familiar visual stimulus presented at unexpected location. a**, Schematic of the experimental design. For the experiments in this figure a shorter virtual corridor was employed as depicted. **b**, Calcium activity of VIP cells in V1 layer 2/3 was recorded during the experiment. **c**, Single-cell responses for all VIP cells (individual rows) in the A session ($n = 289$ cells from 5 mice) to expected B2 (left), unexpected A2 (middle) and expected A1 (right), sorted by response strength to unexpected A2. **d**, Cell- and trial-averaged calcium responses of all VIP cells ($n = 289$) to expected B2 (black), unexpected A2 (red) and expected A1 (blue). Lines and bars are mean, shading and error bars indicate bootstrap 95% CI. $P < 1 \times 10^{-4}$ for all comparisons between expected and unexpected stimuli; Hierarchical bootstrapping test with Bonferroni correction. **e**, Schematic of the experiment. Calcium activity of V1 layer 2/3 cells was recorded while VIP cells were optogenetically silenced. VIP cell silencing started at the onset of visual stimuli and lasted for 3 s. **f**, Top: cell- and trial-averaged responses of V1 neurons significantly responsive to the presented grating stimuli to expected grating B2 (left, $P = 0.98$, Hierarchical bootstrapping test, $n = 69$ cells, 5 mice), unexpected grating A2 (middle, $P < 1 \times 10^{-4}$, $n = 179$) and expected grating A1 (right, $P = 0.11$, $n = 118$) with (amber) or without (black) VIP silencing. Lines and shading are mean and bootstrap 95% CI. Bottom: responses of individual neurons to the grating stimulus indicated above during VIP silencing (LED on), plotted against responses to the same stimulus in control trials (LED off). **g-i**, Same as b-d, but for calcium responses of pulvinar axonal boutons in V1 (see Methods). **g**, Calcium activity of axonal boutons of pulvinar projections was recorded in V1 layer 1. **h**, Single-bouton responses for all pulvinar axonal boutons (individual rows) in the A session ($n = 1,453$ boutons, 6 mice) to expected B2 (left), unexpected A2 (middle) and expected A1 (right), sorted by response strength to unexpected A2. **i**, Bouton- and trial-averaged calcium responses of all pulvinar boutons ($n = 1,453$ boutons) to expected grating B2 (black), unexpected grating A2 (red) and expected grating A1 (blue). Lines and bars are mean, shading and error bars indicate bootstrap 95% CI. $P < 1 \times 10^{-4}$ for all comparisons between expected and unexpected stimuli; Hierarchical bootstrapping test with Bonferroni correction. **j** and **k**, Same as e and f, but with optogenetic silencing of pulvinar axons. **j**, The activity of V1 layer 2/3 cells was recorded while pulvinar axons in V1 were optogenetically silenced (see Methods). **k**, Top: cell- and trial-averaged responses of neurons significantly responsive to the presented grating stimuli to expected grating B2 (left, $P = 0.39$, Hierarchical bootstrapping test, $n = 36$ cells, 5 mice), unexpected grating A2 (middle, $P = 0.033$, $n = 66$) and expected grating A1 (right, $P = 0.44$, $n = 57$) with (amber) or without (black) silencing of pulvinar axons. Lines and shading are mean and bootstrap 95% CI. Bottom: responses of individual neurons to the grating stimulus indicated above during silencing of pulvinar axons (LED on), plotted against responses to the same stimulus in control trials (LED off).

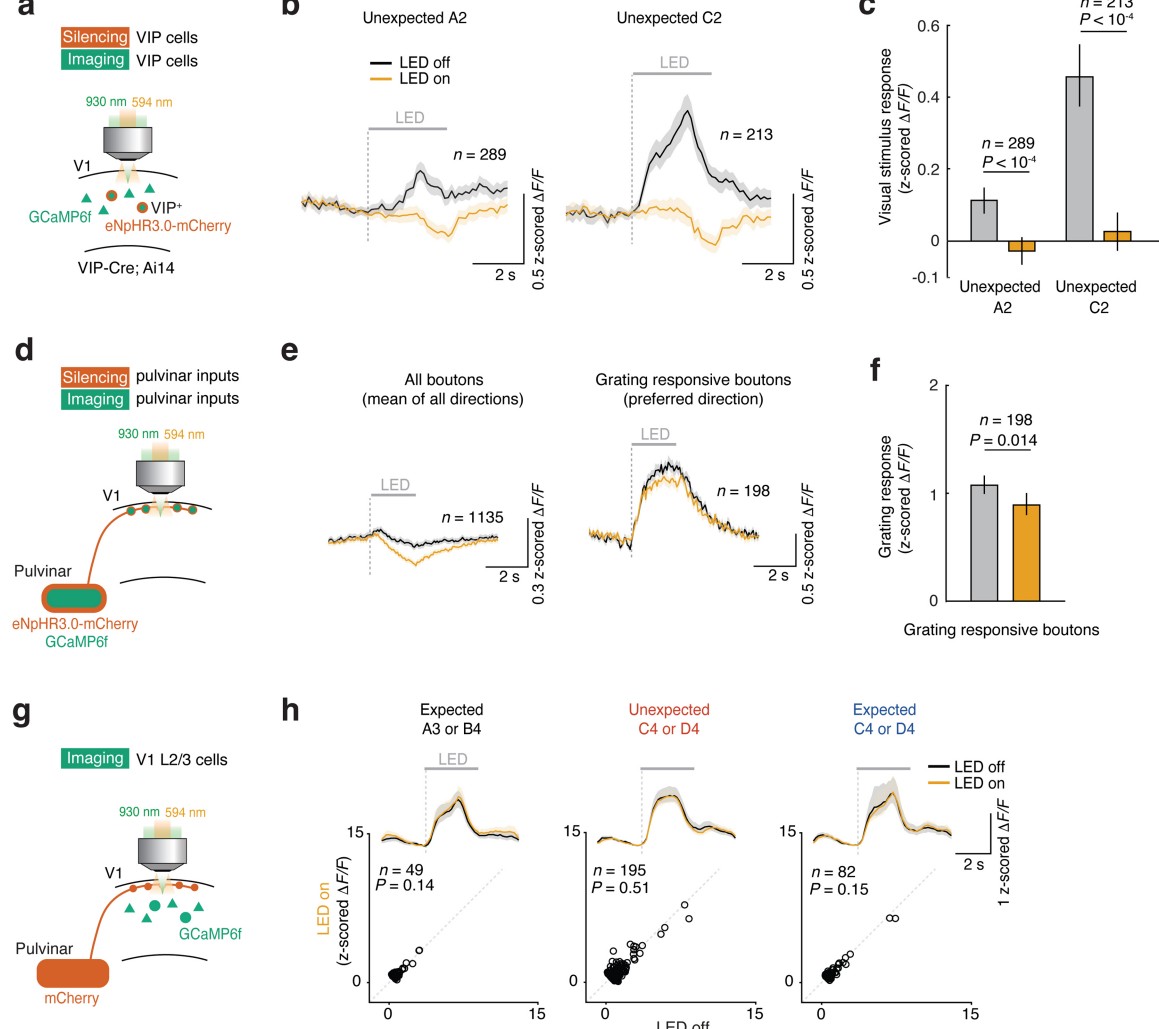

**Extended Data Fig. 9 | Confirmation of optogenetic silencing and control experiment for LED light stimulation. a**, Schematic of the experimental design. Calcium activity of VIP cells in V1 was recorded while they were optogenetically silenced. **b**, Cell- and trial-averaged responses of VIP cells to unexpected stimulus A2 (familiar stimulus at unexpected position; left, $n = 289$ from 5 mice) and unexpected stimulus C2 (right, $n = 213$ from 5 mice) with (amber) or without (black) VIP silencing. Lines and shading are mean and bootstrap 95% CI. **c**, Responses to unexpected A2 (left, $n = 289$, $P < 1 \times 10^{-4}$) and unexpected C2 (right, $n = 213$, $P < 1 \times 10^{-4}$) stimuli with (amber) and without (grey) VIP silencing. Hierarchical bootstrapping test. Bars and error bars are mean and 95% bootstrap CI. **d**, Schematic of the experimental design. Calcium activity of pulvinar boutons was recorded while they were optogenetically silenced during presentation of differently oriented, drifting grating stimuli (see Methods). **e**, Cell- and trial-averaged responses of all pulvinar boutons to all grating directions (left, $n = 1,135$ from 5 sessions, 3 mice) and of visually responsive boutons to the preferred grating direction (right, $n = 198$ from 5

sessions, 3 mice) with (amber) or without (black) pulvinar axonal silencing. Lines and shading are mean and bootstrap 95% CI. **f**, Responses of grating responsive boutons to preferred direction with (amber) and without (grey) pulvinar axonal silencing ($n = 198$, $P = 0.014$). Hierarchical bootstrapping test. Bars and error bars are mean and 95% bootstrap CI. **g**, Schematic of the experimental design. Calcium activity of V1 layer 2/3 cells was recorded during light stimulation without expression of opsins. mCherry was expressed in pulvinar neurons. **h**, Top: cell- and trial-averaged responses to expected stimulus A3 or B4 (left), unexpected stimulus C4 or D4 (middle) and expected C4 or D4 (right) with or without light stimulation (amber and black, respectively). Lines and shading are mean and bootstrap 95% CI ($n = 49, 195, 82$, 3 mice; $P = 0.14$, $P = 0.51$, $P = 0.15$; for expected A3 or B4 responsive cells, unexpected C4 or D4 responsive cells, and expected C4 or D4 responsive cells; Hierarchical bootstrapping test). Bottom: Responses of individual V1 neurons to stimuli indicated above with and without LED light stimulation (LED on vs LED off).

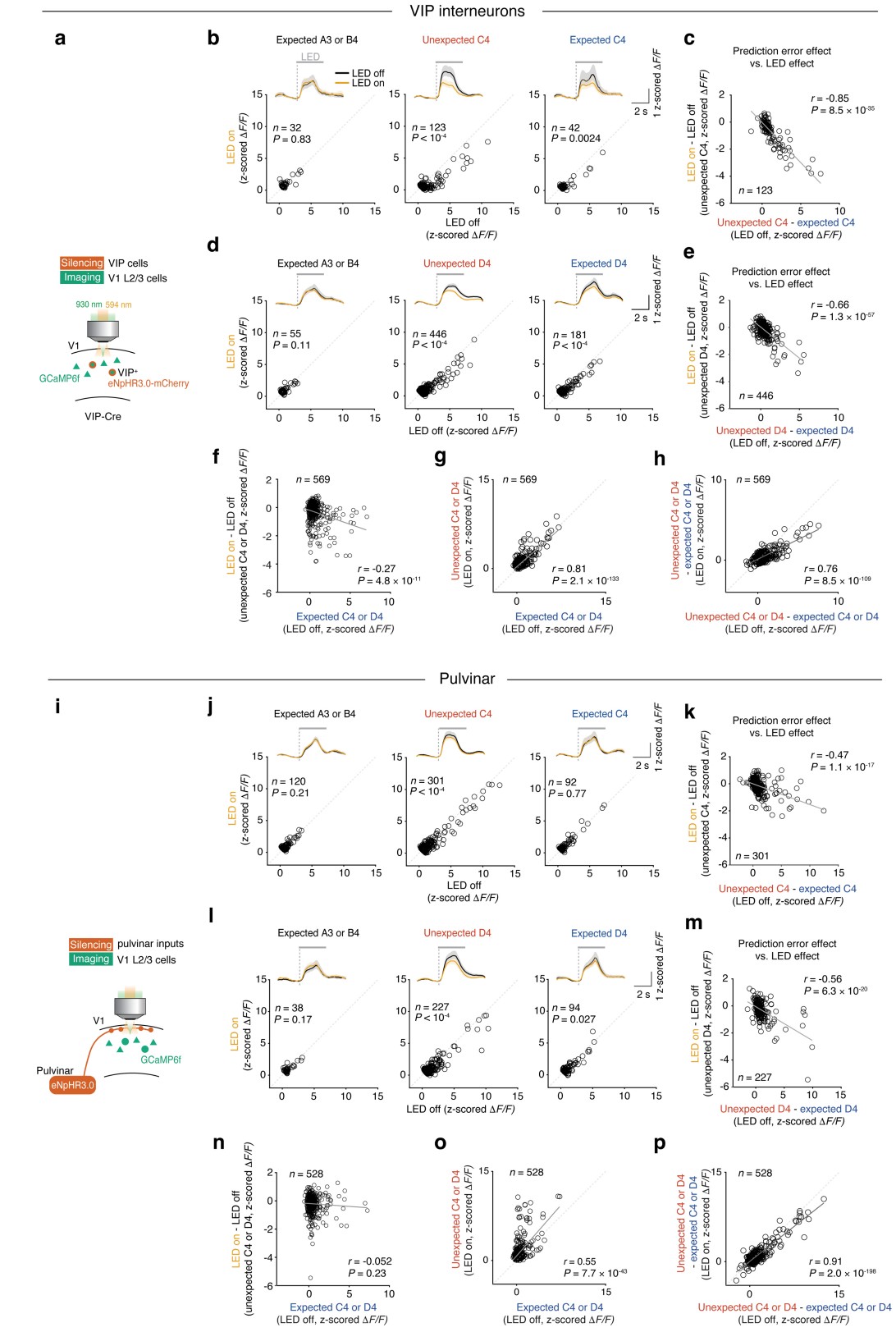

**Extended Data Fig. 10** | See next page for caption.

**Extended Data Fig. 10 | Effect of optogenetic silencing of VIP interneurons or pulvinar input to V1 in C and D sessions.** Related to Fig. 3. **a**-**e**, Same as Fig. 3e–g, but C session (b and c) and D session (d and e) are plotted separately. **a**, Schematic of the experiment. Calcium activity of V1 layer 2/3 cells was recorded while VIP cells were optogenetically silenced in 50% of trials. VIP cell silencing started at the onset of visual stimuli and lasted for 3 s. **b**, Top: cell- and trial-averaged responses of neurons significantly responsive to the presented stimuli to expected grating A3 or B4 (left, $P = 0.83$, Hierarchical bootstrapping test, $n = 32$ cells, 4 mice), unexpected grating C4 (middle, $P < 1 \times 10^{-4}$, $n = 123$) and expected grating C4 (right, $P = 0.0024$, $n = 42$) with (amber) or without (black) VIP silencing. Lines and shading are mean and bootstrap 95% CI. Bottom: responses of individual V1 neurons to the grating stimulus indicated above during VIP silencing (LED on), plotted against responses to the same stimulus in control trials (LED off). **c**, Effect of VIP neuron silencing (LED on - LED off during unexpected grating C4) plotted against the strength of prediction error signals (response to unexpected C4 - response to expected C4). Pearson correlation: $r = -0.85$, $P = 8.5 \times 10^{-35}$. **d**, Top: cell- and trial-averaged responses of neurons significantly responsive to the presented stimuli to expected A3 or B4 (left, $P = 0.11$, Hierarchical bootstrapping test, $n = 55$ cells, 3 mice), unexpected D4 (middle, $P < 1 \times 10^{-4}$, $n = 446$) and expected D4 (right, $P < 1 \times 10^{-4}$, $n = 181$) with (amber) or without (black) VIP silencing. Lines and shading are mean and bootstrap 95% CI. Bottom: responses of individual neurons to the visual stimulus indicated above during VIP silencing (LED on), plotted against responses to the same stimulus in control trials (LED off). **e** Effect of VIP neuron silencing (LED on - LED off during unexpected stimulus D4) plotted against the strength of prediction error signals (response to unexpected D4 - response to expected D4). Pearson correlation: $r = -0.66$, $P = 1.3 \times 10^{-57}$. **f**, Effect of VIP neuron silencing (LED on - LED off during unexpected C4 or D4) plotted against response to expected C4 or D4 for individual V1 neurons; $n = 569$. Pearson correlation: $r = -0.27$, $P = 4.8 \times 10^{-11}$. **g**, Response to unexpected C4 or D4 with VIP silencing plotted against response to expected C4 or D4 without VIP silencing; $n = 569$. Pearson correlation: $r = 0.81$, $P = 2.1 \times 10^{-133}$. **h**, Strength of prediction error signal (response to unexpected C4 or D4 - response to expected C4 or D4) with VIP silencing plotted against strength of prediction error signal without VIP silencing; $n = 569$. Pearson correlation: $r = 0.76$, $P = 8.5 \times 10^{-109}$. **i**-**p**, Same as a-h but for optogenetic silencing of pulvinar inputs. **i**, Calcium activity of V1 layer 2/3 cells was recorded while pulvinar inputs were optogenetically silenced in 50% of trials. **j**, Expected grating A3 or B4 (left, $P = 0.21$, Hierarchical bootstrapping test, $n = 120$ cells, 7 mice), unexpected grating C4 (middle, $P < 1 \times 10^{-4}$, $n = 301$) and expected grating C4 (right, $P = 0.77$, $n = 92$) responses with (amber) or without (black) pulvinar axon silencing. **k**, $n = 301$ cells. Pearson correlation: $r = -0.47$, $P = 1.1 \times 10^{-17}$. **l**, Expected stimuli A3 or B4 (left, $P = 0.17$, Hierarchical bootstrapping test, $n = 38$ cells, 2 mice), unexpected D4 (middle, $P < 1 \times 10^{-4}$, $n = 227$) and expected D4 (right, $P = 0.027$, $n = 94$) responses with (amber) or without (black) VIP silencing. **m**, $n = 227$. Pearson correlation: $r = -0.56$, $P = 6.3 \times 10^{-20}$. **n**, $n = 528$. Pearson correlation: $r = -0.0052$, $P = 0.23$. **o**, $n = 528$. Pearson correlation: $r = 0.55$, $P = 7.7 \times 10^{-43}$. **p**, $n = 528$. Pearson correlation: $r = 0.91$, $P = 2.0 \times 10^{-198}$.

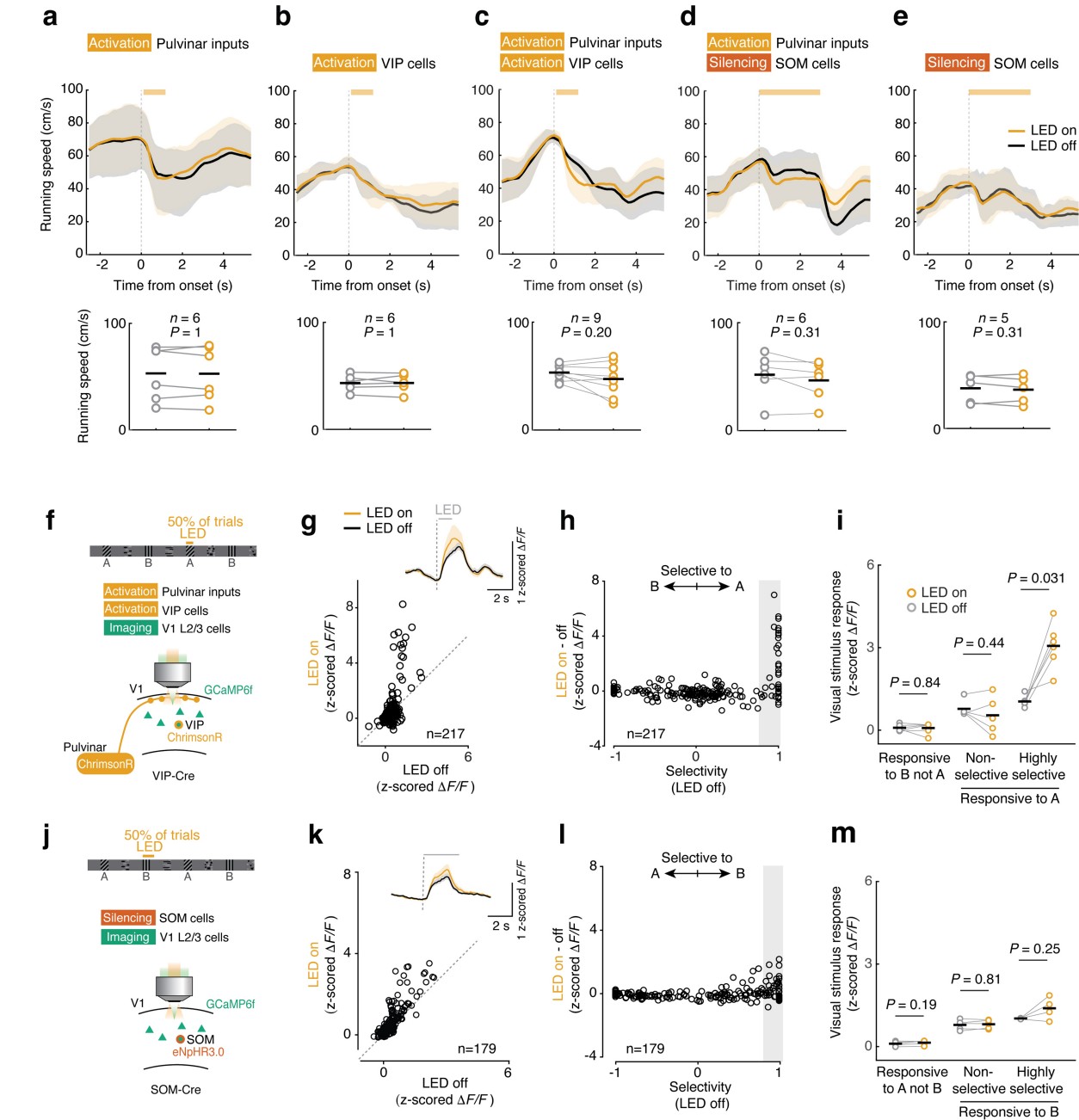

**Extended Data Fig. 11** | See next page for caption.

**Extended Data Fig. 11 | Effect of optogenetic manipulation of pulvinar inputs, VIP cells and SOM cells on running speed and visual responses of V1 layer 2/3 cells (related to Fig. 4). a-e**, Running speed with (amber) or without (black) optogenetic manipulation for activation of pulvinar axons (**a**), activation of VIP neurons (**b**), co-activation of pulvinar axons and VIP neurons (**c**), activation of pulvinar axons and simultaneous silencing of SOM cells (**d**), and silencing of SOM cells (**e**). Top: Lines and shading are mean and bootstrap 95% CI. Orange shading indicates time of optogenetic stimulation. Bottom: Data from the individual animals are shown separately. Data from the same animals are connected by lines. Black horizontal bars represent mean across animals. $P$-values from two-sided signed-rank test. **f-i**, Same as Fig. 4d, but optogenetic stimulation was paired with the grating stimulus A3 instead of B2. **f**, Schematic of the experimental design. The activity of V1 layer 2/3 cells was recorded while pulvinar axons and VIP cells were optogenetically co-stimulated. Stimulation started 0.1 s after visual stimulus onset and lasted for 1 s (see methods). **g**, Response strength to grating stimulus A3 with and without co-stimulation of pulvinar inputs and VIP cells. $n = 217$ grating A or B responsive cells, 6 sessions from 6 mice. Inset: Cell-averaged calcium responses with (amber) or without (black) optogenetic stimulation. **h**, Effect of optogenetic stimulation (difference of response to grating A3 with and without laser stimulation) plotted against response selectivity (difference in response strength between stimulus A and B divided by the sum of responses) of individual V1 neurons. **i**, Calcium response strength to grating stimulus A3 of B selective cells (left, $n = 6$ mice, $P = 0.84$), and non-selective (selectivity A vs B < 0.6, middle, $n = 6$ mice, $P = 0.44$, two-sided signed-rank test) and highly selective (selectivity A vs B > 0.8, right, $n = 6$ mice, $P = 0.031$, two-sided signed-rank test) grating A3 responsive cells in V1 layer 2/3 with (amber) or without (grey) optogenetic stimulation. **j-m**, Same as f-i, but for optogenetic silencing of SOM cells during presentation of grating stimulus B2. **j**, Schematic of the experimental design. The activity of V1 layer 2/3 cells was recorded while SOM cells were optogenetically silenced for 3 s, starting at grating stimulus onset. **k**, Grating B2 responses with and without the silencing of SOM cells. $n = 179$ grating A or B responsive cells, 5 sessions from 3 mice, $P < 1 \times 10^{-4}$, Hierarchical bootstrapping test. **l**, Effect of optogenetic stimulation (difference of response to grating B2 with and without laser stimulation) plotted against response selectivity (difference in response strength between stimulus B and A divided by the sum of responses) of V1 neurons. **m**, Calcium response strength to grating stimulus B2 of A-selective neurons (left, $n = 5$ sessions from 3 mice, $P = 0.19$), and non-selective (selectivity B vs A < 0.6, middle, $n = 5$ sessions from 3 mice, $P = 0.81$, two-sided signed-rank test) and highly selective (selectivity B vs A > 0.8, right, $n = 5$ sessions from 3 mice, $P = 0.25$, two-sided signed-rank test) grating B2 responsive cells in V1 layer 2/3 with (amber) or without (grey) optogenetic stimulation.

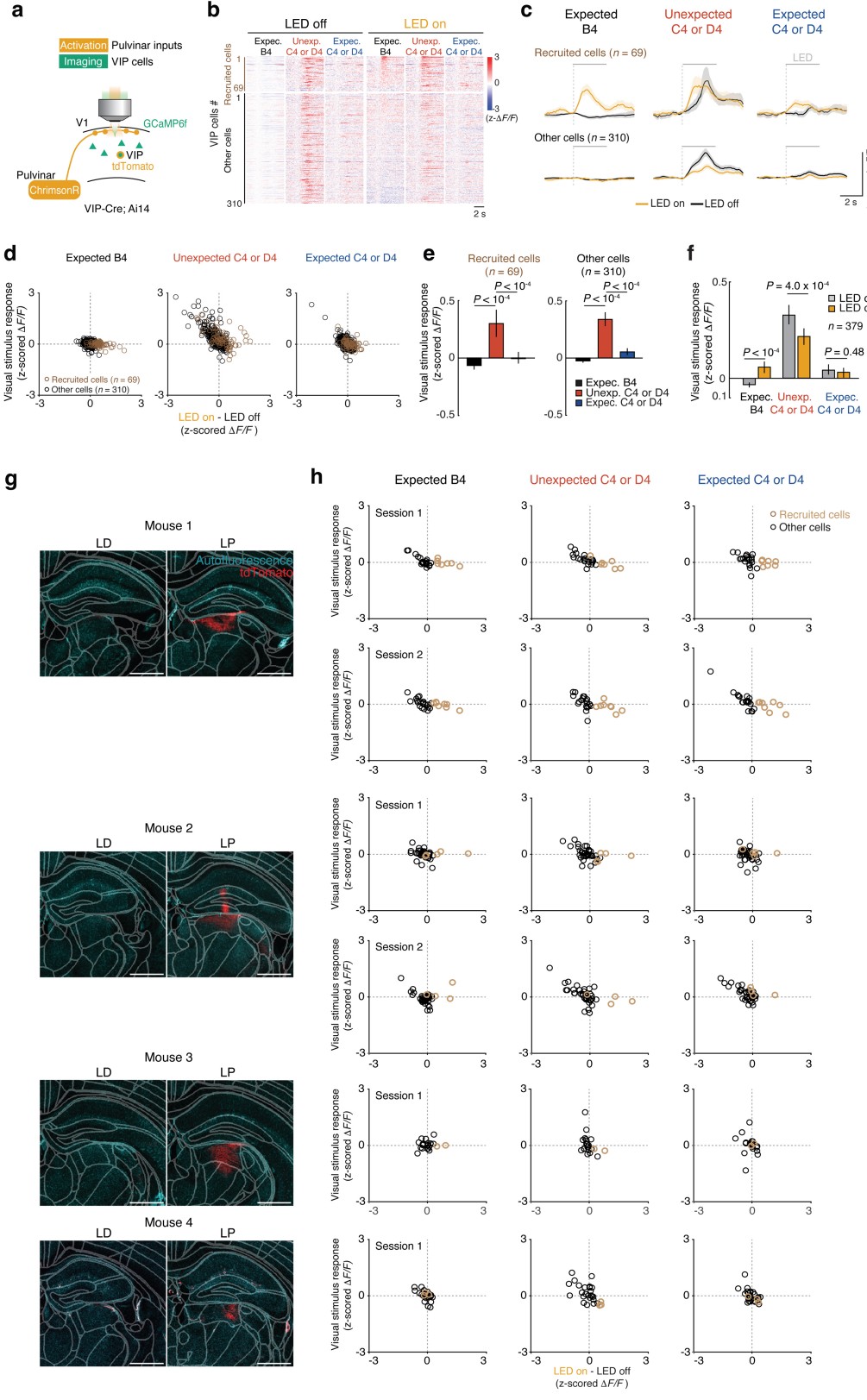

**Extended Data Fig. 12** | See next page for caption.

**Extended Data Fig. 12 | Effect of optogenetic stimulation of pulvinar inputs on visual responses of VIP cells in V1 layer 2/3, and expression of ChrimsonR-tdTomato and LED effect of individual animals used in Fig. 5. a**, Schematic of the experimental design. The activity of VIP cells was recorded while pulvinar axons were optogenetically stimulated for 3 s, starting at the onset of the visual stimulus. **b**, Single-cell responses of pulvinar-recruited VIP cells (individual rows, $n = 69$ cells, 7 sessions from 5 mice) and other non-recruited VIP cells (individual rows, $n = 310$ cells, 7 sessions from 5 mice) to expected B4, unexpected C4 or D4 and expected C4 or D4 stimuli with (right) and without (left) optogenetic stimulation (see Methods). **c**, Cell-averaged calcium responses with (amber) or without (black) optogenetic stimulation of pulvinar-recruited and other non-recruited VIP cells. Lines and shaded regions are mean and bootstrap 95% CI. **d**, Visual stimulus responses of individual VIP neurons without optogenetic stimulation plotted against the effect of pulvinar stimulation (difference of visual responses with and without optogenetic stimulation). **e**, Strength of calcium response to expected B4 (black), unexpected C4 or D4 (red) and expected C4 or D4 (blue) stimuli of pulvinar-recruited VIP cells (left, $n = 69$, B4 vs unexpected C4/D4: $P < 10^{-4}$; unexpected vs expected C4/D4: $P < 10^{-4}$, Hierarchical bootstrapping test with Bonferroni correction) and other VIP cells (right, $n = 310$, B4 vs unexpected C4/D4: $P < 10^{-4}$; unexpected vs expected C4/D4: $P < 10^{-4}$, Hierarchical bootstrapping test with Bonferroni correction). 7 sessions from 5 mice. Bars and error bars indicate mean and 95% bootstrap CI. **f**, Responses to expected B4 (left), unexpected C4 or D4 (middle) and expected C4 or D4 (right) stimuli with (amber) and without (black) pulvinar stimulation. $n = 379$ cells from 7 sessions, 5 mice, LED on vs off during expected B4 stimulus: $P < 1 \times 10^{-4}$; LED on vs off during unexpected C4 or D4 stimulus, $P = 4.0 \times 10^{-4}$; LED on vs off during expected C4 or D4 stimulus, $P = 0.48$; Hierarchical bootstrapping test. Bars and error bars are mean and 95% bootstrap CI. **g**, Coronal slice through the pulvinar injection site (LP, right) and through the laterodorsal nucleus of thalamus (LD, left), showing specific expression of ChrimsonR-tdTomato (red) in LP, not in LD. Scale bars: 100 μm. **h**, Same as Fig. 5d, but plotted for individual mice. Visual stimulus responses of individual SOM neurons to expected B4 stimulus (left), unexpected C4 or D4 stimulus (middle, in block 1) and expected C4 or D4 stimulus (right, in late block 2) plotted against the effect of pulvinar stimulation (difference in strength of visual stimulus responses with and without optogenetic pulvinar axon stimulation) for recruited (brown) and other (black) SOM cells.

# Reporting Summary

Please do not complete any field with "not applicable" or n/a. Refer to the help text for what text to use if an item is not relevant to your study.
For final submission: please carefully check your responses for accuracy; you will not be able to make changes later.

## Statistics

For all statistical analyses, confirm that the following items are present in the figure legend, table legend, main text, or Methods section.

| n/a | Confirmed | |
|---|---|---|
| ☐ | ☑ | The exact sample size (*n*) for each experimental group/condition, given as a discrete number and unit of measurement |
| ☐ | ☑ | A statement on whether measurements were taken from distinct samples or whether the same sample was measured repeatedly |
| ☐ | ☑ | The statistical test(s) used AND whether they are one- or two-sided *Only common tests should be described solely by name; describe more complex techniques in the Methods section.* |
| ☐ | ☑ | A description of all covariates tested |
| ☐ | ☑ | A description of any assumptions or corrections, such as tests of normality and adjustment for multiple comparisons |
| ☐ | ☑ | A full description of the statistical parameters including central tendency (e.g. means) or other basic estimates (e.g. regression coefficient) AND variation (e.g. standard deviation) or associated estimates of uncertainty (e.g. confidence intervals) |
| ☐ | ☑ | For null hypothesis testing, the test statistic (e.g. *F*, *t*, *r*) with confidence intervals, effect sizes, degrees of freedom and *P* value noted *Give P values as exact values whenever suitable.* |
| ☑ | ☐ | For Bayesian analysis, information on the choice of priors and Markov chain Monte Carlo settings |
| ☐ | ☑ | For hierarchical and complex designs, identification of the appropriate level for tests and full reporting of outcomes |
| ☐ | ☑ | Estimates of effect sizes (e.g. Cohen's *d*, Pearson's *r*), indicating how they were calculated |

*Our web collection on statistics for biologists contains articles on many of the points above.*

## Software and code

Policy information about availability of computer code

Data collection: ScanImage, Labview

Data analysis: Matlab 2018b, 2021a

For manuscripts utilizing custom algorithms or software that are central to the research but not yet described in published literature, software must be made available to editors and reviewers. We strongly encourage code deposition in a community repository (e.g. GitHub). See the Nature Portfolio guidelines for submitting code & software for further information.

## Data

Policy information about availability of data

All manuscripts must include a data availability statement. This statement should provide the following information, where applicable:
- Accession codes, unique identifiers, or web links for publicly available datasets
- A description of any restrictions on data availability
- For clinical datasets or third party data, please ensure that the statement adheres to our policy

The data that support the main findings of this study are publicly available at https://doi.org/10.5281/zenodo.11403111. Other data that are generated in this study are available from the corresponding author upon reasonable request. Source data are provided with this paper.

## Research involving human participants, their data, or biological material

Policy information about studies with [human participants or human data](). See also policy information about [sex, gender (identity/presentation), and sexual orientation]() and [race, ethnicity and racism]().

| | |
|---|---|
| Reporting on sex and gender | Not applicable |
| Reporting on race, ethnicity, or other socially relevant groupings | Not applicable |
| Population characteristics | Not applicable |
| Recruitment | Not applicable |
| Ethics oversight | Not applicable |

Note that full information on the approval of the study protocol must also be provided in the manuscript.

## Field-specific reporting

Please select the one below that is the best fit for your research. If you are not sure, read the appropriate sections before making your selection.

☑ Life sciences ☐ Behavioural & social sciences ☐ Ecological, evolutionary & environmental sciences

For a reference copy of the document with all sections, see [nature.com/documents/nr-reporting-summary-flat.pdf]()

## Life sciences study design

All studies must disclose on these points even when the disclosure is negative.

| | |
|---|---|
| Sample size | Matches standards in the field (Voitov et al, Nature, 2022, Kanamori and Mrsic-Flogel, Neuron, 2022) |
| Data exclusions | Data excluded if imaging quality insufficient or expression of constructs failed or in wrong location |
| Replication | Main results contain multiple data sets in which findings could be replicated (see numbers of animals for biological replicates) |
| Randomization | In our study, stimulus presentation and optogenetic manipulation were randomized by software. |
| Blinding | In our study, stimulus presentation and optogenetic manipulation were randomized by software. |

## Behavioural & social sciences study design

All studies must disclose on these points even when the disclosure is negative.

| | |
|---|---|
| Study description | |
| Research sample | |
| Sampling strategy | |
| Data collection | |
| Timing | |
| Data exclusions | |
| Non-participation | |
| Randomization | |

# Ecological, evolutionary & environmental sciences study design

All studies must disclose on these points even when the disclosure is negative.

| | |
|---|---|
| Study description | |
| Research sample | |
| Sampling strategy | |
| Data collection | |
| Timing and spatial scale | |
| Data exclusions | |
| Reproducibility | |
| Randomization | |
| Blinding | |

Did the study involve field work?  ☐ Yes   ☐ No

## Field work, collection and transport

| | |
|---|---|
| Field conditions | |
| Location | |
| Access & import/export | |
| Disturbance | |

# Reporting for specific materials, systems and methods

We require information from authors about some types of materials, experimental systems and methods used in many studies. Here, indicate whether each material, system or method listed is relevant to your study. If you are not sure if a list item applies to your research, read the appropriate section before selecting a response.

### Materials & experimental systems

| n/a | Involved in the study |
|---|---|
| ☑ | ☐ Antibodies |
| ☑ | ☐ Eukaryotic cell lines |
| ☑ | ☐ Palaeontology and archaeology |
| ☐ | ☑ Animals and other organisms |
| ☑ | ☐ Clinical data |
| ☑ | ☐ Dual use research of concern |
| ☑ | ☐ Plants |

### Methods

| n/a | Involved in the study |
|---|---|
| ☑ | ☐ ChIP-seq |
| ☑ | ☐ Flow cytometry |
| ☑ | ☐ MRI-based neuroimaging |

## Antibodies

| | |
|---|---|
| Antibodies used | |
| Validation | |

# Eukaryotic cell lines

Policy information about cell lines and Sex and Gender in Research

| | |
|---|---|
| Cell line source(s) | |
| Authentication | |
| Mycoplasma contamination | |
| Commonly misidentified lines<br>(See ICLAC register) | |

# Palaeontology and Archaeology

| | |
|---|---|
| Specimen provenance | |
| Specimen deposition | |
| Dating methods | |

☐ Tick this box to confirm that the raw and calibrated dates are available in the paper or in Supplementary Information.

| | |
|---|---|
| Ethics oversight | |

Note that full information on the approval of the study protocol must also be provided in the manuscript.

# Animals and other research organisms

Policy information about studies involving animals; ARRIVE guidelines recommended for reporting animal research, and Sex and Gender in Research

| | |
|---|---|
| Laboratory animals | Mice. See manuscript methods for mouse lines used |
| Wild animals | No wild animals were used in the study |
| Reporting on sex | Both male and female were used in the study |
| Field-collected samples | No field collected samples were used in the study |
| Ethics oversight | This study was approved by institutional ethics and UK Home Office |

Note that full information on the approval of the study protocol must also be provided in the manuscript.

# Clinical data

Policy information about clinical studies
All manuscripts should comply with the ICMJE guidelines for publication of clinical research and a completed CONSORT checklist must be included with all submissions.

| | |
|---|---|
| Clinical trial registration | |
| Study protocol | |
| Data collection | |
| Outcomes | |

# Dual use research of concern

Policy information about dual use research of concern

## Hazards

Could the accidental, deliberate or reckless misuse of agents or technologies generated in the work, or the application of information presented in the manuscript, pose a threat to:

| No | Yes | |
|----|-----|---|
| ☐ | ☐ | Public health |
| ☐ | ☐ | National security |
| ☐ | ☐ | Crops and/or livestock |
| ☐ | ☐ | Ecosystems |
| ☐ | ☐ | Any other significant area |

## Experiments of concern

Does the work involve any of these experiments of concern:

| No | Yes | |
|----|-----|---|
| ☐ | ☐ | Demonstrate how to render a vaccine ineffective |
| ☐ | ☐ | Confer resistance to therapeutically useful antibiotics or antiviral agents |
| ☐ | ☐ | Enhance the virulence of a pathogen or render a nonpathogen virulent |
| ☐ | ☐ | Increase transmissibility of a pathogen |
| ☐ | ☐ | Alter the host range of a pathogen |
| ☐ | ☐ | Enable evasion of diagnostic/detection modalities |
| ☐ | ☐ | Enable the weaponization of a biological agent or toxin |
| ☐ | ☐ | Any other potentially harmful combination of experiments and agents |

# Plants

| Seed stocks | |
|---|---|
| Novel plant genotypes | |
| Authentication | |

# ChIP-seq

## Data deposition

☐ Confirm that both raw and final processed data have been deposited in a public database such as GEO.

☐ Confirm that you have deposited or provided access to graph files (e.g. BED files) for the called peaks.

| Data access links
*May remain private before publication.* | |
|---|---|
| Files in database submission | |
| Genome browser session
(e.g. UCSC) | |

## Methodology

| Replicates | |
|---|---|
| Sequencing depth | |
| Antibodies | |
| Peak calling parameters | |
| Data quality | |

| Software | |
|---|---|

# Flow Cytometry

## Plots

Confirm that:

☐ The axis labels state the marker and fluorochrome used (e.g. CD4-FITC).

☐ The axis scales are clearly visible. Include numbers along axes only for bottom left plot of group (a 'group' is an analysis of identical markers).

☐ All plots are contour plots with outliers or pseudocolor plots.

☐ A numerical value for number of cells or percentage (with statistics) is provided.

## Methodology

| Sample preparation | |
|---|---|
| Instrument | |
| Software | |
| Cell population abundance | |
| Gating strategy | |

☐ Tick this box to confirm that a figure exemplifying the gating strategy is provided in the Supplementary Information.

# Magnetic resonance imaging

## Experimental design

| Design type | |
|---|---|
| Design specifications | |
| Behavioral performance measures | |

| Imaging type(s) | |
|---|---|
| Field strength | |
| Sequence & imaging parameters | |
| Area of acquisition | |

Diffusion MRI      ☐ Used      ☐ Not used

## Preprocessing

| Preprocessing software | |
|---|---|
| Normalization | |
| Normalization template | |
| Noise and artifact removal | |
| Volume censoring | |

## Statistical modeling & inference

| Model type and settings | |
|---|---|
| Effect(s) tested | |

Specify type of analysis: ☐ Whole brain ☐ ROI-based ☐ Both

Statistic type for inference

[                                                                ]

(See Eklund et al. 2016)

Correction

[                                                                ]

## Models & analysis

| n/a | Involved in the study |
|-----|----------------------|
| ☐ | ☐ Functional and/or effective connectivity |
| ☐ | ☐ Graph analysis |
| ☐ | ☐ Multivariate modeling or predictive analysis |

Functional and/or effective connectivity

[                                                                ]

Graph analysis

[                                                                ]

Multivariate modeling and predictive analysis

[                                                                ]

