## [Peer Review File · Nature]

Manuscript Title: Cooperative thalamocortical circuit mechanism for sensory prediction errors

Reviewer Comments & Author Rebuttals

Reviewer Reports on the Initial Version:

Referees' comments:

Referee #1 (Remarks to the Author):

Furutachi and colleagues investigate prediction error signals in awake, head-fixed mouse primary visual cortex (V1) using a virtual one-dimensional tunnel. At specific locations of the corridor, specific gratings are presented. After familiarizing the mice with the virtual tunnel and the sequence of gratings, individual gratings are replaced causing sensory prediction errors. Based on this paradigm, the authors make several important and exciting claims:

1. Violations of the mice's predictions cause a specific boost in those neurons that are most selective to the unexpected grating.
2. This boost is an amplification of the unexpected visual input rather than a general surprise signal or difference signal.
3. This boost relies on a cooperative effect of pulvinar input to V1 and a local disinhibitory circuit within V1.

4. This thalamocortical disinhibitory circuit is required for the generation of sensory prediction errors. Taken together, these results give us an interesting view of how sensory prediction errors arising from spatial predictions of visual input may be implemented. Investigating the neuronal circuits involved in predictive processing is a timely topic and this manuscript tackles this challenge in multiple ways. However, some of the major claims are not entirely backed up by their data. Additionally, the authors state that the described thalamocortical disinhibitory circuit is required for the generation of sensory prediction errors. However, several such pathways have been described previously, including higher visual areas to V1 (Rao and Ballard, 1999), ACC to V1 (Leinweber et al. 2017), and also LP to V1 (Roth et al. 2016). How their results link to other circuits that contribute to the generation of prediction errors remains unclear. In conclusion, there are numerous issues that leave the main findings in some doubt, so the authors have some work to do before I would find this story compelling.

Major points:

1. The authors claim that predictions cause a specific boost in those neurons that are most selective to the unexpected grating (Fig. 2d). This is a strong statement which partially goes against current views of predictive processing (Keller and Mrsic-Flogel 2018). In "thought experiment 1" (Fig. 1e), I would have expected positive C-prediction neurons, and negative A and B-prediction neurons to strongly respond to an unexpected C stimulus. However, the authors argue that only positive C-prediction neurons (i.e. neurons tuned to the visual stimulus C) increase their responses, particularly those with a high selectivity (Fig. 2d). The data, as they are presented now, do not entirely back up this claim.

1.1 The authors distinguish between “C4 unexpected” and “C4 expected”. However, there are several plots that suggest that the mice might not be expecting C4 for “C4 expected” and might still be learning the rule change. This, however, would make the results harder to interpret (e.g. silencing results in Fig. 3f or selectivity in Fig. 2d, see also next point on negative prediction error neurons).

1.1.1 In Ext. Data Fig. 1a, right, the authors argue that mice reduce their running speed for unexpected C4 trials (red trace) compared to expected B4 trials (black trace; see also lines 29-32). However, mice also slow down for expected C4 trials.

1.1.2 In Ext. Data Fig. 1d, the pupil size is larger than in other conditions.

1.1.3 In Fig. 3f, right, silencing VIP neurons also reduces responses to the expected C4 or D4 stimuli.

1.1.4 In Ext. Data Fig. 2, the responses to expected C4 for all grating responsive neurons is higher than those to expected A1, B2, or A3.

1.2 Given that the response to “C4 expected” does not seem to be entirely expected (see point above), negative A or B-prediction neurons could falsely be classified as “C selective”, which would not be consistent with the authors claim.

1.3 In Ext. Data Fig 2b, bottom (“grating responsive cells”), why are the responses to C4 expected lower than the responses to A1, B2 or A3 expected? Could this again suggest that some of these neurons are indeed negative A or B-prediction neurons that do not strongly respond to C (nor to all other expected visual stimuli)?

1.4 The authors should further show that “tuned otherwise” (A1, B2, A3, or B4) or “untuned” (all but A1, B2, A3, B4, and C4 expected) neurons do not show the increase to C4 unexpected.

1.5 In general, the authors should tie their results more into the literature.

1.5.1 How are their results related to positive and negative prediction errors?

1.5.2 The response profile of omission neurons to stimuli A1, B2, and A3 in Ext. Data Fig. 2f look like “predictive neurons” in Fiser et al. 2016 (see pre-stimulus ramp up), all other neurons look like “visual neurons” in that paper. Are those results consistent with the ones presented here?

1.6 Lines 99-104 should be clarified.

1.6.1 The authors state that “prediction errors have been proposed to encode the difference between predicted and actual visual input”. Do they mean on a single-cell level or on a population level?

1.6.2 Further the authors state that “error responses could also represent a more unspecific surprise signal (encoding only the magnitude of the deviation)”. Are they referring to an unsigned error signal?

1.7 Lines 114-116 should further specify that this statement is based on grating responsive cells.

1.8 Lines 161-163 seems overstated – this would mean that there are no negative prediction error neurons in layer 2/3 of V1.

2. The thought experiment 2 (Fig. 1j) distinguishes between two scenarios – neurons coding for an unspecific surprise signal or an amplified sensory response. However, there are several points that remain puzzling.

2.1 Stimuli C and D are radically different stimuli recruiting different neurons in the cortex (prediction error so far in orientation space). Can the results be explained by the different nature of the stimuli? The authors should demonstrate that this effect holds true when using for example a horizontal grating as stimulus D.

2.2 Assuming an unspecific surprise signal in addition to tuned responses (Fig. 1j, left option), I would

expect a noisy version of Fig. 1j, right, rather than a diagonal. Perhaps the authors mislabeled the axis and it should say “unexpected C – expected C” and “unexpected D – expected D” for x and y-axis, respectively, as in Fig. 1n.

2.3 VIP responses in Ext. Data Fig. 5d resemble an “unspecific surprise signal”. The authors may consider highlighting this fact since it highlights the special role VIP neurons play in this computation.

2.4 Ext. Data Fig. 5, 7 for VIP neurons and pulvinar inputs, respectively, are not plotted the same way. The authors should reanalyze the data as for excitatory neurons in Fig. 1n, 2d,e.

2.5 Is the distribution in Fig. 1n random? In other words, is the density of neurons in the top right quadrant (z value > 0.5 for both C4 and D4) lower than expected by chance?

3. The responses to C4 and D4 unexpected (Fig. 1c,l) appear to be significantly larger than those to C2 and C3 unexpected (Fig. 1g). Is this pointing to a potential confounder due to upcoming reward signaling? Pakan et al. 2018 show that activity prior to the reward starts ramping up largely before the reward.

3.1 The authors should add an analysis excluding neurons that have responses associated with the reward.

3.2 The authors should show B4 in Fig. 1c and 2b (ideally from block 1 and from late training sessions).

4. In the current manuscript, the data do not clearly “...demonstrate that the recruitment of VIP interneurons is required for the generation of prediction error signals...” (lines 196, 197).

4.1 The results are equally consistent with a general reduction in gain, independent of a contribution to the computation of a prediction error signal.

4.2 The authors should also plot “unexpected C4 – expected C4, both LED on” vs “unexpected C4 – expected C4, both LED off”.

4.3 What do the authors conclude from the significant effect of VIP silencing in Figure 3f, right?

4.4 How well are the authors silencing VIP cells? They should add a plot of the activity of VIP during silencing.

4.4 In Ext. Data Fig. 5j,k, why can the VIP not be visualized (line 961) despite the fact that they express mCherry?

4.5 The authors pool responses to C4 and D4, even though the stimuli evoke very different responses in VIP neurons and potentially pulvinar inputs. Do all the results in Fig. 3 hold for C4 or D4 stimuli separately?

5. The authors causally demonstrate the involvement of pulvinar circuits in generating prediction error signals. There are some aspects which leave this statement in some doubt.

5.1 The authors express eNpHR3.0 in the pulvinar and optogenetically silence their axons in V1 (Fig. 3l). Are they actually silencing the axons with their manipulation?

5.1.1 The authors could either record in axons during optogenetic silencing of the same axons, or

5.1.2 Implant a fiber in the pulvinar and silence cell bodies in a few animals to confirm the effectiveness of their manipulation.

5.2 As for the VIP manipulation (see major point above), the results presented in Fig. 3m could also be a result of a general effect on the gain.

5.3 As above, the authors should plot “unexpected C4 – expected C4, both LED on” vs “unexpected C4 – expected C4, both LED off”.

6. It is surprising that pulvinar input and VIP neurons act synergistically (Fig. 4d) although the individual manipulations either show no effect (Fig. 4c) or even the opposite (Fig. 4b, for highly selective neurons). This result would be more convincing if the authors show the individual and combined manipulations in the same mice (using soma-targeted LP expression and implanting a fiber). This would further allow to see if the two mechanisms act on the same neurons.

7. The authors target the pulvinar (or LP) for viral expression of opsins and calcium indicators. LP, however, is adjacent to LD and dLGN, which have different functions and projection patterns targeting specific neuronal subtypes (Ma et al. 2021). Based on their Fig. S13 (Ma et al. 2021), LD seems to mainly target SOM neurons while LP mainly targets VIP neurons. Hence, it is crucial that the injections in the current manuscript do not leak into other thalamic nuclei. For this, the authors should present representative images of the injection site in LP for all animals individually together with the corresponding effect magnitude.

Minor points:

- Redo all stats with hierarchical bootstrap sampling (Saravanan et al. 2020) or similar.
- For several plots (e.g. Fig. 2d) the effects is seen only in few cells. To ensure that the effects are consistent across animals, the authors should plot all major effects for all mice individually.
- Line 30/Ext. Data Fig 1: The authors should also show running traces during training.
- Line 79/Fig. 1d: Did the response to B4 change when presenting C4 unexpected? Please show how the response to B4 evolves during Block 1.
- Line 83: Please rephrase – the pupil size was different for expected C4 in Ext. Data Fig. 1d. Why do the authors think that the pupils were dilated in this condition? Could this be because the mouse is still learning/updating? Comparing these data points to the pupil size during learning might explain the difference.
- Fig. 1a: Was A always oblique and B always vertical or did the authors correct for this asymmetry in their paradigm?
- The authors use z-scored dFF throughout the manuscript. While this facilitates comparison across sessions and mice, it biases the analysis towards neurons with low baseline activity. Do the main results hold with dFF?
- Fig. 1d: For consistency with Fig. 3d,k, please show the mean of A1, B2, A3 rather than just A3.
- Fig. 1e,j: It has to be very clear that these are thought experiments and should not have plots that resemble real data (individual dots with noise).
- Fig. 1g: Scale missing.
- Line 92: For clarity, “highly selective” should be in the text.
- Line 93: For clarity, “for omission responsive neurons” should be in the sentence but then the statement becomes trivial. Please rephrase.
- Fig. 2e, 4b-e: Some of the “non-selective” neurons are highly selective for the other stimulus. This should be divided into 3 categories with the highly-selective neurons of the other stimulus in the Ext.

Data Figs.

- Fig. 3f,m, 5c: Add LED-on time as in Fig. 4.
- Fig 5: How do the authors explain the small responses of SOM neurons to full-field stimuli (Adesnik et al. 2012)?
- Ext. Data Fig. 5e,i: The legends state “polar plot” instead of “pie chart”.
- Ext. Data Fig. 9d,e: For consistency with Fig. 5, these plots should be broken down into “recruited cells” and “other cells”.
- Citation 21 has no date.

Referee #2 (Remarks to the Author):

In this paper, Furutachi et al., investigate how circuits of the visual cortex and higher order visual thalamus give rise to the observation of strengthened neuronal responses following unexpected stimuli or events, also referred to as mismatch or prediction error responses e.g. in the context of predictive coding. The authors discovered that the enhanced neuronal response to an unexpected stimulus does not merely reflect a generalized mismatch signal, nor does it reflect how the unexpected stimulus deviates from expectation (as suggested in predictive coding theory), but the enhanced response specifically encodes information of that particular stimulus. The main result of this paper is the unraveling of a novel synergistic circuit in which projections from the pulvinar modulate the interaction of local vasoactive-intestinal-peptide (VIP) expressing and somatostatin (SOM) expressing interneurons, thereby gating/amplifying stimulus specific responses in excitatory visual cortex neurons.

This is an important and well executed study. Firstly, it identifies several key components of the circuit that drives the amplification of neuronal responses to unexpected stimuli in primary visual cortex. While these findings are not completely in disagreement with predictive coding theory, i.e. they show that prediction errors are an important signal in visual coding, the study demonstrates in a series of very clear and elegant experiments how early visual processing reflects the sensory aspect of visual information, rather than the deviation from the expected input. Beyond implications for predictive coding theory, these insights are an important step to a better understanding of the computations and functions of the neocortex. Secondly, the authors elegantly demonstrate the (very likely general) principle of how neuronal activity can depend on multiple, often synergistic pathways, which might be misunderstood (or their complexity might be underestimated) when studied in isolation. In conclusion, I am very enthusiastic about this work. Nevertheless, I do have number of concerns regarding methodology and the interpretation of the findings that need to be addressed.

Major:

1. Important to this study is the concept of an internal model. While the internal model cannot be directly observed, violations of the model’s predictions are thought to change behavior. The authors also report a change in behavior to the unexpected stimulus C4, in block 1, arguing that this indeed shows that mice did not expect the novel stimulus. However, the behavior of the mice does not show a

difference between the period when the novel stimulus is unexpected (C4 block 1) and when it is expected (C4 block 2). This is at odds with the interpretation of the neuronal data, in which stimulus C4 in block 2 is supposedly expected, presumably following a learning process that updates the internal model. Can the authors provide behavioral evidence of their assumption that mice learned to expect stimulus C4 in block 2?

Related to this, ED Fig 1h shows that the mouse's running speed was not correlated to the response strength to grating C. However, this appears to be only for block 2. It is not clear for which data plot ED Fig 1i is. The relationship between running speed and response strength should also be investigated for block 1 (when C4 was newly introduced).

2. In the introduction, the authors propose a line of reasoning in which the internal model provides a prediction, which when violated leads to increased V1 responses and updating of the model. The authors suggest that the observed increased sensory responses are instructive in updating the model. If that is the case, the updating of the model should be affected when activity is suppressed (e.g. as in some of the optogenetic experiments). Can the authors confirm this? And related to this, some mice might learn to expect the stimulus faster than others, is this correlated with the neuronal effects?

3. The authors treat single neurons as independent observations. This makes sense in some cases, e.g. when the focus lies on how specific neurons change response properties across conditions or over time. However, in the case of studying effects that are expressed broadly across circuits spanning many neurons (such as the violation of a prediction of the internal model), the correlations/dependencies between neurons within a single biological replicate could artificially overpower the study. While I understand that not every effect can be tested across groups of animals (in a study as complex as this), the authors should convincingly show that the main results hold across -real- biological replicates. Related to this, in scatter plot figures like Fig 1h or 1i, it is unclear whether the z-scored dF is also normalized per mouse. If this was not done, systematic differences in group means across animals could result in spurious correlations. There should be (at least) a control for this.

4. I have a hard time following some of the reasoning for using specific analysis approaches and statistical tests. This should be streamlined. For instance, why do the authors report both the Pearson and the Spearman correlation coefficients for a single scatter plot? Why did the authors use a longer time window for investigating neuronal responses in Fig 5 and ED Fig 9? Why do the authors use two different methods for calculating selectivity, arguing that one of the methods is less sensitive to noise than the other; would it not be better to use the best/most appropriate method throughout? Why do the authors occasionally show 95% confidence intervals (e.g. ED Fig 3b,c) and in other plots use a Wilcoxon signed-rank test (e.g. ED Fig 6b) to compare group wise differences?

5. The conclusion of the paper is that the responses of the most stimulus-selective neurons are amplified. How does this rhyme with that there are many more cells recruited (significantly responsive) for the new grating (C4)? Following the reasoning of amplifying only highly stimulus-selective neurons, I would have expected the number of responsive cells for C4 to be the same as for e.g. A3, with the C4 responsive neurons having larger response amplitudes. This requires a proper quantification and

explanation.

Minor:

1. In Fig 1a. the grey parts in the 'stimulus timeline' suggest that the corridor had large sections of uniform gray wall, which I assume was not the case?
2. Something that puzzles me is that mice typically do ~90 traversals a day and then on the C session do 200 traversals. How come the mice do more trials? And related, why is the pupil size overall larger in late sessions? Could this reflect some change in overall internal state in block 2? Could this affect the interpretation of the results?
3. In Fig 1m and 1n, there are a couple of neurons on the diagonal. Does this mean that there might be some, albeit few, V1 neurons reliably coding for unspecific surprise?
4. ED Fig 2 demonstrates that showing the same grating in different positions leads to slightly different responses. Considering the study of Saleem et al. (2018), I had expected these differences to be larger. Is there an explanation for observing smaller differences here?
5. The authors state that neurons with a selectivity index < 0.6 are all unselective (e.g. Fig 2d). However, a large fraction of these neurons has strongly negative selectivity indices. In my opinion, these are selective as well, just in the opposite direction. Can the authors explain why they do not observe 'inverse' effects for this subgroup?
6. Neurons appear to find the 'D' stimulus more unexpected than the 'C' stimulus. Do the animals agree? And does this reveal something about the involved circuits?
7. In Fig 3i, a number of axons show negative responses, how should this be interpreted?
8. Do the authors propose that there are two types of SOM neurons, those recruited by pulvinar and part of a general prediction error circuit, and others that are not? Or do the authors explain their observations by subsets of SOM neurons being part of different stimulus-specific circuits?
9. In Fig 5d, some 'recruited cells' have a negative response to gratings, and to 'led on/off'. How exactly would these be considered 'recruited'?
10. The authors use a Gaussian mixture model to estimate F_0 . I can see advantages of this approach, but is there any evidence that it is better than established methods? Related to this, do the authors correct for a potential slow drift in F_0 values over the course of the experiment?
11. The authors should explain better what they mean with "signal contribution" and how the threshold prevents this? (see e.g. line 645)
12. In ED Fig 5f I would like to see the VIP response to expected B4 in the plot as well.

Typos:

- Fig 5e: 'n' is not stated (neither in figure, nor in legend)
- Line 853: The bold 'd' should be 'f'
- ED Fig 5e,i: These are -not- polar plots
- Line 952: "Difference" should not be capitalized.
- Line 1031: "ontogenetic" should be "optogenetic"

Referee #3 (Remarks to the Author):

In this manuscript, Furutachi and colleagues report that prediction errors in mouse V1 amplify layer 2/3 neuronal responses to unexpected stimuli, rather than a surprise signal or a computation of the difference between expected and actual visual input. They describe a circuit mechanism for this computation: inputs from the pulvinar nucleus enhance visual responses to unexpected stimuli while simultaneously inducing inhibition in V1 through the activation of a subset of somatostatin interneurons. This circuit interacts cooperatively with VIP interneuron activation by unexpected stimuli, leading to the inhibition of somatostatin interneurons that gate excitatory pulvinar input to V1. This results in the amplification of V1 responses to unexpected stimuli.

The experiments presented in the article are well-executed and described in great details. The article includes datasets of high quality that will be valuable for the circuit neuroscience community. The question of the neuronal circuits underlying predictive processing of sensory information is of high relevance for this field of research. This article brings a valuable contribution towards this endeavour consistent with recent articles in the field (articles for G. Keller's group; Fang et al., *Neuron* 2020; Hu et al., *Neuron* 2019).

One major challenge to test prediction errors is to experimentally constrain predictions. My major concern with this manuscript is the unclear distinction between (1) an unexpected stimulus in the sense of a known stimulus but presented at an unexpected location (ext. fig. 3) and (2) a complete novel stimulus (all main figures). Both are unexpected stimuli but the results presented for case (1) (ext. fig. 3) only show a very small effect on neuronal responses in V1 and the underlying mechanisms are unexplored. It is thus unclear whether the same mechanisms underlie these two types of unexpected stimuli. This distinction needs strong clarification in order to support the strong, general conclusion about neuronal circuits underlying sensory prediction errors.

Major issues:

- 1. The data shown in the manuscript indicates that the enhanced responses to unexpected stimuli depend on both expectation and novelty, and mostly on the latter. This is clear when comparing ext. fig. 3c and 3e and Fig 2b, 2e (it would be informative to have the same y-axis for panels e in both figures). While neuronal responses to a familiar stimulus A (in unexpected position 4) are slightly larger than those to the same familiar stimulus A at an expected position (position 3, Ext. Fig 3c,e), this difference is much smaller than that observed when comparing responses to an unexpected novel stimulus at an unexpected location vs an expected familiar stimulus (stimulus C in position 4 in early and late trials (Fig 1c, d)). This result indicates that most of the change observed in these experiments relates to novelty and not to violated expectations. To directly test this, authors should compare the enhancement of responses induced by expectation (A3 vs A4) with that induced by novelty + expectation (unexpected C4 vs expected C4). This comparison together with data shown in Ext. Fig3 should be presented in the main figure. In order to support general conclusions about prediction error responses, the authors should investigate the mechanisms underlying the difference observed between expected and unexpected familiar stimuli (A3 vs A4; Ext Fig 3), as shown for novel stimuli in figures 2, 3 & 5.
- 2. If prediction error responses are an amplification of sensory responses, there should be a strong correlation between individual neuron responses to expected and unexpected stimulus C. Including this quantification (response to unexpected versus expected, by modifying fig. 2c) would strengthen this

point.

- 3. It would be useful to compare the responses to unexpected stimulus during suppression of VIP cells versus the responses to the same stimulus when it is expected. The potential difference in responses will indicate the relative contribution of VIP neurons to prediction error responses. Same point for the suppression of pulvinar inputs.
- 4. Most statistical tests presented in the manuscript are done pulling neurons from different animals to increase statistical power. The statistics for right column's histograms should be done per animal (and the authors should show the distribution of each average value per animal). (e.g. it would be interesting to see whether it changes results in Fig 4c).

Minor issues:

- In lines 29 to 31 authors indicate that "mice interrupted their running behaviour when their expectations were violated by encountering grating C". This creates a difference between locomotion behaviour in the unexpected grating C compared with other gratings. Surprisingly, Ext. Fig 1 shows that animals specifically interrupted running when encountering grating C, irrespectively of whether this stimulus is expected or unexpected (Ext. Fig1b). Is the reduction of running speed really related to violated expectation? Showing running speed for block1, 2 early/late would be informative. Other point: Is running speed affected by the light used for optogenetic stimulation?
- In Fig3 responses to expected C and expected D stimuli are pooled together. Ext. Fig5 shows responses to each stimulus separately. A similar figure would be useful for the responses of pulvinar inputs.
- Authors refer to pulvinar nucleus/ pulvinar inputs, that is usually used for the primate higher order thalamic nucleus, while the murine homologous is called lateral posterior nucleus. This could be clarified in the manuscript.

Author Rebuttals to Initial Comments:

Response to the reviewers

We thank the reviewers for their feedback and the thoughtful and constructive comments. We have carried out a large number of new experiments and analyses to address the reviewers' comments. Major additions include:

- New experiments of two-photon calcium imaging of V1 neuronal activity combined with optogenetic silencing of VIP neurons in behaving mice encountering *familiar* visual stimuli at an unexpected location in the virtual corridor (Extended Data Fig. 11e,f).
- New experiments of two-photon calcium imaging of V1 neuron activity combined with optogenetic silencing of pulvinar inputs to V1 in behaving mice encountering *familiar* visual stimuli at an unexpected location in the virtual corridor (Extended Data Fig. 11j,k).
- New experiments of two-photon calcium imaging of V1 VIP neurons and pulvinar axons in V1 to quantify their responses to familiar visual stimuli at unexpected locations (Extended Data Fig. 11a-d, g-i).
- New experiments of two-photon calcium imaging of VIP neurons in V1 while they are optogenetically silenced to quantify the effect sizes of this manipulation (Extended Data Fig. 12a-c).
- New experiments of two-photon calcium imaging of pulvinar axons in V1 while they are optogenetically silenced to quantify the effect size of this manipulation (Extended Data Fig. 12d-f).
- New experiments of two-photon calcium imaging of V1 neurons in mice encountering either the omission of a visual stimuli or an unexpected novel stimulus at the same virtual corridor location to compare the response of individual neurons to negative and positive prediction errors (Extended Data Fig. 9).
- New experiments of two-photon calcium imaging of V1 neurons in mice encountering the same unexpected stimuli at various locations to further quantify prediction error responses (Extended Data Fig. 4).
- New experiments of two-photon calcium imaging of V1 neurons in mice encountering different unexpected stimuli at the same location to further confirm stimulus-specificity of prediction error responses (Extended Data Fig. 6l-p).
- Further analyses including optimization of statistical analyses.

Detailed point-by-point responses to the individual comments are listed below.

Referee #1:

Furutachi and colleagues investigate prediction error signals in awake, head-fixed mouse primary visual cortex (V1) using a virtual one-dimensional tunnel. At specific locations of the corridor, specific gratings are presented. After familiarizing the mice with the virtual tunnel and the sequence of gratings, individual gratings are replaced causing sensory prediction errors. Based on this paradigm, the authors make several important and exciting claims:

1. *Violations of the mice's predictions cause a specific boost in those neurons that are most selective to the unexpected grating.*
2. *This boost is an amplification of the unexpected visual input rather than a general surprise signal or difference signal.*
3. *This boost relies on a cooperative effect of pulvinar input to V1 and a local disinhibitory circuit within V1.*
4. *This thalamocortical disinhibitory circuit is required for the generation of sensory prediction errors.*

Taken together, these results give us an interesting view of how sensory prediction errors arising from spatial predictions of visual input may be implemented. Investigating the neuronal circuits involved in predictive processing is a timely topic and this manuscript tackles this challenge in multiple ways. However, some of the major claims are not entirely backed up by their data. Additionally, the authors state that the described thalamocortical disinhibitory circuit is required for the generation of sensory prediction errors. However, several such pathways have been described previously, including higher visual areas to V1 (Rao and Ballard, 1999), ACC to V1 (Leinweber et al. 2017), and also LP to V1 (Roth et al. 2016). How their results link to other circuits that contribute to the generation of prediction errors remains unclear.

We thank the reviewer for their encouraging remarks, their very thorough review and constructive criticism, and their interesting questions. We have conducted a large amount of new experiments and analysis that we hope will address the reviewer's concerns. We have also included more discussion of the different potential pathways for sensory prediction error computations from the previous literature in the manuscript.

In conclusion, there are numerous issues that leave the main findings in some doubt, so the authors have some work to do before I would find this story compelling.

Major points:

1. *The authors claim that predictions cause a specific boost in those neurons that are most selective to the unexpected grating (Fig. 2d). This is a strong statement which partially goes against current views of predictive processing (Keller and Mrsic-Flogel 2018). In "thought experiment 1" (Fig. 1e), I would have expected positive C-prediction neurons, and negative A and B-prediction neurons to strongly respond to an unexpected C stimulus. However, the authors argue that only positive C-prediction neurons (i.e. neurons tuned to the visual stimulus C) increase their responses, particularly those with a high selectivity (Fig. 2d). The data, as they are presented now, do not entirely back up this claim.*

1.1 *The authors distinguish between "C4 unexpected" and "C4 expected". However, there are several plots that suggest that the mice might not be expecting C4 for "C4 expected" and might still be learning the rule change. This, however, would make the results harder to interpret (e.g. silencing results in Fig. 3f or selectivity in Fig. 2d, see also next point on negative prediction error neurons).*

The reviewer is correct that, while prediction error responses dramatically decrease as the animal repeatedly encounters the same initially unexpected stimulus C, neuronal responses to stimulus C in the second half of block 2 are still slightly larger than to stimuli A and B. This is not surprising given that the animals have encountered stimuli A and B for many hundreds of trials over several days, but the novel stimuli C or D only for tens of trials. If we understand correctly, the reviewer argues that this means there is still a small prediction error response present in the second half of block 2 which may cause misclassification of some neuronal responses, and for instance affect our claim that only the visual responses of the most selective neurons are amplified by prediction errors. We believe we have several strong lines of evidence showing that this is not the case (see below). Most importantly, we also observe highly selective amplification of visual responses to a familiar stimulus at an unexpected location (Extended Data Fig. 5). In this case misclassification is not an issue as we use the response to the expected and familiar stimulus at the familiar location to calculate selectivity of neurons. Furthermore, in a large set of new experiments we could now also show that these prediction error responses to a familiar stimulus at an unexpected location also rely on VIP neuron and pulvinar activity, revealing a general mechanism for prediction-error induced response amplification in V1 (new Extended Data Fig. 11). We will discuss these and other evidence in more detail in response to the specific points below.

1.1.1 In Ext. Data Fig. 1a, right, the authors argue that mice reduce their running speed for unexpected C4 trials (red trace) compared to expected B4 trials (black trace; see also lines 29-32). However, mice also slow down for expected C4 trials.

We agree that while most animals slowed down in response to an unexpected stimulus, running speed appeared not to be a reliable indicator across all mice for how ‘expected’ this stimulus became after repeated exposure. We have now adapted the text (lines 29 - 32) to clarify this. However, please note that a subset of mice did gradually recover their running speed in response to stimulus C by the end of block 2 (Rebuttal Figure 1b). Importantly, V1 responses to the C stimulus were similar in the two groups with and without this recovery of running speed (Rebuttal Figure 1, bottom) and differences in running speed could not explain V1 prediction error responses (Extended Data Fig. 3).

Moreover, we performed a new set of experiments (with slightly different virtual corridor design, see Rebuttal Figure 2a) in which we doubled the length of block 2. In these experiments, all mice showed recovery of the running deceleration induced by an unexpected stimulus by the end of the experiment. Importantly, neural responses to stimulus C did not further decrease during the extended block 2 (Extended Data Fig. 2b, right) and were similar to the responses observed in previous experiments in which animals did not recover their running speed.

Rebuttal Figure 1.a-c, Top: average running speed aligned to onset of stimuli B or C, of all mice (a, all experiments with novel stimulus C (C sessions) without optogenetic manipulations), of mice showing recovery of the running deceleration caused by the unexpected stimulus C when C is repeatedly encountered (b), and the remaining mice without running speed recovery (c). Bottom: average neural responses in V1 of C-responsive neurons of the subsets of mice shown on top.

Rebuttal Figure 2. a, Experimental design with extended block 2. Note that grating stimuli presented at different locations differed from the design used previously. **b**, Left: average running speed aligned to stimulus onset for expected stimulus B and novel stimulus A in block 1, in trial 21 to 40 of block 2 (expected, equivalent to late block 2 trials in previous experiments, dark blue) and late extended block 2 (light blue). Right: average V1 calcium responses to novel stimulus A in block 1 (unexpected, red), in what was previously defined as 'late block 2' (dark blue) and late extended block 2 (light blue). Calcium signal block 2 late vs block 2 extended, $p = 0.49$, $n = 310$ neurons from 4 mice, Hierarchical Bootstrapping test.

1.1.2 In Ext. Data Fig. 1d, the pupil size is larger than in other conditions.

The pupil size is on average slightly increased in block 2 of imaging sessions with unexpected grating C, independent of stimulus presentation or predictability of stimuli. This change in pupil size is therefore likely due to a general state change of some of the animals over the session. Please note that now we have optimised the statistical analysis across the paper and corrected for multiple comparisons, this trend for increased pupil size is not significant (Extended Data Fig. 1d). No such increase of pupil size was apparent in imaging sessions with unexpected grating D (new Extended Data Fig. 1i,j), instead there was a trend for pupil size to slightly decrease over time in these experiments, showing that there was no consistent directional change in pupil size caused by the introduction of a new stimulus. Importantly, V1 neural responses to expected stimuli A1, B2 and A3 did not change over time but were stable and similar in block 1 and 2 (Figure 1c, Extended Data Fig. 2), showing that such subtle changes in pupil size do not affect responses in V1.

1.1.3 In Fig. 3f, right, silencing VIP neurons also reduces responses to the expected C4 or D4 stimuli.

We thank the reviewer for pointing this out. The reduction of V1 responses to stimuli C and D in block 2 during VIP neuron silencing (Fig.3f, right) is indeed interesting, because in imaging sessions without optogenetic manipulations VIP neuron activity is suppressed in response to the C stimulus in block 2 (Figure 3c, Rebuttal Figure 3, left). Therefore, optogenetic silencing of VIP neurons in this condition (in which they are already silent in control trials) should have little or no effect on the stimulus response of V1 neurons, as is observed for the response to expected stimuli A and B (Fig. 3f left). However, interestingly, we observed that only in sessions in which VIP neurons were silenced in 50% of trials, their response to the C stimulus was still increased in the second half of block 2 in control trials (Rebuttal Figure 3, right). We speculate that when VIP neurons were optogenetically silenced for some encounters of the unexpected stimulus (and therefore prediction error signals in V1 experimentally suppressed), this may delay the update of the internal model and therefore the cessation of the prediction error response. This delayed reduction of VIP responses in VIP silencing sessions likely explains why VIP silencing still has an effect on V1 responses to stimulus C in block 2 under these conditions (while under normal control conditions VIP neurons would not contribute to the response to C in late block 2 since they are not active (Rebuttal Figure 3, left)). While this is an interesting observation, obtaining stronger evidence for our theory that reducing V1 prediction error signals delays updating of predictions would necessitate major changes to our experimental protocols and a large amount of new data including recordings in multiple brain areas. While we are pursuing this question in the future, convincingly addressing it is beyond the scope of this manuscript.

Response of VIP neurons in control trials

Rebuttal Figure 3. Left: calcium responses of VIP neurons in V1 to expected and unexpected visual stimuli during experiments without optogenetic manipulation. Right: calcium responses of VIP neurons in V1 in control trials (without optogenetic manipulations, ‘LED off’ trials) during experimental sessions in which VIP neurons were silenced in the remainder of the trials (50%).

1.1.4 In Ext. Data Fig. 2, the responses to expected C4 for all grating responsive neurons is higher than those to expected A1, B2, or A3.

Indeed, the population response to stimulus C in the second half of block 2 is still slightly larger than to stimuli A and B, which is not surprising given that stimuli A and B are highly expected as they have been encountered over many hundreds of trials over several days while stimulus C has only been introduced in the current imaging session for few trials. However, the response amplitude to stimulus C has asymptoted by the end of block 2, and the number of neurons significantly responding to stimulus C in late block 2 is roughly comparable to those responding to A and B (162 C-responsive neurons vs 158, 125 and 147 A1-, B1-, and A2-responsive neurons). Moreover, V1 VIP neurons - which mediate prediction error signals in V1 - respond similarly to B and C in late block 2 (Figure 3c). These findings together with the data shown in response to the above points 1.1.1-1.1.3 indicate that prediction error responses have mostly ceased at this point, and differences in response strength between C in late block 2 and familiar A/B may be due to more long-term changes in the visual feedforward pathway or local circuits in V1 caused by continuous exposure to a visual stimulus over many days.

1.2 Given that the response to “C4 expected” does not seem to be entirely expected (see point above), negative A or B-prediction neurons could falsely be classified as “C selective”, which would not be consistent with the authors claim.

We thank the reviewer for this interesting question. We assume with ‘negative A or B-prediction neurons’ the reviewer refers to neurons that signal the absence of an expected stimulus A or B, rather than the presence of an unexpected stimulus? We expected such negative prediction error responses to form a component of V1 responses to a novel instead of an expected stimulus, but surprisingly we did not observe strong negative prediction error responses in this case. This was apparent from:

- 1) Neurons that signal the absence of stimulus B should respond similarly to unexpected stimulus C and D, since in both cases stimulus B was expected and is absent. However, we see very few such neurons (Fig. 1n and Extended Data Fig. 6a-e). This is also the case when comparing responses to more similar unexpected stimuli (see response to point 2.1 and new data in Extended Data Fig.6l-p).
- 2) Neurons that specifically respond to the absence of A or the absence of B should respond differently to the unexpected stimulus C2 (stimulus B was expected) versus unexpected stimulus C3 (stimulus A was expected). However V1 neurons respond very similarly to C2 and C3 (Fig. 1h,i).

We have now also performed new experiments and analysis to further investigate this important question:

We performed two-photon calcium imaging of the same V1 neurons in response to omission of stimulus B and to unexpected presentation of stimulus D instead of stimulus B. In both cases B is expected and not present. First, fewer neurons responded to the stimulus omission compared with the unexpected new stimulus, and none with very strong responses (new Extended Data Fig. 9). Importantly, neurons responsive to stimulus B omissions did on average not respond when stimulus D was unexpectedly presented instead of B and vice versa (Extended Data Fig. 9e-j). These results together with the evidence above show that negative prediction error responses (i.e. responses specific to the absence of a particular stimulus) are a negligible component of V1 responses to an unexpected stimulus under our experimental conditions.

These findings show that negative prediction error neurons are not misclassified as ‘C selective’ neurons in our experiments. However, this does not exclude that some positive prediction error neurons may be misclassified as ‘C selective’ if the animal still shows prediction error responses at the end of block 2, which could compromise our claim that part of the prediction error signal is a selective amplification of visual information. However, the data and arguments presented above (points under 1.1.) indicate that prediction error responses have mostly ceased in the time period we use to calculate response selectivity (late block 2), and differences in response strength between stimulus C in late block 2 and familiar stimuli A/B are likely due to more long-term changes in the visual feedforward pathway or local circuits in V1 caused by continuous exposure to a visual stimulus over many days.

Most importantly, we find selective amplification of visual responses also in cases in which we do not use this method of calculating response selectivity: prediction errors in response to a familiar stimulus at an unexpected location also lead to amplification of visual responses of only the most selective neurons (Extended Data Fig. 5). In this case no misclassification is possible because selectivity is calculated using neural responses to the familiar stimulus at the familiar location, which has been encountered by the animal for hundreds of trials since the beginning of training.

1.3 In Ext. Data Fig 2b, bottom (“grating responsive cells”), why are the responses to C4 expected lower than the responses to A1, B2 or A3 expected? Could this again suggest that

some of these neurons are indeed negative A or B-prediction neurons that do not strongly respond to C (nor to all other expected visual stimuli)?

The “grating responsive cells” in Extended Data Figure 2b bottom include neurons that significantly respond to the grating presented either in block 1 or in the second half of block 2 or both. There is indeed a large group of neurons that respond to unexpected stimulus C but very weakly or not to expected stimulus C. We have perhaps not made this sufficiently clear in the first version of the manuscript, although it was apparent from Figure 2a. We have now added a specific description of these neurons in the manuscript (lines 143 - 146 and new Extended Data Fig. 6f-k and Extended Data Fig. 8) and further clarify that the prediction error response is composed of visually-responsive neurons that are selective for the encountered stimulus and additional neurons that are only activated specifically by prediction errors. However, importantly, the large majority of these latter neurons still respond highly specifically to the unexpected stimulus presented, because they do not respond to other unexpected stimuli (Fig. 1, 2, new Extended Data Fig. 6) and they are not negative prediction error neurons because they do not respond to omissions of a stimulus (Rebuttal Fig. 4, see also response to point 1.2. for details).

1.4 The authors should further show that “tuned otherwise” (A1, B2, A3, or B4) or “untuned” (all but A1, B2, A3, B4, and C4 expected) neurons do not show the increase to C4 unexpected.

We have now included plots showing that “tuned otherwise” neurons (responding to A1, A3, B2 or B4 but not to expected C or D) do not respond to unexpected stimulus C or D (Figs. 2e, Extended Data Fig. 7e). As described in detail above (point 1.3.), some neurons respond exclusively to unexpected stimulus C and do not seem to be driven by visual stimuli (new Extended Data Fig. 6f-h).

1.5 In general, the authors should tie their results more into the literature.

We have now added potential pathways for prediction error-related computations from previous literature in the discussion (lines 450 - 470).

1.5.1 How are their results related to positive and negative prediction errors?

We do observe negative prediction error responses in V1, i.e. responses to the omission of an expected stimulus as described in Fiser et al. 2016 (new Extended Data Figures 2f and 9). However, these responses are in general weaker than responses to unexpected stimuli and they do not significantly contribute to the prediction error response when an unexpected stimulus is shown instead of an expected one (new Extended Data Fig. 9). For a more detailed explanation, please see point 1.2. We have now added this new data and description in the manuscript text (lines 146 - 151).

1.5.2 The response profile of omission neurons to stimuli A1, B2, and A3 in Ext. Data Fig. 2f look like “predictive neurons” in Fiser et al. 2016 (see pre-stimulus ramp up), all other neurons

look like “visual neurons” in that paper. Are those results consistent with the ones presented here?

Indeed, this is an interesting observation. We believe these neurons are not stimulus-predictive unlike the predictive neurons described in Fiser et al. 2016, because they show ramping activity during all inter-stimulus virtual corridor sections and these responses are not stimulus-selective. Moreover, in a different set of experiments we observed that such ‘ramping’ activity appears to be already present in the first training session when stimuli are not yet predictable. One possibility is that in our experimental condition (some of) these neurons respond to the grey walls of the virtual corridor present between stimuli and landmarks (i.e. the absence of a visual stimulus), and that this response is amplified when the grey wall is unexpected (in the absence of an expected stimulus).

1.6 Lines 99-104 should be clarified.

1.6.1 The authors state that “prediction errors have been proposed to encode the difference between predicted and actual visual input”. Do they mean on a single-cell level or on a population level?

The concept of the difference signal was originally developed on the level of individual neurons or the population average (e.g. Rao & Ballard, 1999, studies on mismatch negativity (MMN) etc.), but in principle it is applicable to high-dimensional activities of neural populations (Stalnaker et al., 2019). To the best of our knowledge, papers outlining this hypothesis - if they specify the neuronal implementation at all - usually refer to single cells (e.g. Rao & Ballard 1999). We had already stated in the last version of the manuscript that our results indicate that little information about how the actual input differs from predictions is represented at the level of individual V1 neurons (lines 113 - 117), but that we do not exclude that such information may be encoded on the population level in V1 (lines 479 - 481). Unfortunately, we have too few trials to test this convincingly in our data.

1.6.2 Further the authors state that “error responses could also represent a more unspecific surprise signal (encoding only the magnitude of the deviation)”. Are they referring to an unsigned error signal?

We were initially reluctant to use the term ‘unsigned prediction error’, because it is usually used in the context of reward stimuli, and often describes a neural signal of the same sign observed for both positive and negative valences (the unexpected presence and absence of a reward). In contrast, we have tested if the prediction errors observed in V1 are selective to a specific stimulus or if the same cells respond to any unexpected stimulus (unspecific surprise signal). Nonetheless, as in theoretical studies ‘unsigned prediction error’ and ‘surprise signal’ are often equivalent, we have now added this term to the manuscript text (line 104 - 105).

1.7 Lines 114-116 should further specify that this statement is based on grating responsive cells.

We did not base this statement only on grating-responsive neurons, since the plot in Figure 1i includes a large number of neurons that are not visually-responsive, but only respond to stimulus C when it is unexpected (not when it is expected or to any other stimulus we have tested). Below, we re-plotted Figure 1i only for these non-grating responsive neurons (Rebuttal Fig. 4). The prediction error response of these neurons is still very similar for the same unexpected stimulus C presented in different locations (meaning that expectations were different), indicating that they do not strongly encode the specific difference between actual and predicted stimulus.

Rebuttal Figure 4a, Average neural responses to stimulus C presented at location 2 (C2) or at location 3 (C3) in the virtual corridor of V1 neurons responsive to unexpected C but not significantly responsive to expected C. **b**, Prediction error response strength of individual V1 neurons to stimulus C2 plotted against stimulus C3 for V1 neurons responding to unexpected but not expected stimulus C. **c**, Average neural responses to stimulus omission (top) and stimulus D (bottom) of V1 neurons responsive to unexpected stimulus D, but not significantly responsive to expected stimulus D. **d**, Prediction error response strength of individual V1 neurons to stimulus omission plotted against stimulus D for V1 neurons responding to unexpected but not to expected stimulus D.

1.8 Lines 161-163 seems overstated – this would mean that there are no negative prediction error neurons in layer 2/3 of V1.

We have now added “...when the animal encounters an unexpected stimulus...” to specify that we are not discussing negative prediction error signals here.

2. The thought experiment 2 (Fig. 1j) distinguishes between two scenarios – neurons coding for an unspecific surprise signal or an amplified sensory response. However, there are several points that remain puzzling.

2.1 Stimuli C and D are radically different stimuli recruiting different neurons in the cortex (prediction error so far in orientation space). Can the results be explained by the different nature of the stimuli? The authors should demonstrate that this effect holds true when using for example a horizontal grating as stimulus D.

This experiment was specifically designed to show that the responses are stimulus-specific, i.e. different for different stimuli, while a pure surprise or motor signal should be similar independent of how different the unexpected stimuli are. Nevertheless, as suggested by the reviewer, we have now repeated the experiment with more similar unexpected stimuli: the virtual corridor was changed such that mice encountered unexpected gratings of either +45 degrees or -45 degrees orientation. The results were very similar to Figure 1m,n; the large majority of neurons showed prediction error signals only to one but not both stimuli (new Extended Data Fig. 6l-p).

2.2 Assuming an unspecific surprise signal in addition to tuned responses (Fig. 1j, left option), I would expect a noisy version of Fig. 1j, right, rather than a diagonal. Perhaps the authors mislabeled the axis and it should say “unexpected C – expected C” and “unexpected D – expected D” for x and y-axis, respectively, as in Fig. 1n.

To clarify we have now changed the axes labels according to the reviewer’s suggestion.

2.3 VIP responses in Ext. Data Fig. 5d resemble an “unspecific surprise signal”. The authors may consider highlighting this fact since it highlights the special role VIP neurons play in this computation.

Thank you for the suggestion, we now emphasise this in the manuscript text (lines 197 - 199).

2.4 Ext. Data Fig. 5, 7 for VIP neurons and pulvinar inputs, respectively, are not plotted the same way. The authors should reanalyze the data as for excitatory neurons in Fig. 1n, 2d,e.

Unlike excitatory neurons, VIP neurons are suppressed by expected visual stimuli, making it impossible to calculate response selectivity in the same way. For pulvinar inputs, we unfortunately do not have the data of responses to two different unexpected stimuli from the same boutons. However, we have now adapted the analysis in new Extended Data Figure 14 to match Figure 2d,e.

2.5 Is the distribution in Fig. 1n random? In other words, is the density of neurons in the top right quadrant (z value > 0.5 for both C4 and D4) lower than expected by chance?

Our main interest was to determine if there was a positive correlation between the prediction error responses to unexpected stimulus C and unexpected stimulus D. This was clearly not the case. However, we have now tested if fewer neurons than expected by chance lie in the top right quadrant (responding to both C and D, quantified by calculating the absolute selectivity or responses $(C - D) / (C + D)$ and comparing their distribution with a shuffled data set (Rebuttal Figure 5). We find that V1 responses to the two stimuli are more selective than expected by chance.

Rebuttal Figure 5. Left: similar to Figure 1n, prediction error response strength to stimulus C plotted against prediction error response strength to stimulus D for all non-VIP neurons responsive to C or D. Right: same as on the left but excluding neurons not responsive to C and D. Inset: Distribution of prediction error response selectivity $((C - D) / (C + D))$ of V1 neurons in the right scatter plot compared to a shuffled data set.

3. The responses to C4 and D4 unexpected (Fig. 1c,l) appear to be significantly larger than those to C2 and C3 unexpected (Fig. 1g). Is this pointing to a potential confounder due to upcoming reward signaling? Pakan et al. 2018 show that activity prior to the reward starts ramping up largely before the reward.

3.1 The authors should add an analysis excluding neurons that have responses associated with the reward.

From the original data it was not clear if the response to the same unexpected stimulus C is larger when it is encountered closer to the reward location, since we did not have the data to compare these responses in the same neurons or the same animal. We have now performed this new experiment in which we compare responses to unexpected stimulus C at position 2 and position 4 in the same neurons. There is only a slightly larger response to the unexpected stimulus C when it is presented at position 4, closer to the reward location (new Extended Data Fig. 4a-c), potentially because the stimuli located closer to the reward may be more relevant to the animal. However, this does not confound the prediction error signal, because if we exclude neurons active around reward onset, the size of the prediction error signal does not change, and is still slightly larger for unexpected stimulus C at position 4 (new Extended Data Fig. 4d,e).

3.2 The authors should show B4 in Fig. 1c and 2b (ideally from block 1 and from late training sessions).

We have not added B4 responses to Figures 1c and 2b, since B4 is not presented in block 2, preventing us from plotting the data in the same way which could be confusing to the reader. However, V1 responses to B4 are shown in new Extended Data Figure 2g, and directly compared to B2 responses of the same neurons (new Extended Data Figure 5c), showing that responses to the same expected stimulus B at the two different positions were very similar.

4. In the current manuscript, the data do not clearly "...demonstrate that the recruitment of VIP interneurons is required for the generation of prediction error signals..." (lines 196, 197).

4.1 The results are equally consistent with a general reduction in gain, independent of a contribution to the computation of a prediction error signal.

Optogenetic silencing of VIP neurons very specifically and strongly reduces V1 prediction error responses and has no effect on the responses to expected stimuli A or B (Figure 3f, left and middle). Moreover, there is a striking correlation between the strength of the prediction error signal and the effect of VIP silencing for individual V1 neurons (Figure 3g). Finally, VIP neurons respond only to unexpected, not to expected visual stimuli and can therefore not contribute to purely visual responses of V1 neurons (Figure 3b,c). These results indicate that VIP neurons directly contribute to the generation of prediction error responses in L2/3 neurons in V1, and are not consistent with a general reduction in gain. We are, however, not claiming that VIP neurons are involved in computing the prediction error from visual input and internal predictions, they may simply inherit prediction error signals from a different source and our experiments can not distinguish these possibilities. We have now rephrased relevant text to make this clearer.

4.2 The authors should also plot "unexpected C4 – expected C4, both LED on" vs "unexpected C4 – expected C4, both LED off".

We have now added this plot to Extended Data Figure 13h and p.

4.3 What do the authors conclude from the significant effect of VIP silencing in Figure 3f, right?

Please see our response to point 1.1.3. in which this question was also raised.

4.4 How well are the authors silencing VIP cells? They should add a plot of the activity of VIP during silencing.

We have now performed new experiments that enabled us to image the activity of VIP neurons while they are optogenetically silenced using Cre-dependent eNpH3.0 and GCaMP6f in VIP-Cre: Ai14 mice. We find that silencing was highly effective, as prediction error responses of VIP neurons were completely suppressed (new Extended Data Figure 12a-c).

4.4 In Ext. Data Fig. 5j,k, why can the VIP not be visualized (line 961) despite the fact that they express mCherry?

mCherry aggregates in neurons and forms puncta when fused to eNpHR3.0, as has been previously reported (Alemi *et al.*, 2023), VIP neurons expressing eNpHR3.0 can therefore not be clearly identified.

4.5 The authors pool responses to C4 and D4, even though the stimuli evoke very different responses in VIP neurons and potentially pulvinar inputs. Do all the results in Fig. 3 hold for C4 or D4 stimuli separately?

Yes, all results hold when separating experiments with C and D stimulus. We have now separated responses to C and D in Figures 3b,c,i and j and are plotting Figures 3f,g,m and n separately for sessions with C and D stimulus in new Extended Data Fig. 13.

5. The authors causally demonstrate the involvement of pulvinar circuits in generating prediction error signals. There are some aspects which leave this statement in some doubt.

5.1 The authors express eNpHR3.0 in the pulvinar and optogenetically silence their axons in V1 (Fig. 3l). Are they actually silencing the axons with their manipulation?

5.1.1 The authors could either record in axons during optogenetic silencing of the same axons, or

5.1.2 Implant a fiber in the pulvinar and silence cell bodies in a few animals to confirm the effectiveness of their manipulation.

We have performed new experiments to image pulvinar axons in V1 while optogenetically silencing them by expressing GCaMP6f and eNpHR3.0 in the same pulvinar neurons. As expected, we observe a small but significant reduction in pulvinar axon responses during the optogenetic manipulation (new Extended Data Fig. 12d-f). This partial effect explains why there is only a small effect of pulvinar input silencing on prediction error responses of V1 neurons (Figure 3m and Extended Data Figs. 11 and 13).

5.2 As for the VIP manipulation (see major point above), the results presented in Fig. 3m could also be a result of a general effect on the gain.

We are confused by this comment. If pulvinar silencing had a general effect on the gain of V1 responses, the responses to expected stimuli A or B should be affected similarly to prediction error responses. However V1 responses to A/B are not suppressed by pulvinar silencing, if anything there is a weak trend towards increased responses (Figure 3m, $P = 0.074$). Therefore, it is specifically prediction error responses that are reduced by this manipulation, confirmed by the correlation between prediction error response strength and pulvinar silencing effect for individual V1 neurons (Figure 3n). Moreover, there is no significant correlation between the pulvinar silencing effect and the response strength to expected stimulus C or D (new Extended Data Fig. 13n).

5.3 As above, the authors should plot “unexpected C4 – expected C4, both LED on” vs “unexpected C4 – expected C4, both LED off”.

We have now added this plot to Extended Data Fig. 13p.

6. It is surprising that pulvinar input and VIP neurons act synergistically (Fig. 4d) although the individual manipulations either show no effect (Fig. 4c) or even the opposite (Fig. 4b, for highly selective neurons). This result would be more convincing if the authors show the individual and combined manipulations in the same mice (using soma-targeted LP expression and implanting a fiber). This would further allow to see if the two mechanisms act on the same neurons.

While it was initially surprising to us that pulvinar input and VIP neurons act synergistically, these results can be entirely explained by the circuit mechanism we discovered, namely that 1) pulvinar activates a subset of somatostatin-positive (SOM) neurons that have strong inhibitory influence on the V1 network, 2) VIP neurons predominantly target SOM neurons with inhibitory synapses (Pi et al., 2013, Pfeffer et al., 2013, Lee et al., 2013, Schneider-Mizell et al., 2024), such that 3) when VIP neurons are active simultaneously with pulvinar input they inhibit SOM neurons and thus counteract the pulvinar-driven, SOM-mediated inhibition of V1 responses. The key experiment that provides confirmation of this mechanism was the simultaneous activation of pulvinar input and silencing of SOM neurons. Simultaneous SOM inhibition indeed entirely counteracts the strong inhibitory effect of pulvinar activation on the V1 network, and mimics the effect of pulvinar and VIP neuron co-activation (Figures 4d,e).

We do not think that trying to achieve separate and simultaneous activation of pulvinar and VIP neurons in the same animal would be more convincing than the large number of experiments we have already performed. The experiment the reviewer suggests is extremely challenging. We have yet to achieve 2-photon imaging through a cranial window with a precisely targeted implanted fibre close-by, and I have not seen a published study that performed this type of experiment successfully. Moreover, even if achieved I would not be convinced that activation of either only pulvinar input to V1 or only VIP neurons would truly be independent, since (1) soma-targeting is not complete such that soma-targeted constructs are still weakly expressed in the axons, and (2) laser light delivered through the cranial window or the implanted fibre also reaches unintended targets. Moreover, this experiment would be less specific since we would manipulate cell bodies in the pulvinar, which project to many other cortical areas, rather than specifically pulvinar input to V1.

7. The authors target the pulvinar (or LP) for viral expression of opsins and calcium indicators. LP, however, is adjacent to LD and dLGN, which have different functions and projection patterns targeting specific neuronal subtypes (Ma et al. 2021). Based on their Fig. S13 (Ma et al. 2021), LD seems to mainly target SOM neurons while LP mainly targets VIP neurons. Hence, it is crucial that the injections in the current manuscript do not leak into other thalamic nuclei.

For this, the authors should present representative images of the injection site in LP for all animals individually together with the corresponding effect magnitude.

Figure S13 in the study of Ma et al. only shows one slice of LD and lateral LP, with more neurons targeting V1 SOM cells (and also V1 VIP cells) in LD than LP. The data quantification in Figure 1c of Ma et al. and other studies (Yao et al. 2023, Pouchelon et al., 2021) show clearly that neurons in both LP and LD project to SOM neurons in V1. In our experiments, we took explicit care to restrict virus injections to pulvinar/LP. We have now included images of all pulvinar/LP injection sites of Figure 5 in new Extended Data Fig. 18 together with the data recorded in each animal. These injections are representative for pulvinar/LP injections across the study, and are largely restricted to LP.

Minor points:

- Redo all stats with hierarchical bootstrap sampling (Saravanan et al. 2020) or similar.

All stats performed on pooled neural data have now changed to hierarchical bootstrap sampling. For figures showing data points for individual mice and imaging sessions separately, statistics are done with a Wilcoxon signed rank test.

- For several plots (e.g. Fig. 2d) the effects is seen only in few cells. To ensure that the effects are consistent across animals, the authors should plot all major effects for all mice individually.

We have now changed all relevant data figures to show effects for individual mice.

- Line 30/Ext. Data Fig 1: The authors should also show running traces during training.

Unfortunately we did not consistently record running speed during training sessions. However, running speed was very consistent over several days once animals were habituated to the setup.

- Line 79/Fig. 1d: Did the response to B4 change when presenting C4 unexpected? Please show how the response to B4 evolves during Block 1.

We have now included V1 responses to B4 across block 1 (B4 is not presented in block 2) in new Extended Data Fig. 2g, showing that responses to B4 do not change over time.

- Line 83: Please rephrase – the pupil size was different for expected C4 in Ext. Data Fig. 1d. Why do the authors think that the pupils were dilated in this condition? Could this be because the mouse is still learning/updating? Comparing these data points to the pupil size during learning might explain the difference.

Please also see our response to point 1.1.2. Pupil size was on average slightly increased in the second half of block 2 independent of the visual stimuli and stimulus expectedness (at both position 3 and 4). We saw an opposite trend in sessions with unexpected stimulus D, indicating this drift over time is not a general sign of learning/updating.

- *Fig. 1a: Was A always oblique and B always vertical or did the authors correct for this asymmetry in their paradigm?*

In the initial sets of experiments stimuli A and B were consistently oblique and vertical, respectively. However, in new experiments, we changed the design of the corridor and used different grating orientations with similar results (new Extended Data Fig. 6l-p).

- *The authors use z-scored dFF throughout the manuscript. While this facilitates comparison across sessions and mice, it biases the analysis towards neurons with low baseline activity. Do the main results hold with dFF?*

Yes, our results hold when using dF/F instead of z-scored dF/F. We have now included plots showing dF/F response strength to expected and unexpected stimuli for neurons with different response properties in new Extended Data Figure 7g,h.

- *Fig. 1d: For consistency with Fig. 3d,k, please show the mean of A1, B2, A3 rather than just A3.*

To keep Figure 1d consistent with Figure 1c, we would like to keep this panel as it is. However, we have now included a new Extended Data Figure 2g showing mean responses to A1, B2 and A3 over time.

- *Fig. 1e,j: It has to be very clear that these are thought experiments and should not have plots that resemble real data (individual dots with noise).*

We have now adapted panels 1e,j to make clearer that these are thought experiments

- *Fig. 1g: Scale missing.*

We have now added the scale bar.

- *Line 92: For clarity, “highly selective” should be in the text.*

The analysis showing that responses are increased only in those visually-responsive neurons that are selective for the presented stimulus is only introduced later in the manuscript and this addition would therefore be confusing at this location.

- *Line 93: For clarity, “for omission responsive neurons” should be in the sentence but then the statement becomes trivial. Please rephrase.*

We have now re-phrased this sentence (lines 92 - 94).

- *Fig. 2e, 4b-e: Some of the “non-selective” neurons are highly selective for the other stimulus. This should be divided into 3 categories with the highly-selective neurons of the other stimulus in the Ext. Data Figs.*

The neurons included in the original Figure 2e and the right column of Figure 4 were only neurons that are significantly responsive to expected stimulus C or B, respectively. Those neurons were then divided into highly-selective (responding exclusively to this stimulus) and less or non-selective neurons (also responding to other visual stimuli). Neurons selectively responding to other stimuli were not included in those plots. We have now added a third category to all of these plots showing responses of neurons selective to the other (familiar) stimulus.

- Fig. 3f,m, 5c: Add LED-on time as in Fig. 4.

We have now added this information in the figures.

- Fig 5: How do the authors explain the small responses of SOM neurons to full-field stimuli (Adesnik et al. 2012)?

While the study of Adesnik and colleagues indicated that SOM neurons have less surround suppression than other V1 neurons, not all SOM cells in their study responded strongly to large stimuli. Moreover, we included all recorded SOM neurons in the plots in Figure 5d and e, while Adesnik et al. included only SOM neurons that significantly responded to the presented stimulus.

- Ext. Data Fig. 5e,i: The legends state “polar plot” instead of “pie chart”.

We have now corrected this in the figure legends.

- Ext. Data Fig. 9d,e: For consistency with Fig. 5, these plots should be broken down into “recruited cells” and “other cells”.

We have now adapted this figure to match Figure 5.

- Citation 21 has no date.

We have now added the date.

Referee #2:

In this paper, Furutachi et al., investigate how circuits of the visual cortex and higher order visual thalamus give rise to the observation of strengthened neuronal responses following unexpected stimuli or events, also referred to as mismatch or prediction error responses e.g. in the context of predictive coding. The authors discovered that the enhanced neuronal response to an unexpected stimulus does not merely reflect a generalized mismatch signal, nor does it reflect how the unexpected stimulus deviates from expectation (as suggested in predictive coding theory), but the enhanced response specifically encodes information of that particular stimulus. The main result of this paper is the unraveling of a novel synergistic circuit in which projections from the pulvinar modulate the interaction of local vasoactive-intestinal-peptide (VIP) expressing and somatostatin (SOM) expressing interneurons, thereby gating/amplifying stimulus specific responses in excitatory visual cortex neurons.

This is an important and well executed study. Firstly, it identifies several key components of the circuit that drives the amplification of neuronal responses to unexpected stimuli in primary visual cortex. While these findings are not completely in disagreement with predictive coding theory, i.e. they show that prediction errors are an important signal in visual coding, the study demonstrates in a series of very clear and elegant experiments how early visual processing reflects the sensory aspect of visual information, rather than the deviation from the expected input. Beyond implications for predictive coding theory, these insights are an important step to a better understanding of the computations and functions of the neocortex. Secondly, the authors elegantly demonstrate the (very likely general) principle of how neuronal activity can depend on multiple, often synergistic pathways, which might be misunderstood (or their complexity might be underestimated) when studied in isolation. In conclusion, I am very enthusiastic about this work.

We thank the reviewer for their encouraging remarks.

Nevertheless, I do have a number of concerns regarding methodology and the interpretation of the findings that need to be addressed.

Major:

1. Important to this study is the concept of an internal model. While the internal model cannot be directly observed, violations of the model's predictions are thought to change behavior. The authors also report a change in behavior to the unexpected stimulus C4, in block 1, arguing that this indeed shows that mice did not expect the novel stimulus. However, the behavior of the mice does not show a difference between the period when the novel stimulus is unexpected (C4 block 1) and when it is expected (C4 block 2). This is at odds with the interpretation of the neuronal data, in which stimulus C4 in block 2 is supposedly expected, presumably following a learning process that updates the internal model. Can the authors provide behavioral evidence of their assumption that mice learned to expect stimulus C4 in block 2?

While most animals slowed down in response to an unexpected stimulus, running speed appeared not to be a reliable indicator across all mice for how 'expected' this stimulus became after repeated exposure. We have now adapted the text (lines 29 - 32) to clarify this. However,

please note that a subset of mice did gradually recover their running speed by the end of block 2 (Rebuttal Figure 1), and in a new set of experiments that included an extended block 2 with more trials, all animals showed recovery of the deceleration induced by an unexpected stimulus (Rebuttal Figure 2).

Related to this, ED Fig 1h shows that the mouse's running speed was not correlated to the response strength to grating C. However, this appears to be only for block 2. It is not clear for which data plot ED Fig 1i is. The relationship between running speed and response strength should also be investigated for block 1 (when C4 was newly introduced).

For the plot in the original Extended Data Fig. 1i (now Extended Data Fig. 3e) the entire trace over the recording session (block 1 and block 2) was used. We have now clarified this in the figure and figure legend. We have now also added quantification of the relationship between running speed and response strength in block 1 as suggested by the reviewer (Extended Data Fig. 3a). V1 responses are slightly stronger in trials with faster running speed, which is expected because running speed is known to directly modulate V1 responses (e.g. Niell and Stryker, 2010, Fu et al., 2014). This shows that changes in running speed (deceleration) in response to the unexpected stimulus cannot explain increased V1 responses to the unexpected stimulus, as slower running speeds cause on average decreases in V1 responses (as seen in Extended Data Fig. 3a and c). Thus, the data in Extended Data Figures 1 and 3 show that running speed is not always a reliable behavioural indicator for the strength of prediction error responses, especially because the effect of running deceleration generally decreases V1 activity while prediction violations increase neural activity.

2. In the introduction, the authors propose a line of reasoning in which the internal model provides a prediction, which when violated leads to increased V1 responses and updating of the model. The authors suggest that the observed increased sensory responses are instructive in updating the model. If that is the case, the updating of the model should be affected when activity is suppressed (e.g. as in some of the optogenetic experiments). Can the authors confirm this? And related to this, some mice might learn to expect the stimulus faster than others, is this correlated with the neuronal effects?

This is a very interesting and, in our opinion, fundamental question in neuroscience: what is the function of prediction errors and are they actually used to update internal models and therefore predictions about the world. We plan to address this question in the future as our study was not designed to answer it, instead focusing on the circuit mechanisms of error signals in V1.

In the discussion, we provide a narrative about the potential functions of sensory prediction error signals in V1, one of them being the involvement in updating internal models. Interestingly, some of our results do suggest that prediction error responses in V1 may indeed be important for updating predictions (see also our response to reviewer #1's point 1.1.3). We observe that when VIP neurons are silenced in 50% of presentations of unexpected stimulus C (and the prediction error signal in V1 therefore suppressed), responses of VIP neurons to stimulus C are higher in block 2 than in control animals (Rebuttal Fig. 3), suggesting that the

updating of predictions may indeed be delayed. However, the extant experiments are not suitable to comprehensively address this question, which would require a new experimental design and a large amount of new data.

In response to the reviewer's second question, there is indeed a trend that animals that recover their running speed upon encountering stimulus C in block 2 (indicating that they may be faster in learning to predict the novel stimulus) show a slightly faster decrease in prediction error responses in V1 (Rebuttal Fig. 6). This inter-animal variability in speed of learning will also be explored in future work.

Rebuttal Figure 6. Average calcium responses to novel stimulus C across trials and blocks of mice showing recovery of the running deceleration caused by the initially unexpected stimulus C (dark gray) and the remainder of the mice (light gray). Symbols and error bars depict mean and bootstrap 95% CI.

3. The authors treat single neurons as independent observations. This makes sense in some cases, e.g. when the focus lies on how specific neurons change response properties across conditions or over time. However, in the case of studying effects that are expressed broadly across circuits spanning many neurons (such as the violation of a prediction of the internal model), the correlations/dependencies between neurons within a single biological replicate could artificially overpower the study. While I understand that not every effect can be tested across groups of animals (in a study as complex as this), the authors should convincingly show that the main results hold across -real- biological replicates.

We have now changed the main data figures to show data points and statistics for individual animals and imaging sessions, and have used hierarchical statistics for pooled neural responses.

Related to this, in scatter plot figures like Fig 1h or 1i, it is unclear whether the z-scored dF is also normalized per mouse. If this was not done, systematic differences in group means across animals could result in spurious correlations. There should be (at least) a control for this.

dF/F was z-scored, such that average dF/F over the recording session is the same across mice, but the average response strengths to specific stimuli as in Figure 1h are not normalised across mice. We have now replotted figure 1h with prediction error responses normalised across animals showing identical results (new Extended Data Fig. 4g).

4. I have a hard time following some of the reasoning for using specific analysis approaches and statistical tests. This should be streamlined. For instance, why do the authors report both the Pearson and the Spearman correlation coefficients for a single scatter plot? Why did the authors use a longer time window for investigating neuronal responses in Fig 5 and ED Fig 9? Why do the authors use two different methods for calculating selectivity, arguing that one of the methods is less sensitive to noise than the other; would it not be better to use the best/most appropriate method throughout? Why do the authors occasionally show 95% confidence intervals (e.g. ED Fig 3b,c) and in other plots use a Wilcoxon signed-rank test (e.g. ED Fig 6b) to compare group wise differences?

We thank the reviewer for pointing this out and have now further streamlined our analyses and statistical tests. We now use only Pearson correlation coefficients, and the same time window for Figure 5 and Extended Data Figure 9 as the other figures (initially the time window for SOM recordings in Figure 5 was longer since responses of SOM neurons are on average slower, but results are similar for different analysis time windows). We have now adapted the method for calculating selectivity in new Extended Data Figure 14 to match Figure 2d,e. Moreover, we now use hierarchical bootstrap sampling when comparing pooled neural responses, and Wilcoxon signed-rank test to compare data points across animals.

5. The conclusion of the paper is that the responses of the most stimulus-selective neurons are amplified. How does this rhyme with that there are many more cells recruited (significantly responsive) for the new grating (C4)? Following the reasoning of amplifying only highly stimulus-selective neurons, I would have expected the number of responsive cells for C4 to be the same as for e.g. A3, with the C4 responsive neurons having larger response amplitudes. This requires a proper quantification and explanation.

This is a good point that we did not emphasise enough in the initial version of the manuscript. We have now added a more detailed description of the neurons that respond to prediction errors (e.g. unexpected stimulus C), but little or not to the same stimulus when it is expected (i.e. to stimulus C in the second half of block 2). (lines 143 - 146 and new Extended Data Fig. 8 and 6f-k), and clarify that the prediction error response is composed of visually-responsive neurons that are selective for the encountered stimulus and additional neurons that are only activated specifically by prediction errors. Importantly, these additionally recruited neurons respond highly specifically to the unexpected stimulus presented, they do not respond to other unexpected stimuli or omissions of a stimulus (Figures. 1, 2, Extended Data Figs. 6 and 9). See also responses to reviewer #1's points 1.2. and 1.3.

Minor:

1. In Fig 1a, the grey parts in the 'stimulus timeline' suggest that the corridor had large sections of uniform gray wall, which I assume was not the case?

There were indeed sections of grey wall between the different stimuli and landmarks to ensure other stimuli are not visible when one stimulus is encountered and to ensure sufficient decay of calcium signals between stimuli. However, the virtual corridor included a patterned floor that was always visible, ensuring that the animals experienced optic flow during running even when the walls were uniform.

2. Something that puzzles me is that mice typically do ~90 traversals a day and then on the C session do 200 traversals. How come the mice do more trials? And related, why is the pupil size overall larger in late sessions? Could this reflect some change in overall internal state in block 2? Could this affect the interpretation of the results?

Fully trained mice can run for more than 200 traversals if they are well habituated to the experiment and well-motivated by food restriction. The 90 traversals are an average over training, including sessions when animals were still not fully habituated and performed fewer traversals.

Pupil size is not larger in later sessions, but is on average slightly increased (non-significantly, now that we optimised the statistical analyses throughout the paper and corrected for multiple comparisons) towards the end of the imaging sessions in which stimulus C was introduced. This increase in pupil size is independent of stimulus presentation or predictability of stimuli and is therefore likely due to a small change in the general state of some of the animals over the session. Moreover, pupil size does not increase but slightly decreases in imaging sessions with unexpected grating D (Extended Data Fig. 1i,j), showing that over time measures like pupil size likely drift. However, importantly, V1 responses to expected stimuli A1, B2 and A3 do not change over time but are stable and similar in block 1 and 2 (Figure 1c, Extended Data Fig. 2), showing that such subtle changes in the pupil size of the mouse did not noticeably affect neuronal responses in V1.

3. In Fig 1m and 1n, there are a couple of neurons on the diagonal. Does this mean that there might be some, albeit few, V1 neurons reliably coding for unspecific surprise?

Please note that Figures 1m and n include VIP neurons, which respond much less selectively than excitatory neurons to different unexpected stimuli (see Extended Data Figure 10, and we have now emphasized this difference more in the text). When excluding VIP neurons (Extended Data Figure 6c-e), there are fewer neurons around the diagonal than expected by chance (please see Rebuttal Figure 5).

4. ED Fig 2 demonstrates that showing the same grating in different positions leads to slightly different responses. Considering the study of Saleem et al. (2018), I had expected these differences to be larger. Is there an explanation for observing smaller differences here?

We indeed observed that responses to the same stimulus when presented at different locations of the virtual corridor were very similar. As mentioned above, our corridor included long stretches of grey wall between stimuli, such that visual input of the same stimulus at different locations was identical. The study of Saleem et al. used a very short corridor with patterned stimuli throughout and importantly the corridor ahead was visible at all times on the central monitor. While this design may have helped mice build a better sense of space, the visual input mice received at two different locations was not identical - which may explain the larger difference they observe.

5. The authors state that neurons with a selectivity index < 0.6 are all unselective (e.g. Fig 2d). However, a large fraction of these neurons has strongly negative selectivity indices. In my opinion, these are selective as well, just in the opposite direction. Can the authors explain why they do not observe 'inverse' effects for this subgroup?

We apologize if this was unclear in the manuscript. We do of course not claim that neurons with a selectivity index < 0.6 in Figure 2d are unselective, neurons with a negative selectivity index towards -1 are selective for the other stimulus A or B. Importantly, the original Figure 2e did not include neurons selective for stimulus A or B, but only neurons that significantly responded to expected stimulus C, either selectively or non-selectively (also responding to other visual stimuli). We have now included a third category of neurons selective to stimuli A/B in this and all equivalent plots. These neurons do not respond either to expected or unexpected stimulus C (or D). More generally, the feed-forward visual drive determines which visually-responsive population of V1 neurons will be facilitated, e.g if an unexpected A stimulus is presented, A-selective neurons are facilitated (see e.g. Extended Data Fig. 5).

6. Neurons appear to find the 'D' stimulus more unexpected than the 'C' stimulus. Do the animals agree? And does this reveal something about the involved circuits?

Stimulus D indeed evoked stronger prediction error responses, likely because it is a different type of stimulus than the gratings mice have encountered before. Mice also slowed down in response to unexpected stimulus D (Extended Data Fig. 1g,h), but as discussed above, we do not find running speed to be a reliable measure of how surprised an animal was. We show that the same circuit mechanism underlies the amplification of V1 responses to both unexpected C and D. In an extensive set of new experiments we now show that the weaker prediction error responses to a familiar stimulus (A) at an unexpected location also rely on VIP neurons and pulvinar input (new Extended Data Fig. 11). The difference in strength of prediction error signals in response to unexpected D, C and A suggests that prediction errors in V1 are scaled according to how 'surprising' stimuli are. Our experiments were not designed to reveal the mechanism for the scaling of the prediction error signal, but one hypothesis is that it may involve neuromodulatory input onto VIP neurons.

7. In Fig 3i, a number of axons show negative responses, how should this be interpreted?

Some pulvinar axons are suppressed by expected or unexpected visual stimuli. Negative stimulus responses are not uncommon in V1 and other stages of the visual processing hierarchy and are thus also expected for pulvinar (see e.g. Poort, Khan et al. 2015).

8. Do the authors propose that there are two types of SOM neurons, those recruited by pulvinar and part of a general prediction error circuit, and others that are not? Or do the authors explain their observations by subsets of SOM neurons being part of different stimulus-specific circuits?

We do not believe that our observations can be explained by stimulus-specific SOM circuits. Our data indicate that there are (at least) two types of SOM neurons in layer 2/3 with different functional input/output connectivity: one type is driven by pulvinar, not strongly visually responsive and suppressed by prediction errors - indicating that these neurons are not strongly recruited by the local excitatory V1 network and receive VIP inhibition. The remainder of SOM cells is not driven by pulvinar, visually responsive and activated by prediction errors - indicating that these neurons are recruited by the local excitatory V1 network (as previously described for SOM neurons, e.g. Adesnik et al. 2012), and do not receive strong VIP inhibition. Recent studies utilising single-cell transcriptomics or electron microscopy have shown that there are several molecularly-distinguishable subtypes of SOM-expressing cells in V1 layer 2/3 (Tasic et al. 2016), and only a subset of (putative) SOM cells receives strong inhibition from (putative) VIP cells in V1 layer 2/3 (Schneider-Mizell et al., 2023 bioRxiv), consistent with our observations.

9. In Fig 5d, some 'recruited cells' have a negative response to gratings, and to 'led on/off'. How exactly would these be considered 'recruited'?

Neurons are included in the 'recruited cells' group if they significantly increase their responses to any of the stimuli during pulvinar activation. This measure is independent of these neurons' visual response and the plots in Figure 5d show that they are not visually responsive. A small fraction of these neurons is modulated by pulvinar activation only in some but not all stimulus conditions such that their data points are close to or below zero in some of the plots.

10. The authors use a Gaussian mixture model to estimate F0. I can see advantages of this approach, but is there any evidence that it is better than established methods? Related to this, do the authors correct for a potential slow drift in F0 values over the course of the experiment?

The Gaussian mixture model gave the most reliable results in previous studies of the lab, but is comparable to other established methods. There is no post-hoc correction for potential slow drift, but stimulus-evoked responses are normalised to the pre-stimulus baseline to correct for slow drift.

11. The authors should explain better what they mean with "signal contribution" and how the threshold prevents this? (see e.g. line 645)

Thanks for pointing this out, we have now rephrased this sentence (lines 698 - 700).

12. *In ED Fig 5f I would like to see the VIP response to expected B4 in the plot as well.*

We have now added VIP responses to B4 in Extended Data Figure 10b and non-VIP responses to B4 in Extended Data Figure 6b.

Typos:

- *Fig 5e: 'n' is not stated (neither in figure, nor in legend)*
- *Line 853: The bold 'd' should be 'f'*
- *ED Fig 5e,i: These are -not- polar plots*
- *Line 952: "Difference" should not be capitalized.*
- *Line 1031: "ontogenetic" should be "optogenetic"*

Thank you, we have corrected these omissions and typos.

Referee #3:

In this manuscript, Furutachi and colleagues report that prediction errors in mouse V1 amplify layer 2/3 neuronal responses to unexpected stimuli, rather than a surprise signal or a computation of the difference between expected and actual visual input. They describe a circuit mechanism for this computation: inputs from the pulvinar nucleus enhance visual responses to unexpected stimuli while simultaneously inducing inhibition in V1 through the activation of a subset of somatostatin interneurons. This circuit interacts cooperatively with VIP interneuron activation by unexpected stimuli, leading to the inhibition of somatostatin interneurons that gate excitatory pulvinar input to V1. This results in the amplification of V1 responses to unexpected stimuli.

The experiments presented in the article are well-executed and described in great details. The article includes datasets of high quality that will be valuable for the circuit neuroscience community. The question of the neuronal circuits underlying predictive processing of sensory information is of high relevance for this field of research. This article brings a valuable contribution towards this endeavour consistent with recent articles in the field (articles for G. Keller's group; Fang et al., Neuron 2020; Hu et al., Neuron 2019).

One major challenge to test prediction errors is to experimentally constrain predictions. My major concern with this manuscript is the unclear distinction between (1) an unexpected stimulus in the sense of a known stimulus but presented at an unexpected location (ext. fig. 3) and (2) a complete novel stimulus (all main figures). Both are unexpected stimuli but the results presented for case (1) (ext. fig. 3) only show a very small effect on neuronal responses in V1 and the underlying mechanisms are unexplored. It is thus unclear whether the same mechanisms underlie these two types of unexpected stimuli. This distinction needs strong clarification in order to support the strong, general conclusion about neuronal circuits underlying sensory prediction errors.

We thank the reviewer for their positive comments. We have now performed a large set of new experiments to confirm that increased V1 responses to a familiar stimulus at an unexpected location are also mediated by the same circuit mechanisms as those underlying amplified responses to a novel stimulus. See our detailed responses below and Extended Data Figs. 5 and 11.

Major issues:

- 1. The data shown in the manuscript indicates that the enhanced responses to unexpected stimuli depend on both expectation and novelty, and mostly on the latter. This is clear when comparing ext. fig.3c and 3e and Fig 2b, 2e (it would be informative to have the same y-axis for panels e in both figures). While neuronal responses to a familiar stimulus A (in unexpected position 4) are slightly larger than those to the same familiar stimulus A at an expected position (position 3, Ext. Fig 3c,e), this difference is much smaller than that observed when comparing responses to an unexpected novel stimulus at an unexpected location vs an expected familiar stimulus (stimulus C in position 4 in early and late trials (Fig 1c, d)). This result indicates that most of the change observed in these experiments relates to novelty and not to violated

expectations. To directly test this, authors should compare the enhancement of responses induced by expectation (A3 vs A4) with that induced by novelty + expectation (unexpected C4 vs expected C4). This comparison together with data shown in Ext. Fig3 should be presented in the main figure. In order to support general conclusions about prediction error responses, the authors should investigate the mechanisms underlying the difference observed between expected and unexpected familiar stimuli (A3 vs A4; Ext Fig 3), as shown for novel stimuli in figures 2, 3 & 5.

We agree with the reviewer that prediction error responses to familiar stimuli at an unexpected location were smaller than responses to completely novel stimuli. In our opinion this is not surprising given that novel stimuli are least expected. On the other hand, novel stimulus D, a stimulus quite different to the usually encountered grating stimuli in the corridor, evokes even stronger prediction error responses than grating C. Interestingly, V1 activity appears to therefore be scaled by how ‘surprising’ a stimulus is. But please note that the original figures showing responses to expected and unexpected stimulus A did not include neurons that exclusively respond to unexpected but not expected stimulus A, while such neurons (responding to unexpected but not expected stimulus C) were included in Figure 1 (together with neurons responding to expected C only or both expected and unexpected C). We now show a direct comparison with equivalent neuron inclusion criteria in Extended Data Figure 5d and 5h for unexpected vs expected C and A. While prediction error responses to A are indeed smaller than to novel stimulus C, we observe a clear, consistent and highly significant increase in V1 responses when familiar stimulus A is encountered at an unexpected location.

Importantly, as suggested by the reviewer, we have performed a large number of new experiments of imaging and optogenetically manipulating VIP neurons and pulvinar inputs to V1 as mice encounter familiar stimulus A at an unexpected location. These experiments show that both VIP neurons and pulvinar axons in V1 respond more strongly to the unexpected than the expected stimulus A (Extended Data Figure 11a-d, g-i). Importantly, silencing of either VIP neurons or pulvinar axons significantly decreases V1 responses to unexpected stimulus A (Extended Data Figure 11e,f,j,k). Moreover, prediction error responses to novel and familiar stimuli share similar properties, they are both highly specific such that only those visually-responsive neurons that are selective to the presented stimulus show amplification by prediction error (Figure 2 and Extended Data Figure 5 and 7). These findings show that similar circuit mechanisms underlie the amplification of both novel and familiar unexpected stimuli. The effect of pulvinar silencing on V1 responses to unexpected stimuli (A as well as C or D) was small. This is expected, given that, optogenetic pulvinar axon silencing only partially reduced pulvinar axon activity (new experiments, Extended Data Figure 12d-f).

- 2. If prediction error responses are an amplification of sensory responses, there should be a strong correlation between individual neuron responses to expected and unexpected stimulus C. Including this quantification (response to unexpected versus expected, by modifying fig. 2c) would strengthen this point.

A strong correlation between responses to expected and unexpected stimuli would only be expected if prediction errors cause a general, multiplicative gain change in V1 responses.

However, we show in our study that this is not the case, as only the most selective visually-responsive neurons have increased responses to unexpected stimuli, relatively independently of their response strength (Figure 2c,d, Extended Data Figure 5e and 7d). Moreover, as described now more comprehensively in the manuscript, while one component of the prediction error response is the amplification of selective sensory responses, prediction error signals in V1 also include responses of neurons that are only weakly or not visually-responsive (Extended Data Figure 6f-k, 8a-c). We have now clarified this in the manuscript.

- 3. It would be useful to compare the responses to unexpected stimulus during suppression of VIP cells versus the responses to the same stimulus when it is expected. The potential difference in responses will indicate the relative contribution of VIP neurons to prediction error responses. Same point for the suppression of pulvinar inputs.

Thanks for the suggestion, we have carried out this analysis. VIP silencing was very effective (see new Extended Data Figure 12a-c), indeed allowing us to determine the relative contribution of VIP neurons to V1 prediction error responses. V1 responses to an unexpected stimulus during suppression of VIP neurons and to the same stimulus when it was expected (without VIP suppression) were very similar for most neurons, indicating that VIP silencing removes most of the V1 prediction error signal (see Extended Data Fig. 13g). As shown in new Extended Data Figure 12d-f, suppression of pulvinar inputs was only partial, preventing us to determine their relative contribution to V1 prediction error responses.

- 4. Most statistical tests presented in the manuscript are done pulling neurons from different animals to increase statistical power. The statistics for right column's histograms should be done per animal (and the authors should show the distribution of each average value per animal). (e.g. it would be interesting to see whether it changes results in Fig 4c).

According to the reviewer's suggestion we have now changed the relevant plots to show individual animals and statistics are done per animal.

Minor issues:

- In lines 29 to 31 authors indicate that "mice interrupted their running behaviour when their expectations were violated by encountering grating C". This creates a difference between locomotion behaviour in the unexpected grating C compared with other gratings. Surprisingly, Ext. Fig 1 shows that animals specifically interrupted running when encountering grating C, irrespectively of whether this stimulus is expected or unexpected (Ext. Fig1b). Is the reduction of running speed really related to violated expectation? Showing running speed for block1, 2 early/late would be informative. Other point: Is running speed affected by the light used for optogenetic stimulation?

While many animals interrupted their running when encountering unexpected stimulus C, the trajectory of running behaviour was less consistent as the stimulus became expected late in the session. However, a subset of animals did recover their running behaviour in the second half of block 2 (see Rebuttal Fig. 1b). Moreover, in a new set of experiments with slightly altered

design (see Rebuttal Fig. 2a) that included an extended block 2 with more trials, all animals showed recovery of the deceleration induced by an unexpected stimulus (Rebuttal Figure 2b).

The animals' running speed was not affected by optogenetic laser stimulation (Extended Data Figure 15).

- In Fig3 responses to expected C and expected D stimuli are pooled together. Ext. Fig5 shows responses to each stimulus separately. A similar figure would be useful for the responses of pulvinar inputs.

We have now separated VIP and pulvinar responses to stimuli C and D (see Figure 3 and Extended Data Fig.13).

- Authors refer to pulvinar nucleus/ pulvinar inputs, that is usually used for the primate higher order thalamic nucleus, while the murine homologous is called lateral posterior nucleus. This could be clarified in the manuscript.

We use the term pulvinar instead of lateral posterior nucleus, since it is now established in the field that the mouse lateral posterior nucleus is homologous to the pulvinar in higher mammals, while lateral posterior nucleus refers to a different structure in some species. We have now clarified in the manuscript that pulvinar and lateral posterior nucleus are the same in mice.

Reviewer Reports on the First Revision:

Referees' comments:

Referee #1 (Remarks to the Author):

The authors have commendably addressed my previous concerns, as well as those raised by the other reviewers, leaving me with no major comments at this point.

Minor Comments:

1. I would suggest adding Rebuttal Figure 5 to the Extended Data Figures – also since Reviewer #2 asked for a similar analysis.
2. In Extended Data Fig. 5e, the legend should say “for all expected-grating-responsive cells” or similar (as it will not include neurons responsive to grating A4).
3. Line 467: “ccortex”

Referee #2 (Remarks to the Author):

The authors have addressed my questions and concerns satisfactorily. While I would have appreciated to see a somewhat more precise quantification of the ‘surprise’ behavior (e.g. using video tracking), allowing to explore the link between behavior, error signals and updating of the internal model in more detail, I understand that this angle goes a little beyond the scope of the present manuscript and is probably better addressed in a future study, as the authors suggest they are already planning. All in all, this is an impressive piece of work, providing important novel insights into how prediction errors are processed by, and integrated into sensory circuits.

Referee #3 (Remarks to the Author):

The authors have performed extensive new experiments in order to address the main concern of this reviewer. The results are convincing and the update of the manuscript is appropriate.

I have the following remaining suggestions:

- Figure 2c,d, Extended Data Figure 5e and 7d and Extended Data Figure 6f-k, 8a-c: as indicated by the authors, while one component of the prediction error response is the amplification of selective sensory responses, prediction error signals in V1 also include responses of neurons that are only weakly or not visually-responsive. It would be informative to indicate the fraction of selective neurons that increase their responses vs the fraction of non-selective neurons that respond to an unexpected stimulus among

all neurons that respond to prediction errors.

- Suppression of pulvinal inputs. The p-values for the effect of pulvinal axon silencing in responses to expected stimuli (fig 3m) are 0.074 and 0.088 with ~150-200 neurons, while the effect on unexpected stimuli is significant $P < 0.0004$ with over 500 neurons. To show that the effect on unexpected stimuli is stronger, the authors could compare the effect size of pulvinal axons suppression in expected vs unexpected stimuli responses.

- Check extended figure 9f/i. Do the cells in red in these panels correspond to cells selectively responding to either omission or D4, respectively, but not both? Some neurons do respond more strongly to the other condition.

**Author Rebuttals to First Revision:
Response to the reviewers, manuscript 2023-06-09795A**

We very much thank the reviewers for their encouraging comments. Below are our responses to the remaining points after the second review:

Referee #1:

The authors have commendably addressed my previous concerns, as well as those raised by the other reviewers, leaving me with no major comments at this point.

Minor Comments:

1. I would suggest adding Rebuttal Figure 5 to the Extended Data Figures – also since Reviewer #2 asked for a similar analysis.

We have now added this figure to Extended Data Figure 6.

2. In Extended Data Fig. 5e, the legend should say “for all expected-grating-responsive cells” or similar (as it will not include neurons responsive to grating A4).

We have now amended the figure legends accordingly.

3. Line 467: “ccortex”.

We have corrected this typo.

Referee #2:

The authors have addressed my questions and concerns satisfactorily. While I would have appreciated to see a somewhat more precise quantification of the ‘surprise’ behavior (e.g. using video tracking), allowing to explore the link between behavior, error signals and updating of the internal model in more detail, I understand that this angle goes a little beyond the scope of the present manuscript and is probably better addressed in a future study, as the authors suggest they are already planning. All in all, this is an impressive piece of work, providing important novel insights into how prediction errors are processed by, and integrated into sensory circuits.

We thank the reviewer for their positive feedback and understanding that such a study, which in itself would be a substantial amount of work, goes beyond the scope of the present manuscript.

Referee #3:

The authors have performed extensive new experiments in order to address the main concern of this reviewer. The results are convincing and the update of the manuscript is appropriate.

I have the following remaining suggestions:

- Figure 2c,d, Extended Data Figure 5e and 7d and Extended Data Figure 6f-k, 8a-c: as indicated by the authors, while one component of the prediction error response is the

amplification of selective sensory responses, prediction error signals in V1 also include responses of neurons that are only weakly or not visually-responsive. It would be informative to indicate the fraction of selective neurons that increase their responses vs the fraction of non-selective neurons that respond to an unexpected stimulus among all neurons that respond to prediction errors.

The corresponding numbers of neurons are described in the manuscript, for instance in Extended Data Figure 8, 482 neurons respond to the unexpected but not the expected stimulus C, 162 neurons respond to both the expected and unexpected stimulus C. But please note that these numbers depend strongly on the threshold used to define significant responsiveness which is relatively conservative in our study, such that many neurons that we define as not responsive to the stimulus when it is expected may have a weak or noisy response.

- Suppression of pulvinar inputs. The p-values for the effect of pulvinar axon silencing in responses to expected stimuli (fig 3m) are 0.074 and 0.088 with ~150-200 neurons, while the effect on unexpected stimuli is significant $P < 0.0004$ with over 500 neurons. To show that the effect on unexpected stimuli is stronger, the authors could compare the effect size of pulvinar axons suppression in expected vs unexpected stimuli responses.

We have now added effect size calculations to Figure 3m using Cohen's D. Please note that pulvinar silencing has a reverse effect on responses to expected stimuli A and B compared to unexpected C or D, with a larger, not smaller response when pulvinar is silenced (Cohen's D of -0.15).

- Check extended figure 9f/i. Do the cells in red in these panels correspond to cells selectively responding to either omission or D4, respectively, but not both? Some neurons do respond more strongly to the other condition.

No, in Extended Data Fig. 9e-g neurons that show a significant prediction error response to stimulus B omissions are highlighted, and in Extended Data Fig. 9h-j neurons that show a significant prediction error response to stimulus D when stimulus B is expected are highlighted, irrespective of how they respond to the other condition. Therefore, neurons are included if they respond to both omission and stimulus D or even more strongly to the other condition (although very few neurons do so). We apologize if this was unclear from the figure legends, we have now amended the text to make this clearer.